# Early Directional Convergence in Deep Homogeneous Neural Networks for Small Initializations

**Akshay Kumar**                                                                    *kumar511@umn.edu*
*Department of Electrical and Computer Engineering*
*University of Minnesota, Minneapolis, MN*

**Jarvis Haupt**                                                                    *jdhaupt@umn.edu*
*Department of Electrical and Computer Engineering*
*University of Minnesota, Minneapolis, MN*

**Reviewed on OpenReview:** *https://openreview.net/forum?id=VNM6V1gi3k*

## Abstract

This paper studies the gradient flow dynamics that arise when training deep homogeneous neural networks assumed to have *locally Lipschitz gradients* and an order of homogeneity strictly greater than two. It is shown here that for sufficiently small initializations, during the early stages of training, the weights of the neural network remain small in (Euclidean) norm and approximately converge in direction to the Karush-Kuhn-Tucker (KKT) points of the recently introduced *neural correlation function*. Additionally, this paper also studies the KKT points of the neural correlation function for feed-forward networks with (Leaky) ReLU and polynomial (Leaky) ReLU activations, deriving necessary and sufficient conditions for rank-one KKT points.

## 1 Introduction

Neural networks have achieved remarkable success across various tasks, yet the precise mechanism driving this success remains theoretically elusive. The training of neural networks involves optimizing a non-convex loss function, where the training algorithm typically is a first-order method such as gradient descent or its variants. A particularly puzzling aspect is how these training algorithms succeed in finding a solution with good generalization capabilities despite the non-convexity of the loss landscape. In addition to the choice of the training algorithm, the choice of initialization in these algorithms plays a crucial role in determining the neural network performance. Indeed, recent works have made increasingly clear the benefit of *small* initializations, revealing that neural networks trained using (stochastic) gradient descent with small initializations exhibit feature learning (Yang & Hu, 2021) and also generalize better for various tasks (Chizat et al., 2019; Geiger et al., 2020; Woodworth et al., 2020); see Section 2 for more details into the impact of initialization scale. However, for small initializations, the training dynamics of neural networks is extremely non-linear and not well understood so far. Our focus in this paper is on understanding the effect of small initialization on the training dynamics of neural networks.

In pursuit of a deeper understanding of the training mechanism for small initializations, researchers have uncovered the phenomenon of directional convergence in the neural network weights during the *early* phases of training (Maennel et al., 2018; Luo et al., 2021). The authors of Maennel et al. (2018) study the gradient flow dynamics of training two-layer Rectified Linear Unit (ReLU) neural networks, and demonstrate that in the early stages of training, the weights of two-layer ReLU neural networks converge in direction while their norms remain small. This phenomenon is referred to as *early directional convergence*. In a recent work, Kumar & Haupt (2024) further established early directional convergence for *two*-homogeneous[1] neural

---

[1] A neural network $\mathcal{H}$ is defined to be $L$-(positively) homogeneous, if its output $\mathcal{H}(\mathbf{x}; \mathbf{w})$ satisfies $\mathcal{H}(\mathbf{x}; c\mathbf{w}) = c^L \mathcal{H}(\mathbf{x}; \mathbf{w}), \forall c \geq 0$, where $\mathbf{x}$ is the input and $\mathbf{w}$ is a vector containing all the weights.

networks, illustrating the importance of homogeneity for such types of phenomenon (as opposed to the specific choice of activation function or architecture).

To describe the phenomenon of early directional convergence for two-homogeneous neural networks, Kumar & Haupt (2024) introduced the notion of the *Neural Correlation Function* (NCF). For a given neural network and a vector, the NCF quantifies the correlation between the output of neural network and the vector; we provide a more precise definition in Section 3. They show that in the early stages of training, the weights of the neural network remain small and converge in direction to a KKT point of the *constrained NCF*, an optimization problem that maximizes the NCF on unit sphere.

So far, all the works on the early stages of training dynamics of neural networks for small initialization have only considered two-layer neural networks (or more broadly, two-homogeneous networks). In contrast, our understanding for *deep* neural networks is much more limited. In the first part of this work, we establish early directional convergence for *deep* homogeneous neural networks. We particularly consider $L$-homogeneous neural networks, where $L > 2$, that have *locally Lipschitz gradients*, and our main result can be summarized as follows:

- In Theorem 1 we describe the gradient flow dynamics of deep homogeneous neural networks, with order of homogeneity strictly greater than two, trained with small initializations. Specifically, for square and logistic losses, we show that if the initialization is sufficiently small, then in the early stages of training the weights remain small and either approximately converge in direction to a non-negative KKT point of the constrained NCF defined with respect to the labels in the training set, or are approximately zero.

It is worth noting that we assume neural networks to have locally Lipschitz gradients, which excludes deep ReLU networks. However, it does include deep linear networks (Arora et al., 2019a; Saxe et al., 2014), and deep neural networks with differentiable and homogeneous activation such as polynomial ReLU $\phi(x) := \max(x, 0)^p$ for $p \geq 2$ (Gribonval et al., 2022; Klusowski & Barron, 2018; Zhong et al., 2017) and monomials (Livni et al., 2014; Soltanolkotabi et al., 2019). We also discuss the challenges in extending our results for deep ReLU networks in Appendix D.

The second part of our paper studies the KKT points of the constrained NCF for feed-forward homogeneous neural networks. For such networks, we observe empirically that, along with converging towards a KKT point of the constrained NCF during the early stages of training, the hidden weights also exhibit a *rank one* structure. Motivated by this observation, we mathematically characterize the rank-one KKT points of feed-forward homogeneous neural networks. Our second main result can be summarized as follows:

- In Section 5, we derive sufficient conditions for a rank-one KKT point of the constrained NCF for feed-forward neural network with activation function $\sigma(x) = \max(x, \alpha x)^p$, for some $p \in \mathbb{N}$, $\alpha \in \mathbb{R}$ and arbitrary training data. We also provide matching necessary conditions under additional assumptions and validate these assumptions empirically. Our results also provide a method to obtain rank-one KKT points of the constrained NCF using KKT points of a smaller optimization problem.

## 2 Related Works

Understanding the training dynamics of neural networks has been the subject of numerous investigations, including (Arora et al., 2019b; Chizat & Bach, 2020; Chizat et al., 2019; Jacot et al., 2018; Mei et al., 2019). These investigations have revealed two contrasting viewpoints depending on the scale of initialization. For large initialization regimes, the training dynamics of wide neural networks are effectively captured by a kernel referred to as Neural Tangent Kernel (NTK) Jacot et al. (2018). In this regime, the weights remain close to initialization during training, and neural networks behave like their linearizations around their initializations (Chizat et al., 2019). On the other hand, in small initialization regimes, the training dynamics are extremely non-linear, and the change in weights during training is significant (Chizat & Bach, 2018; Geiger et al., 2020; Mei et al., 2019; Yang & Hu, 2021). Moreover, studies like Chizat et al. (2019); Geiger et al. (2020) have observed improved generalization with decreasing initialization scale for various tasks, which makes understanding the dynamics of neural networks in the small initialization regime crucial for better understanding of practical success of deep networks.

While considerable progress has been made on understanding the dynamics of neural networks in the large initialization regime, the theoretical investigations into the small initialization regime have been comparatively limited, primarily due to the extremely non-linear dynamics during training. Such investigations have predominantly focused on linear neural networks (Arora et al., 2019a; Gunasekar et al., 2017; Woodworth et al., 2020) and shallow non-linear neural networks (Chizat & Bach, 2018; 2020; Mei et al., 2019; Rotskoff & Vanden-Eijnden, 2018). Even for these networks, a comprehensive understanding of the gradient descent dynamics is further limited to diagonal linear networks (Woodworth et al., 2020), shallow matrix factorization (Jin et al., 2023; Stöger & Soltanolkotabi, 2021), and two-layer (Leaky) ReLU networks under various simplifying assumptions on the datasets (Boursier et al., 2022; Brutzkus & Globerson, 2019; Lyu et al., 2021; Min et al., 2024; Wang & Ma, 2023). Our work could be the first step towards a greater understanding of the training dynamics of deep neural networks.

The phenomenon of early directional convergence for small initializations was first investigated by Maennel et al. (2018) for two-layer ReLU neural networks. This was later extended to two-homogeneous neural networks by Kumar & Haupt (2024), which includes, in addition to two-layer ReLU neural networks, single layer squared ReLU neural networks and deep ReLU neural networks with only two trainable layers. Other empirical investigations have observed early directional convergence in some deeper networks as well. For example, Zhou et al. (2022) looks at the early training dynamics of three-layer ReLU neural networks and observe that weights converge along some specific orientations. Also, Atanasov et al. (2022) empirically show an alignment effect in the kernel of the neural network during early stages of the training. However, rigorous investigations thus far have been limited to shallow neural networks. Despite focusing solely on the early phases of training, these works demonstrate that interesting behaviors begin to emerge very early in the training dynamics. Moreover, these insights into the initial stages of training have been useful towards a comprehensive understanding of the entire training dynamics in some special cases (Boursier et al., 2022; Min et al., 2024; Lyu et al., 2021; Wang & Ma, 2023).

## 3 Problem Formulation

**Notation:** For any $N \in \mathbb{N}$, we let $[N] = \{1, 2, \ldots, N\}$ denote the set of positive integers less than or equal to $N$. We let $\| \cdot \|_2$ denote the $\ell_2$ norm for a vector and the spectral norm for a matrix. For a vector $\mathbf{z} \in \mathbb{R}^n$, $z_i$ denotes its $i$th entry, $|\mathbf{z}| = [|z_1|, |z_2|, \cdots, |z_n|]^\top$, and $\mathbf{z}^q = [z_1^q, z_2^q, \cdots, z_n^q]^\top$, where $q \in \mathbb{N}$. We write $\dot{x}(t) =: \frac{dx(t)}{dt}$, and for the sake of brevity we may remove the independent variable $t$ if it is clear from context. For any locally Lipschitz continuous function $f : X \to \mathbb{R}$, its Clarke sub-differential is denoted by $\partial f(\mathbf{x})$. For $\sigma : \mathbb{R} \to \mathbb{R}$ and $p \in \mathbb{N}$, $\sigma^p(\cdot)$ denotes the function resulting from composing $\sigma(\cdot)$ with itself $p$ times, and we assume $\sigma^0(x) = x$. We define a KKT point of an optimization problem to be a non-negative (non-zero) KKT point if the objective value at that KKT point is non-negative (non-zero).

**Problem setup:** We adopt a supervised learning framework for training, where we assume $\{\mathbf{x}_i, y_i\}_{i=1}^n$ is the training dataset, and let $\mathbf{X} = [\mathbf{x}_1, \ldots, \mathbf{x}_n] \in \mathbb{R}^{d \times n}$ and $\mathbf{y} = [y_1, \ldots, y_n]^\top \in \mathbb{R}^n$. For a neural network $\mathcal{H}$, its output is denoted by $\mathcal{H}(\mathbf{x}; \mathbf{w})$, where $\mathbf{x}$ is the input and $\mathbf{w} \in \mathbb{R}^k$ is the vector containing all the weights, and $\nabla \mathcal{H}(\mathbf{x}; \mathbf{w})$ denotes the gradient of $\mathcal{H}(\mathbf{x}; \mathbf{w})$ with respect to $\mathbf{w}$. We let $\mathcal{H}(\mathbf{X}; \mathbf{w}) = [\mathcal{H}(\mathbf{x}_1; \mathbf{w}), \ldots, \mathcal{H}(\mathbf{x}_n; \mathbf{w})]^\top \in \mathbb{R}^n$ be the vector containing the output of neural network for all inputs, and $\mathcal{J}(\mathbf{X}; \mathbf{w}) : \mathbb{R}^k \to \mathbb{R}^n$ denotes the Jacobian of $\mathcal{H}(\mathbf{X}; \mathbf{w})$ with respect to $\mathbf{w}$.

The following assumptions formalize the properties of neural networks considered in this paper.

**Assumption 1.** *We make the following tripartite assumption on the neural network function:* (i) *For any fixed* $\mathbf{x}$*,* $\mathcal{H}(\mathbf{x}; \mathbf{w})$ *is locally Lipschitz and is definable under some o-minimal structure that includes polynomials and exponential.*[2] (ii) *For all* $c > 0$*,* $\mathcal{H}(\mathbf{x}; c\mathbf{w}) = c^L \mathcal{H}(\mathbf{x}; \mathbf{w})$*, for some* $L > 2$ *(i.e., the network is* $L$*-homogenous).* (iii) $\mathcal{H}(\mathbf{x}; \mathbf{w})$ *is differentiable in* $\mathbf{w}$ *and its gradient,* $\nabla \mathcal{H}(\mathbf{x}; \mathbf{w})$*, is locally Lipschitz.*

---

[2]We note that definability in some o-minimal structure is a mild technical assumption and is satisfied by all modern deep neural networks. It allows us to ensure convergence of certain gradient flow trajectories (Ji & Telgarsky, 2020).

In this paper, we consider the minimization of

$$\mathcal{L}(\mathbf{w}) = \sum_{i=1}^{n} \ell\left(\mathcal{H}(\mathbf{x}_i; \mathbf{w}), y_i\right), \tag{1}$$

where $\ell(\hat{y}, y)$ is the loss function, and $\nabla_{\hat{y}}\ell(\hat{y}, y)$ is assumed to be locally Lipschitz in $\hat{y}$, which is satisfied by typical loss functions such as square and logistic losses. For $\mathbf{z}, \mathbf{y} \in \mathbb{R}^n$, we define $\ell'(\mathbf{z}, \mathbf{y}) = [\nabla_{\hat{y}}\ell(z_1, y_1), \ldots, \nabla_{\hat{y}}\ell(z_n, y_n)]^\top \in \mathbb{R}^n$. We minimize eq. (1) using gradient flow, which leads to the following differential equation

$$\dot{\mathbf{w}}(t) = -\nabla\mathcal{L}(\mathbf{w}(t)), \mathbf{w}(0) = \delta\mathbf{w}_0, \tag{2}$$

where $\delta$ is a positive scalar that controls the scale of initialization, and $\mathbf{w}_0$ is a vector.

Following Kumar & Haupt (2024), for a fixed vector $\mathbf{z} \in \mathbb{R}^n$ and neural network $\mathcal{H}$, the Neural Correlation Function (NCF) is defined here as

$$\mathcal{N}_{\mathbf{z},\mathcal{H}}(\mathbf{w}) = \mathbf{z}^\top \mathcal{H}(\mathbf{X}; \mathbf{w}),$$

and measures the correlation between the vector $\mathbf{z}$ and the output of the neural network. The constrained NCF refers to the following constrained optimization problem

$$\max_{\|\mathbf{w}\|_2^2 = 1} \mathcal{N}_{\mathbf{z},\mathcal{H}}(\mathbf{w}).$$

## 4 Early Directional Convergence

### 4.1 Main Result

The following theorem establishes the approximate directional convergence in the early stages of training $L$-homogeneous neural networks with small initializations.

**Theorem 1.** *Let $\mathbf{w}_0$ be a fixed unit vector. Under Assumption 1, for any $\epsilon \in (0, \eta/2)$, where $\eta$ is a positive constant, there exists $T_\epsilon, \tilde{B}_\epsilon$ and $\overline{\delta} > 0$ such that the following holds: for any $\delta \in (0, \overline{\delta})$ and solution $\mathbf{w}(t)$ of eq. (2) with initialization $\mathbf{w}(0) = \delta\mathbf{w}_0$, we have*

$$\|\mathbf{w}(t)\|_2 \leq \tilde{B}_\epsilon \delta, \text{ for all } t \in \left[0, T_\epsilon/\delta^{L-2}\right].$$

*Further, for $\overline{T}_\epsilon = T_\epsilon/\delta^{L-2}$, either*

$$\|\mathbf{w}(\overline{T}_\epsilon)\|_2 \geq \delta\eta/2, \text{ and } \mathbf{u}_*^\top \mathbf{w}(\overline{T}_\epsilon)/\|\mathbf{w}(\overline{T}_\epsilon)\|_2 \geq 1 - (1 + 3/\eta)\,\epsilon,$$

*where $\mathbf{u}_*$ is a non-negative KKT point of*

$$\max_{\|\mathbf{u}\|_2^2 = 1} \mathcal{N}_{-\ell'(\mathbf{0},\mathbf{y}),\mathcal{H}}(\mathbf{u}) = \max_{\|\mathbf{u}\|_2^2 = 1} -\ell'(\mathbf{0}, \mathbf{y})^\top \mathcal{H}(\mathbf{X}; \mathbf{u}), \tag{3}$$

*or $\|\mathbf{w}(\overline{T}_\epsilon)\|_2 \leq 2\delta\epsilon$.*

The above theorem first shows that for a given $\epsilon$, we can choose $\delta$ small enough such that for $t \in [0, T_\epsilon/\delta^{L-2}]$ the norm of the weights increases at most by a multiplicative factor. Thus, for small $\delta$ the weights will remain small for $t \in [0, T_\epsilon/\delta^{L-2}]$. Furthermore, since $L > 2$, for smaller $\delta$ the weights will stay small for a longer duration of time.

Next, the theorem describes what happens at $\overline{T}_\epsilon = T_\epsilon/\delta^{L-2}$, the duration for which the norm of the weights remain small. There are two possible scenarios. In the first scenario, the weights approximately converge in direction along a non-negative KKT point of eq. (3), which is the constrained NCF defined with respect to $-\ell'(\mathbf{0}, \mathbf{y})$ and neural network $\mathcal{H}$. Thus, in the early stages of training, the weights of the neural network converge in direction while their norm remains small. In the second scenario, the weights approximately converge to $\mathbf{0}$, which can be observed by noting that $\|\mathbf{w}(\overline{T}_\epsilon)\|_2 \leq 2\delta\epsilon$, where we can choose $\epsilon$ to be arbitrarily

small. This is in contrast with the first scenario, where $\|\mathbf{w}(\overline{T}_\epsilon)\|_2 \geq \delta\eta/2$ and $\eta$ is a constant that does not depend on $\delta$. This shows that in the second scenario, compared to the first scenario, the weights become much smaller. The second scenario arises due to the convergence of the gradient dynamics of the NCF towards $\mathbf{0}$; see the proof for more details. We also note that for a fixed training data and neural network, $\mathbf{w}_0$ determines which of the above two scenarios will occur. In addition to this, the positive constant $\eta$ in the above theorem is also determined by $\mathbf{w}_0$ and is independent of $\delta$.

It can be shown that for square loss, $\ell(\hat{y}, y) = \frac{1}{2}(\hat{y} - y)^2$ and $-\ell'(0, \mathbf{y}) = \mathbf{y}$, and for logistic loss, $\ell(\hat{y}, y) = \ln(1 + e^{-\hat{y}y})$ and $-\ell'(0, \mathbf{y}) = \mathbf{y}/2$. Therefore, for square and logistic loss, the objective function of the constrained NCF in eq. (3) will be $\mathbf{y}^\top \mathcal{H}(\mathbf{X}; \mathbf{u})$. We also study KKT points of the constrained NCF for feed-forward neural networks in Section 5, which some readers may find helpful.

Overall, the phenomenon of early directional convergence in deep homogeneous networks is similar to two-homogeneous networks (Kumar & Haupt, 2024, Theorem 5.1), except for one difference. The amount of time required for directional convergence in deep homogeneous networks has an additional factor of $1/\delta^{L-2}$. Thus, directional convergence takes longer as depth increases.

### 4.1.1 Proof Sketch of Theorem 1

We next provide a brief proof sketch of the above theorem. We begin by describing the (positive) gradient flow dynamics of the NCF.

For a given vector $\mathbf{z}$ and a neural network $\mathcal{H}$, the gradient flow resulting from maximizing the NCF is

$$\dot{\mathbf{u}} = \nabla \mathcal{N}_{\mathbf{z}, \mathcal{H}}(\mathbf{u}), \mathbf{u}(0) = \mathbf{u}_0, \tag{4}$$

where $\mathbf{u}_0$ is the initialization. Note that the NCF may not be bounded from above. Therefore, gradient flow may potentially diverge to infinity. However, the following lemma shows that the gradient flow either converges in direction to a KKT point of the constrained NCF, or converges to $\mathbf{0}$.

**Lemma 2.** *Under Assumption 1, for any solution $\mathbf{u}(t)$ of eq. (4), one of the following two possibilities is true.*

1. *There exists some finite $T^*$ such that $\lim_{t \to T^*} \|\mathbf{u}(t)\|_2 = \infty$. Further, $\lim_{t \to T^*} \mathbf{u}(t)/\|\mathbf{u}(t)\|_2$ exists and is equal to a non-negative KKT point of the optimization problem*

$$\max_{\|\mathbf{u}\|_2^2 = 1} \mathcal{N}_{\mathbf{z}, \mathcal{H}}(\mathbf{u}) = \mathbf{z}^\top \mathcal{H}(\mathbf{X}; \mathbf{u}). \tag{5}$$

2. *For all $t \geq 0$, $\|\mathbf{u}(t)\|_2 < \infty$, and either $\lim_{t \to \infty} \mathbf{u}(t)/\|\mathbf{u}(t)\|_2$ exists or $\lim_{t \to \infty} \mathbf{u}(t) = 0$. If $\lim_{t \to \infty} \mathbf{u}(t)/\|\mathbf{u}(t)\|_2$ exists then it is equal to non-negative KKT point of eq. (5).*

To prove the above lemma, we require the neural networks to be definable with respect to an o-minimal structure. Such a requirement allows us to use the unbounded version of Kurdyka-Lojasiewicz inequality proved in Ji & Telgarsky (2020) for establishing directional convergence. A similar result was demonstrated in Kumar & Haupt (2024) for two-homogeneous networks. However, in that instance, the solution $\mathbf{u}(t)$ remains finite for all finite time (see Lemma 18), whereas in our case, $\mathbf{u}(t)$ may become unbounded at some finite time. Also, for our proof, we had to derive the rate at which $\mathbf{u}(t)$ grows, showing that, if $\mathbf{u}(t)$ becomes unbounded, then $\|\mathbf{u}(t)\|_2 \geq \kappa/(T^* - t)^{1/(L-2)}$, for some $\kappa > 0$ and for all $t$ near $T^*$.

We next explain our proof technique for Theorem 1 assuming square loss is used for training. The evolution of $\mathbf{w}(t)$ is governed by the differential equation

$$\dot{\mathbf{w}} = \sum_{i=1}^n (y_i - \mathcal{H}(\mathbf{x}_i; \mathbf{w})) \nabla \mathcal{H}(\mathbf{x}_i; \mathbf{w}) = \mathcal{J}(\mathbf{X}; \mathbf{w})^\top (\mathbf{y} - \mathcal{H}(\mathbf{X}; \mathbf{w})), \mathbf{w}(0) = \delta\mathbf{w}_0, \tag{6}$$

where (as mentioned) $\delta > 0$, and recall $\mathcal{J}(\mathbf{X}; \mathbf{w}) : \mathbb{R}^k \to \mathbb{R}^n$ is the Jacobian of the function $\mathcal{H}(\mathbf{X}; \mathbf{w})$ with respect to $\mathbf{w}$. We define $\mathbf{s}(t) = \frac{1}{\delta}\mathbf{w}\left(\frac{t}{\delta^{L-2}}\right)$; then, $\mathbf{s}(0) = \mathbf{w}_0$. Further, by using the $L$-homogeneity of $\mathcal{H}(\mathbf{X}; \mathbf{w})$

and $(L-1)$-homogeneity of $\mathcal{J}(\mathbf{X}; \mathbf{w})$ (Lemma 9), we can derive the evolution of $\mathbf{s}(t)$ as follows

$$
\frac{d\mathbf{s}}{dt} = \frac{d}{dt}\left(\frac{1}{\delta}\mathbf{w}\left(\frac{t}{\delta^{L-2}}\right)\right) = \frac{1}{\delta^{L-1}}\dot{\mathbf{w}}\left(\frac{t}{\delta^{L-2}}\right)
$$

$$
= \frac{1}{\delta^{L-1}}\mathcal{J}\left(\mathbf{X}; \mathbf{w}\left(\frac{t}{\delta^{L-2}}\right)\right)^{\top}\left(\mathbf{y} - \mathcal{H}\left(\mathbf{X}; \mathbf{w}\left(\frac{t}{\delta^{L-2}}\right)\right)\right)
$$

$$
= \mathcal{J}\left(\mathbf{X}; \frac{1}{\delta}\mathbf{w}\left(\frac{t}{\delta^{L-2}}\right)\right)^{\top}\left(\mathbf{y} - \delta^{L}\mathcal{H}\left(\mathbf{X}; \frac{1}{\delta}\mathbf{w}\left(\frac{t}{\delta^{L-2}}\right)\right)\right).
$$

Overall, this implies

$$
\dot{\mathbf{s}} = \mathcal{J}(\mathbf{X}; \mathbf{s})^{\top}(\mathbf{y} - \delta^{L}\mathcal{H}(\mathbf{X}; \mathbf{s})), \mathbf{s}(0) = \mathbf{w}_0. \tag{7}
$$

Now, we compare the dynamics for $\mathbf{s}(t)$ in eq. (7) with the dynamics of gradient flow of the NCF defined with respect to $\mathbf{y}$ and neural network $\mathcal{H}$,

$$
\dot{\mathbf{u}} = \nabla\mathcal{N}_{\mathbf{y}, \mathcal{H}}(\mathbf{u}) = \mathcal{J}(\mathbf{X}; \mathbf{u})^{\top}\mathbf{y}, \mathbf{u}(0) = \mathbf{w}_0. \tag{8}
$$

Observe that if $\delta = 0$, then the two dynamics are exactly the same, and since $\mathbf{s}(0) = \mathbf{w}_0 = \mathbf{u}(0)$, we have that $\mathbf{s}(t)$ and $\mathbf{u}(t)$ will also be same. For $\delta > 0$, there is an extra term $\delta^{L}\mathcal{H}(\mathbf{X}; \mathbf{s})$ present in eq. (7). For small $\delta$, we show that this extra term is small and while $\mathbf{s}(t)$ and $\mathbf{u}(t)$ are not exactly the same, the difference between them is small, i.e., $\|\mathbf{s}(t) - \mathbf{u}(t)\|_2$ is small. Moreover, we show that the difference can be made sufficiently small for a sufficiently long time by choosing $\delta$ small enough. We combine this fact with Lemma 2 to prove Theorem 1. Specifically, from Lemma 2, we know that $\mathbf{u}(t)$ will either converge to $\mathbf{0}$ or converge in direction to a KKT point of the constrained NCF. Therefore, we first choose a time (say $T$) such that $\mathbf{u}(T)$ has approximately converged in direction to a KKT point of the constrained NCF or is in the neighborhood of $\mathbf{0}$. Then, we choose $\delta$ sufficiently small such that $\|\mathbf{s}(t) - \mathbf{u}(t)\|_2$ is small for all $t \in [0, T]$. Since $\mathbf{s}(t) = \frac{1}{\delta}\mathbf{w}\left(\frac{t}{\delta^{L-2}}\right)$, we have that $\|\frac{1}{\delta}\mathbf{w}\left(\frac{T}{\delta^{L-2}}\right) - \mathbf{u}(T)\|_2$ is also small. Finally, since $\delta$ is a positive scalar, and $\mathbf{u}(T)$ approximately converges in direction to a KKT point of the constrained NCF or is approximately $\mathbf{0}$, it follows that $\mathbf{w}\left(\frac{T}{\delta^{L-2}}\right)$ also approximately converges in direction to a KKT point of the constrained NCF or is approximately $\mathbf{0}$.

As a quick aside, we note that there are two key aspects in the definition $\mathbf{s}(t) = \frac{1}{\delta}\mathbf{w}\left(\frac{t}{\delta^{L-2}}\right)$. First, we have divided by $\delta$. We know that $\delta$ controls the scale of initialization and is assumed to be small. Dividing by $\delta$ removes that scaling and makes $\mathbf{s}(t)$ a constant scale vector. More importantly, dividing by $\delta$ does not alter the direction, i.e., the direction of $\mathbf{s}(t)$ and $\mathbf{w}(t)$ are always the same. Thus, establishing directional convergence of $\mathbf{s}(t)$ will also imply directional convergence of $\mathbf{w}(t)$. Second, we rescale the time by $\delta^{L-2}$. To understand the importance of this time rescaling, let us look at the magnitude of the gradient of the loss at initialization in terms of $\delta$. Since $\delta$ is small and $\mathbf{w}_0$ has unit norm, $\|\mathcal{J}(\mathbf{X}; \mathbf{w}(0))^{\top}(\mathbf{y} - \mathcal{H}(\mathbf{X}; \mathbf{w}(0)))\|_2 \approx \|\mathcal{J}(\mathbf{X}; \mathbf{w}(0))^{\top}\mathbf{y}\|_2 = O(\delta^{L-1})$, where in the last equality we used $(L-1)$-homogeneity of the Jacobian. We observe that for initialization of scale $\delta$, the gradient scales as $\delta^{L-1}$. Thus, in the early stages the gradient will have negligible effect on the weights, in terms of both magnitude and direction. Moreover, based on the relative scaling of the gradient and the initialization, the gradient will only start impacting the weights after $O(1/\delta^{L-2})$ time has elapsed. Hence, for smaller initialization, the gradient flow will take longer to make an impact on the weights. Thus, in a way, rescaling time by $\delta^{L-2}$ brings the time of $\mathbf{s}(t)$ to a "constant scale". We also highlight the importance of homogeneity. It allows the above two changes to interact nicely with the gradient flow and results in dynamics of $\mathbf{s}(t)$ that are essentially independent of $\delta$, provided $\delta$ is small.

Finally, in comparison to Kumar & Haupt (2024), which studies early directional convergence for two-homogeneous neural networks, the main conceptual innovation in this paper is the time rescaling by $\delta^{L-2}$. In fact, the proof in that work relies on showing that the difference between $\mathbf{w}(t)/\delta$ and $\mathbf{u}(t)$ remains small for a sufficiently long time. To see a motivation for this, note that we can get eq. (7) from eq. (6) for $L = 2$ as well. However, for $L = 2$, the time rescaling factor $\delta^{L-2}$ will become unity, yielding $\mathbf{s}(t) = \mathbf{w}(t)/\delta$. Now, since the gradient flow dynamics for the NCF of two-homogeneous neural networks also converges to a KKT point of the constrained NCF, to establish early directional convergence one must show that the difference between $\mathbf{w}(t)/\delta$ and $\mathbf{u}(t)$ remains small for a sufficiently long time. Another notable difference with Kumar & Haupt (2024) is in the gradient flow dynamics of the NCF. As mentioned previously, from Lemma 2, we know that

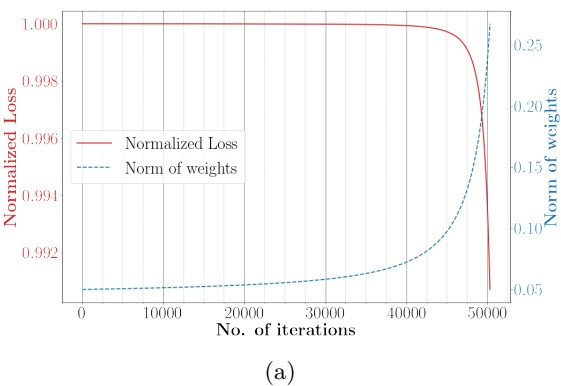
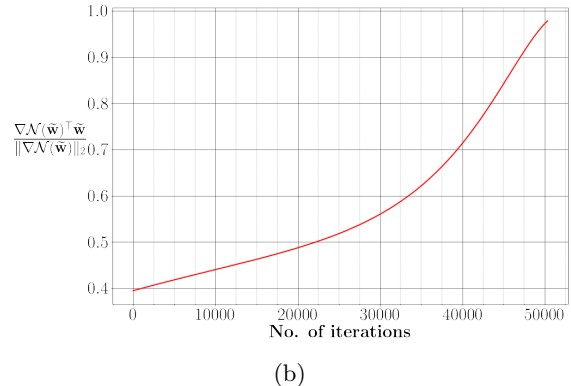

(a)

(b)

Figure 1: Early training dynamics of a squared-ReLU neural network. Panel ($a$): the evolution of training loss (normalized with respect to loss at initialization) and the $\ell_2$-norm of all the weights with iterations. Panel ($b$): the evolution of $\nabla\mathcal{N}(\widetilde{\mathbf{w}}(t))^\top\widetilde{\mathbf{w}}(t)/\|\nabla\mathcal{N}(\widetilde{\mathbf{w}}(t))\|_2$ (a measure of directional convergence of $\mathbf{w}(t)$). Note that the loss has barely changed, and the norm of weights has increased by a constant factor but remains small. Also, the weights approximately converge in direction to a KKT point of the constrained NCF.

for $L > 2$, the gradient flow dynamics of the NCF can become unbounded at some finite time, whereas for $L = 2$, the dynamics remains finite for finite time. This difference necessitates a more careful analysis of the early stages of training in order to ensure directional convergence for deep homogeneous neural networks.

**Challenges for non-differentiable neural networks:** Theorem 1 holds for neural networks with locally Lipschitz gradient, and thus excludes ReLU neural networks. We briefly describe the challenges in extending Theorem 1 to ReLU networks in Appendix D, which also includes an empirical evidence that early directional convergence also occurs in ReLU networks.

### 4.1.2 A Closer Look at the Weights

In this section, we use a toy example to highlight the emergence of additional structural properties in the weights during the early stage of training that are not explained by Theorem 1.

We train the following $3-$homogeneous neural network $\mathcal{H}(\mathbf{x}; \mathbf{v}, \mathbf{U}) = \mathbf{v}^\top \max(\mathbf{U}\mathbf{x}, \mathbf{0})^2$, where $\mathbf{v} \in \mathbb{R}^{20\times 1}$ and $\mathbf{U} \in \mathbb{R}^{20\times 10}$, by minimizing the square loss with respect to the output of a smaller neural network $\mathcal{H}^*(\mathbf{x}; \mathbf{v}_*, \mathbf{U}_*) = \mathbf{v}_*^\top \max(\mathbf{U}_*\mathbf{x}, \mathbf{0})^2$, where the entries of $\mathbf{v}_* \in \mathbb{R}^{2\times 1}$, $\mathbf{U}_* \in \mathbb{R}^{2\times 10}$ are drawn from the standard normal distribution. The training data has 100 points sampled uniformly from unit sphere in $\mathbb{R}^{10}$. Let $\mathbf{w}$ be a vector containing all the entries of $\mathbf{v}$ and $\mathbf{U}$, and $\widetilde{\mathbf{w}} = \mathbf{w}/\|\mathbf{w}\|_2$. We train for 50360 iterations using gradient descent with step-size $2 \cdot 10^{-2}$, and the initialization $\mathbf{w}(0) = \delta\mathbf{w}_0$, where $\delta = 0.05$ and $\mathbf{w}_0$ is a random unit norm vector. Figure 1a depicts the evolution of training loss and $\ell_2-$norm of the weights. Note that the loss almost remains the same, and the norm of the weights has increased but only by a constant factor and is still small. A unit norm vector is a KKT point of the constrained NCF if the vector and the gradient of the NCF at that vector are parallel (see Lemma 11). Hence, to measure the directional convergence of $\mathbf{w}$ to the KKT point of the constrained NCF, in Figure 1b we plot the evolution of the inner product between $\widetilde{\mathbf{w}}(t)$ and $\nabla\mathcal{N}(\widetilde{\mathbf{w}}(t))/\|\nabla\mathcal{N}(\widetilde{\mathbf{w}}(t))\|_2$, where $\mathcal{N}(\cdot)$ denotes the NCF for this example. Clearly, in accordance with Theorem 1, the weights approximately converge in direction to a KKT point of the constrained NCF.

We next take a closer look at the weights of the neural networks at iteration 50360, i.e., when the weights have approximately converged in direction to a KKT point of the constrained NCF. In Figure 2, we plot the normalized absolute value of the weights at initialization and at iteration 50360. At initialization the weights appear to be random, as expected, but at iteration 50360, they are more structured. The hidden weights ($\mathbf{U}$) appear to be of rank-one as only one row is non-zero, and the outer weight ($\mathbf{v}$) has a single non-zero entry. This suggests that weights converged towards a KKT point of the constrained NCF that have low-rank hidden weights. Similar behavior is observed with deeper networks and other activation functions as well (see Section 5.3), and has also been observed in some previous works (Zhou et al., 2022; Atanasov et al., 2022).

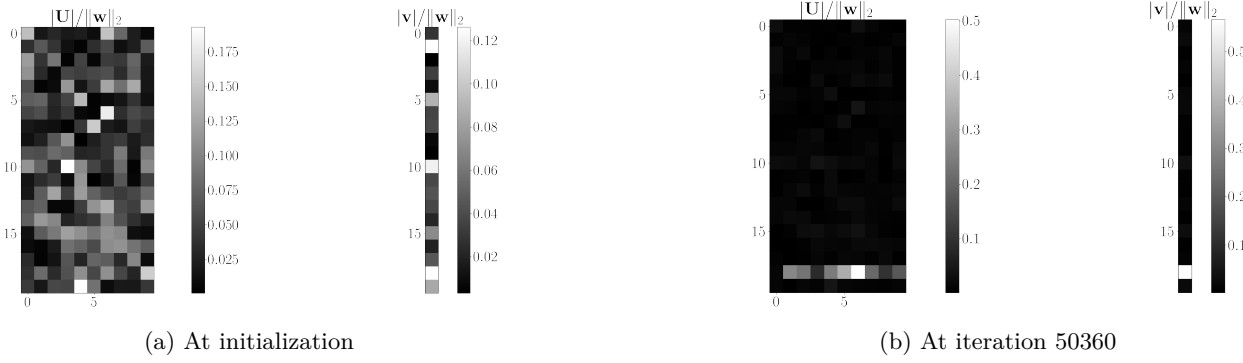

(a) At initialization

(b) At iteration 50360

Figure 2: Normalized absolute value of the weights at different stages of training.

Since the constrained NCF is a non-convex problem with potentially multiple KKT points, an important question arising from the above observation is why gradient descent converges towards low-rank KKT points. A more basic question, addressed in the next section, is whether low-rank KKT points always exist for the constrained NCF, and can they be characterized? Given the complex nature of the problem, their existence is not immediately clear, especially for deep neural networks. We next study the KKT points of the constrained NCF for feed-forward neural networks with Leaky ReLU and polynomial Leaky ReLU activation functions. For arbitrary training data, necessary and sufficient conditions are derived for a set of weights to be a KKT point of the NCF that have rank-one hidden weights.

## 5    KKT Points of Constrained NCF

This section presents necessary and sufficient conditions for a set of weights to be a KKT point of the constrained NCF of feed-forward neural network that have rank-one hidden weights. For a given input $\mathbf{x}$, the output of an $L$-layer feed-forward neural network $\mathcal{H}$ is defined as

$$\mathcal{H}(\mathbf{x}; \mathbf{W}_1, \cdots, \mathbf{W}_L) = \mathbf{W}_L \sigma\left(\mathbf{W}_{L-1} \cdots \sigma\left(\mathbf{W}_1 \mathbf{x}\right) \cdots\right) \in \mathbb{R}, \tag{9}$$

where $\mathbf{W}_l \in \mathbb{R}^{k_l \times k_{l-1}}$ are the trainable weights, and $k_0 = d$ and $k_L = 1$. The activation function $\sigma(\cdot)$ is applied coordinate-wise to vectors, and in this section is assumed to be of the form $\sigma(x) = \max(\alpha x, x)^p$, where $p \in \mathbb{N}$ and $\alpha \in \mathbb{R}$, which includes deep ReLU and polynomial ReLU neural networks. The corresponding constrained NCF is the following optimization problem:

$$\max_{\mathbf{W}_1, \cdots, \mathbf{W}_L} \mathcal{N}(\mathbf{W}_1, \cdots, \mathbf{W}_L) := \mathbf{y}^\top \mathcal{H}(\mathbf{X}; \mathbf{W}_1, \cdots, \mathbf{W}_L), \text{ s.t. } \|\mathbf{W}_1\|_F^2 + \cdots + \|\mathbf{W}_L\|_F^2 = 1. \tag{10}$$

Note that, we have considered the NCF in the above equation with respect to $\mathbf{y}$ for convenience, it may be replaced with any other vector.

### 5.1    Leaky ReLU

We first consider the activation function of the form $\sigma(x) = \max(x, \alpha x)$, for some $\alpha \in \mathbb{R}$.

**Theorem 3.** *Let $\mathcal{H}$ be an $L-$layer feed-forward neural network as defined in eq. (9), where $L \geq 2$ and $\sigma(x) = \max(x, \alpha x)$ for some $\alpha \in \mathbb{R}$. Let $\overline{\mathbf{W}}_{1:L}$ be defined as*

$$\overline{\mathbf{W}}_{1:L} := (\overline{\mathbf{W}}_1, \cdots, \overline{\mathbf{W}}_{L-1}, \overline{\mathbf{W}}_L) = (\mathbf{a}_1 \mathbf{b}_1^\top, \cdots, \mathbf{a}_{L-1} \mathbf{b}_{L-1}^\top, \overline{\mathbf{w}}^\top), \text{ where } \|\mathbf{a}_l\|_2 = \|\mathbf{b}_l\|_2, \text{ for all } l \in [L-1].$$

(i) *Suppose $\alpha \neq 1$ and the entries of $\{\mathbf{a}_l\}_{l=1}^{L-1}, \{\mathbf{b}_l\}_{l=2}^{L-1}$ are non-negative, then $\overline{\mathbf{W}}_{1:L}$ is a non-zero KKT point of eq. (10) if and only if the following holds: $\|\mathbf{a}_l\|_2^2 = 1/\sqrt{L}$, for all $l \in [L-1]$, $\|\overline{\mathbf{w}}\|_2^2 = 1/L$, $\mathbf{a}_l = \mathbf{b}_{l+1}$, for all $l \in [L-2]$, and $q\mathbf{a}_{L-1} = L^{1/4}\overline{\mathbf{w}}$, for some $q \in \{-1, 1\}$. Further, $\mathbf{b}_1$ is a non-zero*

*KKT point of*

$$\max_{\mathbf{u}} \sum_{i=1}^{n} y_i \sigma^{L-1}(\mathbf{x}_i^\top \mathbf{u}), \ such \ that \ \|\mathbf{u}\|_2^2 = 1/\sqrt{L}. \tag{11}$$

(ii) *Next, suppose $\alpha = 1$, then $\overline{\mathbf{W}}_{1:L}$ is a non-zero KKT point of eq. (10) if and only if the following holds: $\|\mathbf{a}_l\|_2^2 = 1/\sqrt{L}$, for all $l \in [L-1]$, $\|\overline{\mathbf{w}}\|_2^2 = 1/L$, $q_l \mathbf{a}_l = \mathbf{b}_{l+1}$, for all $l \in [L-2]$, and $q_{L-1}\mathbf{a}_{L-1} = L^{1/4}\overline{\mathbf{w}}$, where $q_l \in \{-1, 1\}$. Further, $\mathbf{b}_1$ is a non-zero KKT point of*

$$\max_{\mathbf{u}} \sum_{i=1}^{n} y_i \sigma^{L-1}(\mathbf{x}_i^\top \mathbf{u}), \ such \ that \ \|\mathbf{u}\|_2^2 = 1/\sqrt{L}. \tag{12}$$

In the above theorem, $\{\mathbf{a}_i, \mathbf{b}_i\}_{i=1}^{L-1}$ are vectors used to build the rank-one hidden weights. In both cases, we first provide the norm of these vectors. Next, for $\alpha \neq 1$ and assuming the entries of $\{\mathbf{a}_l\}_{l=1}^{L-1}, \{\mathbf{b}_l\}_{l=2}^{L-1}$ are non-negative, we present additional necessary and sufficient conditions, which can be divided in two parts. First is the *alignment*: $\mathbf{a}_l = \mathbf{b}_{l+1}$, for all $l \in [L-2]$, and $q\mathbf{a}_{L-1} = L^{1/4}\overline{\mathbf{w}}$, for some $q \in \{-1, 1\}$, which implies that the column-space of $\overline{\mathbf{W}}_l$ and row-space of $\overline{\mathbf{W}}_{l+1}$ must be aligned. The second part requires $\mathbf{b}_1$ to be a KKT point of eq. (11), a smaller optimization problem, where note that $\sigma^{L-1}(\cdot)$ denotes the function resulting from composing $\sigma(\cdot)$ with itself $(L-1)$ times. The results for $\alpha = 1$ are mostly similar, but do not require any non-negativity assumption.

## 5.2 Polynomial Leaky ReLU

We next provide necessary and sufficient conditions for activation function of the form $\sigma(x) = \max(x, \alpha x)^p$, for some $\alpha \in \mathbb{R}$, $p \geq 2$ and $p \in \mathbb{N}$. In the following theorem, for a given $p$, we define $\hat{p} := p^{L-1} + p^{L-2} + \cdots + 1$.

**Theorem 4.** *Let $\mathcal{H}$ be an $L-$layer feed-forward neural network as defined in eq. (9), where $L \geq 2$ and $\sigma(x) = \max(x, \alpha x)^p$ for some $\alpha \in \mathbb{R}$, $p \geq 2$ and $p \in \mathbb{N}$. Let $\overline{\mathbf{W}}_{1:L}$ be defined as*

$$\overline{\mathbf{W}}_{1:L} := (\overline{\mathbf{W}}_1, \cdots, \overline{\mathbf{W}}_{L-1}, \overline{\mathbf{W}}_L) = (\mathbf{a}_1\mathbf{b}_1^\top, \cdots, \mathbf{a}_{L-1}\mathbf{b}_{L-1}^\top, \overline{\mathbf{w}}^\top), \ where \ \|\mathbf{a}_l\|_2 = \|\mathbf{b}_l\|_2, \ for \ all \ l \in [L-1].$$

(i) *Suppose $\alpha \neq 1$ and the entries of $\{\mathbf{a}_l\}_{l=1}^{L-1}, \{\mathbf{b}_l\}_{l=2}^{L-1}$ are non-negative, then $\overline{\mathbf{W}}_{1:L}$ is a non-zero KKT point of eq. (10) if and only if the following holds: $\|\mathbf{a}_l\|_2^4 = p^{L-l}/\hat{p}$, for all $l \in [L-1]$, $\|\overline{\mathbf{w}}\|_2^2 = 1/\hat{p}$, $\mathbf{b}_l = \mathbf{a}_{l-1}/p^{1/4}$, for $2 \leq l \leq L-1$, and $\overline{\mathbf{w}} = q\mathbf{a}_{L-1}/(p\hat{p})^{1/4}$, for some $q \in \{-1, 1\}$. Further, for $1 \leq l \leq L-1$, each non-zero entry of $\mathbf{a}_l$ has identical values, and $\mathbf{b}_1$ is a non-zero KKT point of*

$$\max_{\mathbf{u}} \sum_{i=1}^{n} y_i \sigma^{L-1}(\mathbf{x}_i^\top \mathbf{u}), \ such \ that \ \|\mathbf{u}\|_2^2 = \sqrt{p^{L-1}/\hat{p}}. \tag{13}$$

(ii) *Suppose $\alpha = 1$, then $\overline{\mathbf{W}}_{1:L}$ is a non-zero KKT point of eq. (10) if and only if the following holds: $\|\mathbf{a}_l\|_2^4 = p^{L-l}/\hat{p}$, for all $l \in [L-1]$, $\|\overline{\mathbf{w}}\|_2^2 = 1/\hat{p}$, $\mathbf{b}_l = q_l \mathbf{a}_{l-1}/p^{1/4}, \overline{\mathbf{w}} = q_L \mathbf{a}_{L-1}/(p\hat{p})^{1/4}$, if $p$ is odd, and $\mathbf{b}_l = q_l |\mathbf{a}_{l-1}|/p^{1/4}, \overline{\mathbf{w}} = q_L |\mathbf{a}_{L-1}|/(p\hat{p})^{1/4}$, if $p$ is even, where $q_l \in \{-1, 1\}$, for all $2 \leq l \leq L$. Further, for $1 \leq l \leq L-1$, each non-zero entry of $\mathbf{a}_l$ has identical absolute values, and $\mathbf{b}_1$ is a non-zero KKT point of*

$$\max_{\mathbf{u}} \sum_{i=1}^{n} y_i \sigma^{L-1}(\mathbf{x}_i^\top \mathbf{u}), \ such \ that \ \|\mathbf{u}\|_2^2 = \sqrt{p^{L-1}/\hat{p}}. \tag{14}$$

Overall, the results are mostly analogous to Theorem 3, with one key difference: the non-zero entries of $\{\mathbf{a}_l\}_{l=1}^{L-1}$ must have identical (absolute) values. This difference is also evident in Figure 3, which depicts KKT points of 3-layer neural network with ReLU and squared-ReLU activation. For ReLU, in contrast with squared-ReLU, multiple rows and columns of the weights are non-zero. Additionally, the KKT point exhibits sparsity, with ReLU activation having multiple rows and columns with zero norm, while squared-ReLU activation has only one non-zero row and column. We observe this difference consistently in our experiments.

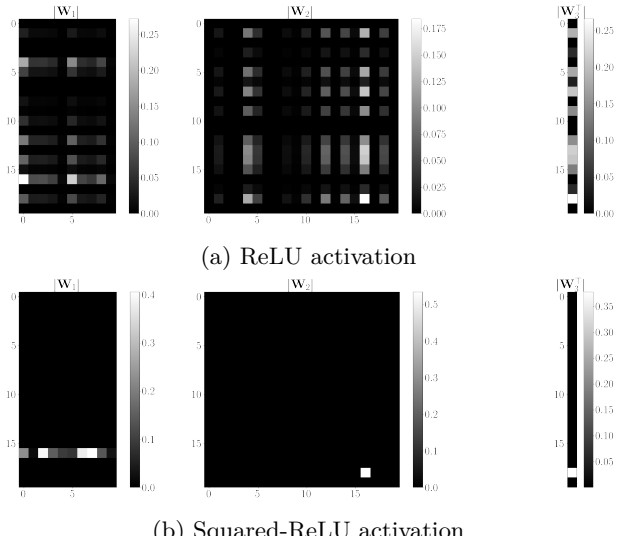

(a) ReLU activation

(b) Squared-ReLU activation

Figure 3: Absolute value of a KKT point $(\overline{\mathbf{W}}_1, \overline{\mathbf{W}}_2, \overline{\mathbf{W}}_3)$ of eq. (10) for ReLU activation $(\max(x, 0))$ and squared-ReLU activation $(\max(x, 0)^2)$, where $L = 3$, $n = 100$ and $\mathbf{x}_i \sim_{\text{i.i.d}} \mathcal{N}(0, \mathbf{I}_{10})$, $y_i \sim_{\text{i.i.d}} \mathcal{N}(0, 1)$. For ReLU, the non-zero rows/columns are scalar multiple of each other, implying the hidden weights have rank one. For squared-ReLU, it is clear that the hidden weights have rank one.

The results of Theorem 3 and 4 contain sufficient conditions which imply that KKT points of the constrained NCF with rank-one hidden weights always exists for arbitrary training data or depth of the neural network. The sufficient conditions also provide an approach to obtain a KKT point of the constrained NCF using KKT points of a smaller optimization problem. Additionally, they also contain matching necessary conditions, where, for $\alpha \neq 1$, we had to further assume that $\{\mathbf{a}_l\}_{l=1}^{L-1}, \{\mathbf{b}_l\}_{l=2}^{L-1}$ have non-negative entries. This implies that if, for $\alpha \neq 1$, there is a KKT point with rank-one hidden weights such that $\{\mathbf{a}_l\}_{l=1}^{L-1}, \{\mathbf{b}_l\}_{l=2}^{L-1}$ have non-negative entries, then it must be of the form described in the above theorems. Interestingly, in our experiments with random training data (Section 5.3), we found that the KKT points found via gradient-based method generally satisfy this non-negativity assumption, along with having rank-one hidden weights.

**Remark 1.** *In the above theorems, we focus on non-zero KKT points, because zero KKT points may not have a rich structure. For instance, using Lemma 11, it can be shown that if the network's output is always zero, the corresponding weights are zero KKT points. For ReLU activation, the output is zero if the input is non-positive. Thus, for an L-layer ReLU network, the output will be zero if any matrix in $(\mathbf{W}_2, \dots, \mathbf{W}_{L-1})$ has non-positive entries, making those weights zero KKT points.*

### 5.3 Numerical Experiments

In the above theorems, we have made two main assumptions on the KKT points of the NCF $(\overline{\mathbf{W}}_1, \cdots, \overline{\mathbf{W}}_L)$:

1. **Rank-one hidden weights.** $\overline{\mathbf{W}}_l = \mathbf{a}_l \mathbf{b}_l^\top$, where $\|\mathbf{a}_l\|_2 = \|\mathbf{b}_l\|_2$, for all $l \in [L-1]$.

2. **Non-negative weights.** If $\sigma(x) = \max(x, \alpha x)^p$ and $\alpha \neq 1$, then $\{\mathbf{a}_l\}_{l=1}^{L-1}, \{\mathbf{b}_l\}_{l=2}^{L-1}$ are non-negative.

To validate these assumptions, we conduct numerical experiments for two and three layer neural network with activation function $\sigma(x) = \max(\alpha x, x)^p$, where $p \in \{1, 2\}$ and $\alpha \in \{0, 0.1, 1\}$. For our experiments, we generate a random set of training data, and then obtain a KKT point of the corresponding constrained NCF by maximizing the NCF via gradient ascent with random initialization;[3] more details are provided in the

---

[3]Since gradient ascent can become unbounded, we use projected gradient ascent, i.e., after every gradient ascent step we normalize the weights to have unit norm. However, it also turns out that for homogeneous objectives, projected gradient ascent with fixed step-size is equivalent to gradient ascent with adaptive step-size, up to a scaling factor (see Lemma 19).

|       | $\alpha = 0$ | $\alpha = 0.1$ | $\alpha = 1$ |
|-------|--------------|----------------|--------------|
| $p = 1$ | $5.93 \cdot 10^{-9}$ | $1.79 \cdot 10^{-8}$ | $2.07 \cdot 10^{-8}$ |
| $p = 2$ | $1.76 \cdot 10^{-10}$ | $1.67 \cdot 10^{-10}$ | $6.86 \cdot 10^{-11}$ |

Three-layer

|       | $\alpha = 0$ | $\alpha = 0.1$ | $\alpha = 1$ |
|-------|--------------|----------------|--------------|
| $p = 1$ | $2.97 \cdot 10^{-12}$ | $1.22 \cdot 10^{-11}$ | $1.9 \cdot 10^{-10}$ |
| $p = 2$ | $1.48 \cdot 10^{-11}$ | $1.12 \cdot 10^{-11}$ | $7.93 \cdot 10^{-11}$ |

Two-layer

Table 1: The left table contains the maximum value of $\kappa(\overline{\mathbf{W}}_1, \overline{\mathbf{W}}_2)$ across 30 different random instances for a three-layer neural network with activation function $\max(x, \alpha x)^p$, where $(\overline{\mathbf{W}}_1, \overline{\mathbf{W}}_2, \overline{\mathbf{W}}_3)$ is the KKT point of the constrained NCF obtained via gradient ascent. The right table is similar but for two-layer neural network.

|       | $\alpha = 0$ | $\alpha = 0.1$ |
|-------|--------------|----------------|
| $p = 1$ | $4.44 \cdot 10^{-16}$ | $3.28 \cdot 10^{-16}$ |
| $p = 2$ | $5.16 \cdot 10^{-6}$ | $5 \cdot 10^{-6}$ |

Three-layer

|       | $\alpha = 0$ | $\alpha = 0.1$ |
|-------|--------------|----------------|
| $p = 1$ | $1.82 \cdot 10^{-7}$ | $2.77 \cdot 10^{-7}$ |
| $p = 2$ | $3.95 \cdot 10^{-7}$ | $3.58 \cdot 10^{-7}$ |

Two-layer

Table 2: Suppose $(\overline{\mathbf{W}}_1, \cdots, \overline{\mathbf{W}}_L)$ is the KKT point of the constrained NCF obtained via gradient ascent for an $L-$layer neural network, and let $\mathbf{a}_l \mathbf{b}_l^\top$ be the rank-one approximation of $\mathbf{W}_l$ such that $\|\mathbf{a}_l\|_2 = \|\mathbf{b}_l\|_2$, for all $l \in [L-1]$. The left (right) table contains the maximum value of $\rho(\mathbf{a}_1, \mathbf{a}_2 \mathbf{b}_2^\top, \cdots \mathbf{a}_{L-1} \mathbf{b}_{L-1}^\top)$ across 30 different random instances for a three-layer (two-layer) neural network with activation function $\max(x, \alpha x)^p$. Note that if $\mathbf{a}_l \mathbf{b}_l^\top$ is non-negative, then we can choose $\mathbf{a}_l$ and $\mathbf{b}_l$ such that they are also non-negative. Hence, showing $\mathbf{a}_1, \mathbf{a}_2 \mathbf{b}_2^\top, \cdots \mathbf{a}_{L-1} \mathbf{b}_{L-1}^\top$ are non-negative is equivalent to showing $\{\mathbf{a}_l\}_{l=1}^{L-1}, \{\mathbf{b}_l\}_{l=2}^{L-1}$ are non-negative.

Appendix C.4. The results presented below are for 30 such random instances. Also, as discussed earlier, the gradient flow of the training loss converges to the same KKT point as the gradient flow of the NCF. Thus, using gradient ascent to find the KKT point of the constrained NCF also provides insights into the KKT point towards which gradient flow of the training loss converges in the early stages of training.

**Rank-one assumption:** For any set of matrices $\{\mathbf{Z}_1, \cdots, \mathbf{Z}_p\}$, we use

$$\kappa(\mathbf{Z}_1, \cdots, \mathbf{Z}_p) =: \max_{1 \leq i \leq p} \left( 1 - \frac{\|\mathbf{Z}_i\|_2}{\|\mathbf{Z}_i\|_F} \right)$$

to measure how close are each of $\{\mathbf{Z}_1, \cdots, \mathbf{Z}_p\}$ to being rank-one. If $\kappa(\mathbf{Z}_1, \cdots, \mathbf{Z}_p) = 0$, then each of $\{\mathbf{Z}_1, \cdots, \mathbf{Z}_p\}$ are rank one, and thus, for smaller $\kappa(\mathbf{Z}_1, \cdots, \mathbf{Z}_p)$, they are approximately rank one. Also, $\kappa(\cdot)$ is related to the notion of numerical rank (Rudelson & Vershynin, 2007), a stable relaxation of the rank. Now, from the values in Table 1, we observe that for both two-layer and three-layer neural network, and with different activation functions, each of the hidden weights in the KKT point obtained via gradient ascent are approximately of rank one in *all the instances considered* (since we are reporting the maximum value of $\kappa(\cdot)$ across all the random instances.)

**Non-negativity assumption:** For any set $\{\mathbf{Z}_1, \cdots, \mathbf{Z}_p\}$ containing matrices and/or vectors, we use

$$\rho(\mathbf{Z}_1, \cdots, \mathbf{Z}_p) =: \max_{1 \leq i \leq p} \left( \frac{\|\max(0, -\mathbf{Z}_i)\|_F}{\|\mathbf{Z}_i\|_F} \right)$$

to measure how close are each of $\{\mathbf{Z}_1, \cdots, \mathbf{Z}_p\}$ to being non-negative. Note that, if $\mathbf{Z}_i$ is a vector, then $\|\mathbf{Z}_i\|_F$ denotes its $\ell_2$-norm. If $\rho(\mathbf{Z}_1, \cdots, \mathbf{Z}_p) = 0$, then each of $\{\mathbf{Z}_1, \cdots, \mathbf{Z}_p\}$ are non-negative, and thus, for smaller $\rho(\mathbf{Z}_1, \cdots, \mathbf{Z}_p)$, they are approximately non-negative. From the values in Table 2, we observe that for both two-layer and three-layer neural network, and with different activation functions, the appropriate set of weights in the KKT point obtained via gradient ascent are approximately non-negative in *all the instances considered*. Note that, we did not consider linear or monomial activation ($\sigma(x) = x^p$) since in those cases we did not make any non-negativity assumption.

We conclude this section by again noting that under the rank-one and non-negativity assumption considered here, the sufficient conditions of Theorem 3 and 4 are also necessary conditions, and thus, the KKT points must be of the form described in those results. Since the rank-one and non-negativity assumption is satisfied for all the instances considered, it implies the KKT points are of form described in Theorem 3 and 4 for all the instances considered. Also, note that the constrained NCF is a non-convex problem and could have KKT points that do not satisfy these assumptions, but gradient ascent seems to avoid them. Understanding this phenomenon is an interesting future direction, which would require analyzing the dynamics of gradient ascent.

## 5.4 Proof Overview

We now provide a brief overview of the proof technique for the above theorems; the details can be found in Appendix C. We start with the following lemma, which is important for our proof.

**Lemma 5.** *Let $\mathcal{H}$ be an $L$-layer feed-forward neural network whose output is as defined in eq. (9), where $L \geq 2$. Suppose $(\overline{\mathbf{W}}_1, \cdots, \overline{\mathbf{W}}_L)$ is a non-zero KKT points of the constrained NCF in eq. (10). If $\sigma(x) = \max(\alpha x, x)^p$, for some $p \in \mathbb{N}$ and $\alpha \in \mathbb{R}$, then*

$$diag\left(\overline{\mathbf{W}}_l \overline{\mathbf{W}}_l^\top\right) = p \cdot diag\left(\overline{\mathbf{W}}_{l+1}^\top \overline{\mathbf{W}}_{l+1}\right), \text{ for all } l \in [L-1], \tag{15}$$

*and, if $\sigma(x) = x$, then*

$$\overline{\mathbf{W}}_l \overline{\mathbf{W}}_l^\top = \overline{\mathbf{W}}_{l+1}^\top \overline{\mathbf{W}}_{l+1}, \text{ for all } l \in [L-1].$$

*Furthermore, if $(\overline{\mathbf{W}}_1, \cdots, \overline{\mathbf{W}}_{L-1}, \overline{\mathbf{W}}_L) = (\mathbf{a}_1 \mathbf{b}_1^\top, \cdots, \mathbf{a}_{L-1} \mathbf{b}_{L-1}^\top, \overline{\mathbf{w}}^\top)$, where $\|\mathbf{a}_l\|_2 = \|\mathbf{b}_l\|_2$, for all $l \in [L-1]$, then $|\mathbf{a}_l| = p^{1/4}|\mathbf{b}_{l+1}|$, for all $l \in [L-2]$, $|\mathbf{a}_{L-1}| = (p\hat{p})^{1/4}|\overline{\mathbf{w}}|$, $\|\mathbf{a}_l\|_2^4 = p^{L-l}/\hat{p}$, for all $l \in [L-1]$, and $\|\overline{\mathbf{w}}\|_2^2 = 1/\hat{p}$, where $\hat{p} = p^{L-1} + p^{L-2} + \cdots + 1$.*

The above lemma implies $|\mathbf{a}_l|$ is parallel to $|\mathbf{b}_{l+1}|$. This is the first step towards establishing the alignment conditions stated in the above theorems. Now, if $\mathbf{a}_l$ and $\mathbf{b}_{l+1}$ are non-negative, which is true in certain cases, we get $\mathbf{a}_l$ is parallel to $\mathbf{b}_{l+1}$. We similarly use additional conditions to get alignment results for other cases.

We next describe how to show $\mathbf{b}_1$ is a KKT point of the smaller optimization problem. For this, we focus on deep ReLU networks and assume that the alignment conditions are true, the proof of other activation functions follow a similar approach. From the KKT conditions, we know $\overline{\mathbf{W}}_1$ must satisfy

$$\mathbf{0} \in \lambda \overline{\mathbf{W}}_1 + \partial_{\mathbf{W}_1}\left(\mathbf{y}^\top \mathcal{H}(\mathbf{X}; \overline{\mathbf{W}}_1, \cdots, \overline{\mathbf{W}}_L)\right) = \lambda \mathbf{a}_1 \mathbf{b}_1^\top + \partial_{\mathbf{W}_1}\left(\mathbf{y}^\top \mathcal{H}(\mathbf{X}; \overline{\mathbf{W}}_1, \cdots, \overline{\mathbf{W}}_L)\right). \tag{16}$$

Since $\{\mathbf{a}_l\}_{l=1}^{L-1}, \{\mathbf{b}_l\}_{l=2}^{L-1}$ have non-negative entries, ReLU is positively homogeneous, $\|\mathbf{a}_l\|_2^2 = 1/\sqrt{L}$, and $\mathbf{a}_l = \mathbf{b}_{l+1}$, for all $l \in [L-2]$, $q\mathbf{a}_{L-1} = L^{1/4}\overline{\mathbf{w}}$, we can write

$$\begin{aligned}
\mathcal{H}(\mathbf{x}; \mathbf{W}_1, \overline{\mathbf{W}}_2 \cdots, \overline{\mathbf{W}}_L) &= \overline{\mathbf{W}}_L \sigma\left(\overline{\mathbf{W}}_{L-1} \cdots \overline{\mathbf{W}}_2 \sigma\left(\mathbf{W}_1 \mathbf{x}\right) \cdots\right) \\
&= q\mathbf{a}_{L-1}^\top \sigma(\mathbf{a}_{L-1} \mathbf{a}_{L-2}^\top \sigma(\mathbf{a}_{L-2} \mathbf{a}_{L-3}^\top \cdots \sigma(\mathbf{a}_2 \mathbf{a}_1^\top \sigma(\mathbf{W}_1 \mathbf{x})) \cdots))/L^{1/4} \\
&= q\|\mathbf{a}_{L-1}\|_2^2 \|\mathbf{a}_{L-2}\|_2^2 \cdots \|\mathbf{a}_2\|_2^2 \sigma^{L-2}(\mathbf{a}_1^\top \sigma(\mathbf{W}_1 \mathbf{x}))/L^{1/4} \\
&= q\left(1/\sqrt{L}\right)^{L-2} \sigma^{L-2}(\mathbf{a}_1^\top \sigma(\mathbf{W}_1 \mathbf{x}))/L^{1/4}.
\end{aligned}$$

We use the above equality to simplify $\partial_{\mathbf{W}_1}\left(\mathbf{y}^\top \mathcal{H}(\mathbf{X}; \overline{\mathbf{W}}_1, \cdots, \overline{\mathbf{W}}_L)\right)$ and use it in eq. (16) to prove $\mathbf{b}_1$ is a KKT point of the smaller optimization problem. Also, note that since ReLU activation is non-differentiable, we used the Clarke sub-differential in the KKT conditions. A challenging aspect of using Clarke sub-differentials is that the chain rule does not hold with equality, instead it yields a set and the true sub-differential belongs to that set (see Lemma 15). Consequently, it is not feasible to simplify $\partial_{\mathbf{W}_1}\left(\mathbf{y}^\top \mathcal{H}(\mathbf{X}; \overline{\mathbf{W}}_1, \cdots, \overline{\mathbf{W}}_L)\right)$ using the above equality by simply applying the chain rule. We follow a more direct approach to simplify the Clarke sub-differential, which uses positive homogeneity of the activation function and non-negativity of $\mathbf{a}_1$.

## 6 Conclusion and Future Directions

In this paper, we studied the gradient flow dynamics resulting from training deep homogeneous neural networks with small initializations, and established the approximate directional convergence of the weights in the early stages of training. However, these results are not applicable for ReLU networks, so one important future direction would be to rigorously establish similar directional convergence for ReLU networks.

We also derived necessary and sufficient conditions for the KKT points of the constrained NCF that have rank-one hidden weights, for feed-forward neural networks with Leaky ReLU and polynomial Leaky ReLU activation functions. The constrained NCF is a non-convex problem and all KKT points may not be low-rank; however, our experiments suggest that gradient ascent tends to converge towards low-rank KKT points. Understanding this phenomenon is also an important area for future exploration, which will also help us understand why gradient descent on training loss converges towards low-rank KKT points of the NCF in the early stages of training. We hope our characterization of rank-one KKT points will aid in this exploration.

### Acknowledgments

The authors graciously acknowledge gift funding from InterDigital, which partially supported this work.

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

## A  Key lemmata

In this section, we state some key lemmata that are important for our results.

### A.1  Kurdyka-Lojasiewicz inequality for definable functions

The Kurdyka-Lojasiewicz Inequality is useful for demonstrating convergence of bounded gradient flow trajectories by establishing the existence of a desingularizing function, which is formally defined as follows.

**Definition A.1.** *A function* $\Psi : [0, \nu) \to \mathbb{R}$ *is called a desingularizing function when* $\Psi$ *is continuous on* $[0, \nu)$ *with* $\Psi(0) = 0$*, and it is continuously differentiable on* $(0, \nu)$ *with* $\Psi' > 0$*.*

In establishing the directional convergence of gradient flow trajectories of the neural correlation function, where the trajectories may not be bounded, we will utilize the following unbounded version of the Kurdyka-Lojasiewicz inequality.

**Lemma 6.** *(Ji & Telgarsky, 2020, Lemma 3.6) Let* $f$ *be a locally Lipschitz function, definable under an o-minimal structure, and having an open domain* $\mathbf{D} \subset \{\mathbf{x} | \|\mathbf{x}\|_2 > 1\}$*. For any* $c, \eta > 0$*, there exists a* $\nu > 0$ *and a definable desingularizing function* $\Psi$ *on* $[0, \nu)$ *such that*

$$\Psi'(f(\mathbf{x})) \|\mathbf{x}\|_2 \|\nabla f(\mathbf{x})\|_2 \geq 1, \;\; if \; f(\mathbf{x}) \in (0, \nu) \; and \; \|\nabla_\perp f(\mathbf{x})\|_2 \geq c \|\mathbf{x}\|_2^\eta \|\nabla_r f(\mathbf{x})\|_2,$$

*where* $\nabla_r f(\mathbf{x}) = \left\langle \nabla f(\mathbf{x}), \frac{\mathbf{x}}{\|\mathbf{x}\|_2} \right\rangle \frac{\mathbf{x}}{\|\mathbf{x}\|_2}$ *and* $\nabla_\perp f(\mathbf{x}) = \nabla f(\mathbf{x}) - \nabla_r f(\mathbf{x})$*.*

### A.2  Euler's Theorem for homogeneous functions

The next lemma states two important properties of homogeneous functions.

**Lemma 7.** *((Lyu & Li, 2020, Theorem B.2), (Ji & Telgarsky, 2020, Lemma C.1)) Let* $F : \mathbb{R}^k \to \mathbb{R}$ *be locally Lipschitz, differentiable, and* $L-$*positively homogeneous for some* $L > 0$*. Then,*

1. *For any* $\mathbf{w} \in \mathbb{R}^k$ *and* $c \geq 0$,
$$\nabla F(c\mathbf{w}) = c^{L-1} \nabla F(\mathbf{w}).$$

2. *For any* $\mathbf{w} \in \mathbb{R}^k$,
$$\mathbf{w}^\top \nabla F(\mathbf{w}) = L F(\mathbf{w}).$$

### A.3  Gronwall's inequality

In our proofs, we will frequently use Gronwall's inequality.

**Lemma 8.** *Let* $\alpha, \beta, u$ *be real-valued functions defined on an interval* $[a, b]$*, where* $\beta$ *and* $u$ *are continuous and* $\min(\alpha, 0)$ *is integrable on every closed and bounded sub-interval of* $[a, b]$*.*

- *If* $\beta$ *is non-negative and if* $u$ *satisfies the integral inequality*

$$u(t) \leq \alpha(t) + \int_a^t \beta(s) u(s) ds, \forall t \in [a, b],$$

  *then*

$$u(t) \leq \alpha(t) + \int_0^t \alpha(s) \beta(s) \exp\left( \int_s^t \beta(r) dr \right) ds, \forall t \in [a, b].$$

- *If, in addition, the function* $\alpha$ *is non-decreasing, then*

$$u(t) \leq \alpha(t) \exp\left( \int_a^t \beta(s) ds \right), \forall t \in [a, b].$$

We will mainly be using the second part of the above lemma.

# B    Proofs omitted from Section 4.1

## B.1    Proof of Lemma 2

Recall, for a given vector $\mathbf{z}$ and a neural network $\mathcal{H}$, the NCF is defined as

$$\mathcal{N}_{\mathbf{z},\mathcal{H}}(\mathbf{u}) = \mathbf{z}^\top \mathcal{H}(\mathbf{X}; \mathbf{u}). \tag{17}$$

In this section, for the sake of conciseness, we will use $\mathcal{N}(\mathbf{u})$ in place of $\mathcal{N}_{\mathbf{z},\mathcal{H}}(\mathbf{u})$. Further, $\mathbf{u}(t)$ satisfies for all $t \geq 0$

$$\frac{d\mathbf{u}}{dt} = \nabla\mathcal{N}(\mathbf{u}) = \mathcal{J}(\mathbf{X}; \mathbf{u})^\top \mathbf{z}, \mathbf{u}(0) = \mathbf{u}_0, \tag{18}$$

where $\mathcal{J}(\mathbf{X}; \mathbf{u}) : \mathbb{R}^k \to \mathbb{R}^n$ is the Jacobian of the function $\mathcal{H}(\mathbf{X}; \mathbf{u})$. Let $\mathcal{S}^{k-1}$ denote the unit sphere on $\mathbb{R}^k$, and define

$$\beta = \sup\{\|\mathcal{H}(\mathbf{X}; \mathbf{w})\|_2 : \mathbf{w} \in \mathcal{S}^{k-1}\}, \text{ and } \alpha = \sup\{\|\mathcal{J}(\mathbf{X}; \mathbf{w})\|_2 : \mathbf{w} \in \mathcal{S}^{k-1}\}. \tag{19}$$

We will start by establishing some auxiliary lemmata. The following lemma follows from $L$-homogeneity of $\mathcal{N}(\mathbf{u})$ and Lemma 7.

**Lemma 9.** *For any $\mathbf{u} \in \mathbb{R}^k$ and $c \geq 0$, $\nabla\mathcal{N}(\mathbf{u})^\top\mathbf{u} = L\mathcal{N}(\mathbf{u})$ and $\nabla\mathcal{N}(c\mathbf{u}) = c^{L-1}\nabla\mathcal{N}(\mathbf{u})$, and $\mathcal{J}(\mathbf{X}; \mathbf{u})\mathbf{u} = L\mathcal{H}(\mathbf{X}; \mathbf{u})$ and $\mathcal{J}(\mathbf{X}; c\mathbf{u}) = c^{L-1}\mathcal{J}(\mathbf{X}; \mathbf{u})$.*

We next define $\tilde{\mathcal{N}}(\mathbf{u})$, which plays an integral part in the proof, and its gradient.

**Lemma 10.** *For any nonzero $\mathbf{u} \in \mathbb{R}^k$ we define $\tilde{\mathcal{N}}(\mathbf{u}) = \mathcal{N}(\mathbf{u})/\|\mathbf{u}\|_2^L$, then,*

$$\nabla\tilde{\mathcal{N}}(\mathbf{u}) = \frac{\nabla\mathcal{N}(\mathbf{u})}{\|\mathbf{u}\|_2^L} - \frac{L\mathcal{N}(\mathbf{u})\mathbf{u}}{\|\mathbf{u}\|_2^{L+2}} = \left(\mathbf{I} - \frac{\mathbf{u}\mathbf{u}^\top}{\|\mathbf{u}\|_2^2}\right)\frac{\nabla\mathcal{N}(\mathbf{u})}{\|\mathbf{u}\|_2^L}.$$

*Proof.* The first equality follows by differentiating $\tilde{\mathcal{N}}(\mathbf{u})$ with respect to $\mathbf{u}$; using Lemma 9 we get the second equality. $\square$

We next describe the conditions for first-order KKT point of the NCF.

**Lemma 11.** *If a vector $\mathbf{u}_* \in \mathbb{R}^{k\times1}$ is a first-order KKT point of*

$$\max_{\|\mathbf{u}\|_2^2=1} \mathcal{N}(\mathbf{u}) = \mathbf{z}^\top\mathcal{H}(\mathbf{X}; \mathbf{u}), \tag{20}$$

*then*

$$\nabla\mathcal{N}(\mathbf{u}_*) = \mathcal{J}(\mathbf{X}; \mathbf{u}_*)^\top\mathbf{z} = \lambda^*\mathbf{u}_*, \|\mathbf{u}_*\|_2^2 = 1, \tag{21}$$

*where $\lambda^* \in \mathbb{R}$ is the Lagrange multiplier. Also, $L\mathcal{N}(\mathbf{u}_*) = \lambda^*$, which implies that $\lambda^* \geq 0$ for a non-negative KKT point.*

*Proof.* For eq. (20), the Lagrangian is

$$L(\mathbf{u}, \lambda) = \mathcal{N}(\mathbf{u}) + \lambda(\|\mathbf{u}\|_2^2 - 1).$$

Hence, if $\mathbf{u}_*$ is a first-order KKT point then $\|\mathbf{u}_*\|_2^2 = 1$, and for some $\overline{\lambda}$ we have

$$\nabla\mathcal{N}(\mathbf{u}_*) + 2\overline{\lambda}\mathbf{u}_* = \mathbf{0},$$

which implies

$$\mathcal{J}(\mathbf{X}; \mathbf{u}_*)^\top\mathbf{z} + 2\overline{\lambda}\mathbf{u}_* = \mathbf{0}.$$

We get eq. (21) by choosing $\lambda^* = -2\overline{\lambda}$. Further, by Lemma 9, we have

$$\lambda^* = \lambda^*\|\mathbf{u}_*\|_2^2 = {\mathbf{u}_*}^\top\nabla\mathcal{N}(\mathbf{u}_*) = L\mathcal{N}(\mathbf{u}_*).$$

$\square$

**Lemma 12.** *For any $t \geq 0$, we have*

$$\frac{d\mathcal{N}(\mathbf{u})}{dt} = \|\nabla\mathcal{N}(\mathbf{u}(t))\|_2^2 = \|\dot{\mathbf{u}}(t)\|_2^2. \tag{22}$$

*Proof.* Using $\dot{\mathbf{u}} = \nabla\mathcal{N}(\mathbf{u}(t))$, we have

$$\frac{d\mathcal{N}(\mathbf{u})}{dt} = \nabla\mathcal{N}(\mathbf{u}(t))^\top \dot{\mathbf{u}} = \|\nabla\mathcal{N}(\mathbf{u}(t))\|_2^2 = \|\dot{\mathbf{u}}(t)\|_2^2.$$

$\square$

**Lemma 13.** *For any $t_0 \geq 0$, if $\mathcal{N}(\mathbf{u}(t_0)) > 0$, then, for all $t \geq t_0$, we have $\mathcal{N}(\mathbf{u}(t)) \geq \mathcal{N}(\mathbf{u}(t_0)) > 0$ and $\|\mathbf{u}(t)\|_2 \geq \|\mathbf{u}(t_0)\|_2$.*

*Proof.* Using Lemma 12, we have

$$\mathcal{N}(\mathbf{u}(t)) - \mathcal{N}(\mathbf{u}(t_0)) = \int_{t_0}^{t} \|\dot{\mathbf{u}}(s)\|_2^2 ds.$$

Therefore, for $t \geq t_0$, $\mathcal{N}(\mathbf{u}(t)) \geq \mathcal{N}(\mathbf{u}(t_0)) > 0$. The second claim is true since for all $t \geq 0$

$$\frac{d\|\mathbf{u}\|_2^2}{dt} = 2\mathbf{u}^\top \dot{\mathbf{u}} = 2\mathbf{u}^\top \nabla\mathcal{N}(\mathbf{u}) = 2L\mathcal{N}(\mathbf{u}),$$

which implies

$$\|\mathbf{u}(t)\|_2^2 - \|\mathbf{u}(t_0)\|_2^2 = 2L \int_{t_0}^{t} \mathcal{N}(\mathbf{u}(s)) ds \geq 0.$$

$\square$

We now turn to proving Lemma 2.

*Proof of Lemma 2:* We begin by showing that either the limit of $\mathbf{u}(t)/\|\mathbf{u}(t)\|_2$ exists or $\mathbf{u}(t)$ converges to $\mathbf{0}$. For this, we consider two cases.

**Case 1:** $\mathcal{N}(\mathbf{u}(0)) > 0$.
In this case, we first show that $\mathbf{u}(t)$ becomes unbounded at some finite time. Suppose for the sake of contradiction $\|\mathbf{u}(t)\|_2 < \infty$ for all finite $t \geq 0$.

Let $\mathcal{N}(\mathbf{u}(0)) = \gamma > 0$, thus $\|\mathbf{u}(0)\|_2 > 0$. Using Lemma 13, for all $t \geq 0$,

$$\|\mathbf{u}(t)\|_2 \geq \|\mathbf{u}(0)\|_2 > 0.$$

Hence, $\tilde{\mathcal{N}}(\mathbf{u}(t))$ (which we recall is equal to $\mathcal{N}(\mathbf{u}(t))/\|\mathbf{u}(t)\|_2^L$) is defined for all $t \geq 0$. Now, by Lemma 12, for all $t \geq 0$, we have

$$\frac{d\mathcal{N}(\mathbf{u})}{dt} = \|\dot{\mathbf{u}}\|_2^2, \text{ and } \dot{\mathbf{u}} = \nabla\mathcal{N}(\mathbf{u}).$$

Therefore, using Lemma 10, for all $t \geq 0$,

$$\frac{d\tilde{\mathcal{N}}(\mathbf{u})}{dt} = \dot{\mathbf{u}}^\top \nabla\tilde{\mathcal{N}}(\mathbf{u}) = \frac{\nabla\mathcal{N}(\mathbf{u})^\top}{\|\mathbf{u}\|_2^L} \left(\mathbf{I} - \frac{\mathbf{u}\mathbf{u}^T}{\|\mathbf{u}\|_2^2}\right) \nabla\mathcal{N}(\mathbf{u}) \geq 0. \tag{23}$$

For all $t_2 \geq t_1 \geq 0$, integrating the above equality on both sides from $t_1$ to $t_2$ we get

$$\tilde{\mathcal{N}}(\mathbf{u}(t_2)) - \tilde{\mathcal{N}}(\mathbf{u}(t_1)) = \int_{t_1}^{t_2} \frac{\nabla\mathcal{N}(\mathbf{u})^\top}{\|\mathbf{u}\|_2^L} \left(\mathbf{I} - \frac{\mathbf{u}\mathbf{u}^T}{\|\mathbf{u}\|_2^2}\right) \nabla\mathcal{N}(\mathbf{u}) dt \geq 0. \tag{24}$$

Thus, $\tilde{\mathcal{N}}(\mathbf{u}(t))$ is an increasing function. Now if we define $\tilde{\mathcal{N}}(\mathbf{u}(0)) = \tilde{\gamma}$, then for any $t \geq 0$ we have

$$\tilde{\mathcal{N}}(\mathbf{u}(t)) \geq \tilde{\gamma} > 0,$$

which implies

$$\mathcal{N}(\mathbf{u}(t)) \geq \tilde{\gamma} \|\mathbf{u}(t)\|_2^L, \forall t \geq 0.$$

From the above inequality, we have

$$\frac{1}{2}\frac{d\|\mathbf{u}\|_2^2}{dt} = \mathbf{u}^\top \dot{\mathbf{u}} = L\mathcal{N}(\mathbf{u}(t)) \geq L\tilde{\gamma}\|\mathbf{u}(t)\|_2^L, \forall t \geq 0. \tag{25}$$

Taking $\|\mathbf{u}(t)\|_2^L$ to the LHS and integrating both sides from 0 to $t$, we get

$$\frac{1}{L/2 - 1}\left(\frac{1}{\|\mathbf{u}(0)\|_2^{L-2}} - \frac{1}{\|\mathbf{u}(t)\|_2^{L-2}}\right) \geq 2L\tilde{\gamma}t, \forall t \geq 0.$$

Simplifying the above equation gives us

$$\|\mathbf{u}(t)\|_2^{L-2} \geq \frac{\|\mathbf{u}(0)\|_2^{L-2}}{1 - t\tilde{\gamma}L(L-2)\|\mathbf{u}(0)\|_2^{L-2}}, \forall t \geq 0.$$

The above inequality indicates that $\|\mathbf{u}(t)\|_2$ is not bounded for all finite $t \geq 0$, which leads to a contradiction.

Henceforth, we choose $T^*$ to denote the time when $\mathbf{u}(t)$ becomes infinity. Formally, we assume that

$$\lim_{t \to T^*}\|\mathbf{u}(t)\|_2 = \infty, \text{ and } \|\mathbf{u}(t)\|_2 < \infty, \text{ for all } t \in [0, T^*).$$

We now proceed to prove that $\lim_{t \to T^*}\mathbf{u}(t)/\|\mathbf{u}(t)\|_2$ exists by establishing that the length of the curve swept by $\mathbf{u}(t)/\|\mathbf{u}(t)\|_2$, given by

$$\int_0^{T^*}\left\|\frac{d}{dt}\left(\frac{\mathbf{u}}{\|\mathbf{u}\|_2}\right)\right\|_2 dt,$$

is of finite length. This proof technique is inspired from Ji & Telgarsky (2020), and a similar technique was also employed in Kumar & Haupt (2024).

From eq. (23) and eq. (24), we know $\tilde{\mathcal{N}}(\mathbf{u}(t))$ is an increasing function in $[0, T^*)$. Further, since $\mathcal{N}(\mathbf{u}) = \mathbf{z}^\top \mathcal{H}(\mathbf{X}; \mathbf{u}) \leq \beta\|\mathbf{z}\|_2\|\mathbf{u}\|_2^L$, where the inequality follows from eq. (19), we have that $\tilde{\mathcal{N}}(\mathbf{u})$ is bounded from above. Thus, using monotone convergence theorem, we have $\lim_{t \to T^*}\tilde{\mathcal{N}}(\mathbf{u}(t))$ exists.

Now, for all $t \in [0, T^*)$, we know

$$\frac{d}{dt}\left(\frac{\mathbf{u}}{\|\mathbf{u}\|_2}\right) = \left(\mathbf{I} - \frac{\mathbf{u}\mathbf{u}^T}{\|\mathbf{u}\|_2^2}\right)\frac{\dot{\mathbf{u}}}{\|\mathbf{u}\|_2} = \left(\mathbf{I} - \frac{\mathbf{u}\mathbf{u}^T}{\|\mathbf{u}\|_2^2}\right)\frac{\nabla\mathcal{N}(\mathbf{u})}{\|\mathbf{u}\|_2}. \tag{26}$$

Therefore,

$$\left\|\frac{d}{dt}\left(\frac{\mathbf{u}}{\|\mathbf{u}\|_2}\right)\right\|_2 = \left\|\left(\mathbf{I} - \frac{\mathbf{u}\mathbf{u}^T}{\|\mathbf{u}\|_2^2}\right)\nabla\mathcal{N}(\mathbf{u})\right\|_2\frac{1}{\|\mathbf{u}\|_2}. \tag{27}$$

Now, suppose $\lim_{t \to T^*}\tilde{\mathcal{N}}(\mathbf{u}(t)) = \overline{f}$. If $\tilde{\mathcal{N}}(\mathbf{u}(t))$ converges to $\overline{f}$ before $T^*$, i.e., $\tilde{\mathcal{N}}(\mathbf{u}(T)) = \overline{f}$ for some $T < T^*$. Then, for all $t \in [T, T^*)$,

$$\frac{d\tilde{\mathcal{N}}(\mathbf{u})}{dt} = 0,$$

which implies

$$\left\|\left(\mathbf{I} - \frac{\mathbf{u}(t)\mathbf{u}(t)^T}{\|\mathbf{u}(t)\|_2^2}\right)\nabla\mathcal{N}(\mathbf{u}(t))\right\|_2 = 0, \forall t \in [T, T^*).$$

Hence, from eq. (27), we have

$$\frac{d}{dt}\left(\frac{\mathbf{u}}{\|\mathbf{u}\|_2}\right) = \mathbf{0}, \forall t \in [T, T^*),$$

which implies $\lim_{t \to T^*} \frac{\mathbf{u}(t)}{\|\mathbf{u}(t)\|_2}$ exists and is equal to $\frac{\mathbf{u}(T)}{\|\mathbf{u}(T)\|_2}$.

Thus, we may assume $\overline{f} - \tilde{\mathcal{N}}(\mathbf{u}(t)) > 0$, for all $t \in [0, T^*)$. We define $g(\mathbf{u}) = \overline{f} - \tilde{\mathcal{N}}(\mathbf{u})$. Then, since

$$\|\nabla_r \tilde{\mathcal{N}}(\mathbf{u})\|_2 = 0,$$

we have that

$$\|\nabla_\perp g(\mathbf{u})\|_2 \geq 0 = \|\mathbf{u}\|_2 \|\overline{\nabla}_r g(\mathbf{u})\|_2.$$

Hence, from Lemma 6, there exists a $\nu > 0$ and a desingularizing function $\Psi(.)$ defined on $[0, \nu)$ such that if $\|\mathbf{u}\|_2 > 1$ and $g(\mathbf{u}) < \nu$, then

$$1 \leq \Psi'(g(\mathbf{u}))\|\mathbf{u}\|_2\|\nabla g(\mathbf{u})\|_2 = \Psi'(\overline{f} - \tilde{\mathcal{N}}(\mathbf{u}))\|\mathbf{u}\|_2\|\nabla\tilde{\mathcal{N}}(\mathbf{u})\|_2. \tag{28}$$

Since $\lim_{t \to T^*} \tilde{\mathcal{N}}(\mathbf{u}(t)) = \overline{f}$, and $\lim_{t \to T^*} \|\mathbf{u}(t)\|_2 = \infty$, we may choose $T$ large enough such that $\|\mathbf{u}(t)\|_2 > 1$, and $g(\mathbf{u}(t)) < \nu$, for all $t \in [T, T^*)$. Hence, for all $t \in [T, T^*)$, we have

$$\begin{aligned}
\frac{d\tilde{\mathcal{N}}(\mathbf{u})}{dt} &= \frac{\nabla\mathcal{N}(\mathbf{u})^\top}{\|\mathbf{u}\|_2^L}\left(\mathbf{I} - \frac{\mathbf{u}\mathbf{u}^T}{\|\mathbf{u}\|_2^2}\right)\nabla\mathcal{N}(\mathbf{u}) \\
&= \left\|\left(\mathbf{I} - \frac{\mathbf{u}\mathbf{u}^T}{\|\mathbf{u}\|_2^2}\right)\frac{\nabla\mathcal{N}(\mathbf{u})}{\|\mathbf{u}\|_2^{L-1}}\right\|_2\left\|\frac{d}{dt}\left(\frac{\mathbf{u}}{\|\mathbf{u}\|_2}\right)\right\|_2 \\
&= \|\mathbf{u}\|_2\left\|\nabla\tilde{\mathcal{N}}(\mathbf{u})\right\|_2\left\|\frac{d}{dt}\left(\frac{\mathbf{u}}{\|\mathbf{u}\|_2}\right)\right\|_2 \geq \frac{1}{\Psi'(\overline{f} - \tilde{\mathcal{N}}(\mathbf{u}))}\left\|\frac{d}{dt}\left(\frac{\mathbf{u}}{\|\mathbf{u}\|_2}\right)\right\|_2.
\end{aligned}$$

In the above chain of equalities and inequalities, we used Lemma 10 in the third equality, and the inequality follows from eq. (28). Now, since $\overline{f} - \tilde{\mathcal{N}}(\mathbf{u}(t)) \in [0, \nu)$, for all $t \in [T, T^*)$, and $\Psi' > 0$ on $[0, \nu)$, we can take $\Psi'(\overline{f} - \tilde{\mathcal{N}}(\mathbf{u}(t)))$ to the left hand side to get

$$\left\|\frac{d}{dt}\left(\frac{\mathbf{u}}{\|\mathbf{u}\|_2}\right)\right\|_2 \leq -\frac{d\Psi(\overline{f} - \tilde{\mathcal{N}}(\mathbf{u}))}{dt}, \forall t \in [T, T^*).$$

Next, integrating the above inequality on both the sides from $T$ to any $t_1 \in [T, T^*)$, we have

$$\int_T^{t_1}\left\|\frac{d}{dt}\left(\frac{\mathbf{u}}{\|\mathbf{u}\|_2}\right)\right\|_2 dt \leq \Psi(\overline{f} - \tilde{\mathcal{N}}(\mathbf{u}(T))) - \Psi(\overline{f} - \tilde{f}(\mathbf{u}(t_1))) \leq \Psi(\overline{f} - \tilde{\mathcal{N}}(\mathbf{u}(T))) < \infty.$$

From the above inequality, we further have

$$\begin{aligned}
\int_0^{T^*}\left\|\frac{d}{dt}\left(\frac{\mathbf{u}}{\|\mathbf{u}\|_2}\right)\right\|_2 dt &= \int_0^T\left\|\frac{d}{dt}\left(\frac{\mathbf{u}}{\|\mathbf{u}\|_2}\right)\right\|_2 dt + \int_T^{T^*}\left\|\frac{d}{dt}\left(\frac{\mathbf{u}}{\|\mathbf{u}\|_2}\right)\right\|_2 dt \\
&\leq \int_0^T\left\|\frac{d}{dt}\left(\frac{\mathbf{u}}{\|\mathbf{u}\|_2}\right)\right\|_2 dt + \Psi(\overline{f} - \tilde{\mathcal{N}}(\mathbf{u}(T))) < \infty,
\end{aligned}$$

which completes the proof. We next show that there exists $\kappa > 0$ such that

$$\|\mathbf{u}(t)\|_2 \geq \frac{\kappa}{(T^* - t)^{1/(L-2)}}, \forall t \in [0, T^*), \tag{29}$$

which will be useful later. Define $\mathbf{h}(t)$ such that

$$\mathbf{u}(t) = \frac{\mathbf{h}(t)}{(T^* - t)^{1/(L-2)}}.$$

Our goal is to show $\|\mathbf{h}(t)\|_2 \geq \kappa$, for all $t \in [0, T^*)$ and for some $\kappa > 0$. Now, since $\|\mathbf{u}(t)\|_2 > 0$, for all $t \in [0, T^*)$, therefore, $\|\mathbf{h}(t)\|_2 > 0$, for all $t \in [0, T^*)$. We next show $\lim_{t \to T^*} \|\mathbf{h}(t)\|_2 > 0$. Since

$$\frac{1}{2} \frac{d\|\mathbf{u}\|_2^2}{dt} = L\mathcal{N}(\mathbf{u}) = L\mathcal{N}\left(\frac{\mathbf{u}}{\|\mathbf{u}\|_2}\right) \|\mathbf{u}\|_2^L,$$

we have

$$\frac{1}{\|\mathbf{u}\|_2^{L-1}} \frac{d\|\mathbf{u}\|_2}{dt} = L\mathcal{N}\left(\frac{\mathbf{u}}{\|\mathbf{u}\|_2}\right).$$

Integrating both sides from $0$ to $t \in (0, T^*)$, we get

$$\frac{1}{L-2}\left(\frac{1}{\|\mathbf{u}(0)\|_2^{L-2}} - \frac{1}{\|\mathbf{u}(t)\|_2^{L-2}}\right) = \int_0^t L\mathcal{N}\left(\frac{\mathbf{u}(s)}{\|\mathbf{u}(s)\|_2}\right) ds.$$

Re-arranging the above equation gives us

$$\frac{1}{\|\mathbf{u}(0)\|_2^{L-2}} - \int_0^t L(L-2)\mathcal{N}\left(\frac{\mathbf{u}(s)}{\|\mathbf{u}(s)\|_2}\right) ds = \frac{1}{\|\mathbf{u}(t)\|_2^{L-2}}. \tag{30}$$

Substituting $\mathbf{u}(t) = \mathbf{h}(t)/(T^* - t)^{1/(L-2)}$, we get

$$\frac{1/\|\mathbf{u}(0)\|_2^{L-2} - \int_0^t L(L-2)\mathcal{N}(\mathbf{u}(s)/\|\mathbf{u}(s)\|_2)ds}{(T^* - t)} = \frac{1}{\|\mathbf{h}(t)\|_2^{L-2}}.$$

In LHS of the above equality, at $t = T^*$, the denominator is obviously $0$, and the numerator is also $0$ since, using eq. (30),

$$\frac{1}{\|\mathbf{u}(0)\|_2^{L-2}} - \int_0^{T^*} L(L-2)\mathcal{N}\left(\frac{\mathbf{u}(s)}{\|\mathbf{u}(s)\|_2}\right) ds$$
$$= \lim_{t \to T^*} \frac{1}{\|\mathbf{u}(0)\|_2^{L-2}} - \int_0^t L(L-2)\mathcal{N}\left(\frac{\mathbf{u}(s)}{\|\mathbf{u}(s)\|_2}\right) ds = \lim_{t \to T^*} \frac{1}{\|\mathbf{u}(t)\|_2^{L-2}} = 0.$$

Therefore, using L'Hopital's rule,

$$\frac{1}{\|\mathbf{h}(T^*)\|_2^{L-2}} = \frac{\lim_{t \to T^*} -L(L-2)\mathcal{N}(\mathbf{u}(t)/\|\mathbf{u}(t)\|_2)}{-1} = \lim_{t \to T^*} L(L-2)\mathcal{N}(\mathbf{u}(t)/\|\mathbf{u}(t)\|_2) > 0.$$

Hence, $\|\mathbf{h}(t)\|_2 > 0$, for all $t \in [0, T^*]$, which implies there exists some $\kappa > 0$ such that $\|\mathbf{h}(t)\|_2 \geq \kappa$, for all $t \in [0, T^*)$ and proving eq. (29).

**Case 2:** $\mathcal{N}(\mathbf{u}(0)) \leq 0$.
In this case, we may further assume that $\mathcal{N}(\mathbf{u}(t)) \leq 0$, for all $t \geq 0$, since if for some $\bar{t}$, $\mathcal{N}(\mathbf{u}(\bar{t})) > 0$, then we can choose $\bar{t}$ be the new starting time and use the proof for Case 1 to show that $\mathbf{u}(t)$ becomes unbounded at some finite time and also converges in direction. Therefore, we assume $\mathcal{N}(\mathbf{u}(t)) \leq 0$, for all $t \geq 0$.

Now, since

$$\frac{1}{2} \frac{d\|\mathbf{u}\|_2^2}{dt} = \mathbf{u}^\top \dot{\mathbf{u}} = L\mathcal{N}(\mathbf{u}(t)) \leq 0, \forall t \geq 0,$$

we have that $\|\mathbf{u}(t)\|_2$ decreases with time, and $\lim_{t \to \infty} \|\mathbf{u}(t)\|_2$ exists. Now, if $\lim_{t \to \infty} \|\mathbf{u}(t)\|_2 = 0$, then we have $\lim_{t \to \infty} \mathbf{u}(t) = \mathbf{0}$ and our proof is complete.

Else, suppose $\lim_{t \to \infty} \|\mathbf{u}(t)\|_2 = \eta > 0$. Then, we have $\|\mathbf{u}(t)\|_2 \geq \eta$, for all $t \geq 0$, since $\|\mathbf{u}(t)\|_2$ decreases with time. We next show that $\lim_{t \to \infty} \mathcal{N}(\mathbf{u}(t)) = 0$. From Lemma 12, we know

$$\frac{d\mathcal{N}(\mathbf{u})}{dt} = \|\dot{\mathbf{u}}\|_2^2, \forall t \geq 0.$$

Thus, $\mathcal{N}(\mathbf{u}(t))$ increases with time. Since we have assumed $\mathcal{N}(\mathbf{u}(t)) \leq 0$ for all $t \geq 0$, by monotone convergence we have that $\lim_{t \to \infty} \mathcal{N}(\mathbf{u}(t))$ exists. Now, to show $\lim_{t \to \infty} \mathcal{N}(\mathbf{u}(t)) = 0$, we assume for the sake of contradiction that $\lim_{t \to \infty} \mathcal{N}(\mathbf{u}(t)) = -\gamma < 0$. Then, for all $t \geq 0$, we have $\mathcal{N}(\mathbf{u}(t)) \geq -\gamma$, since $\mathcal{N}(\mathbf{u}(t))$ is an increasing function of time. Thus,

$$\frac{1}{2}\frac{d\|\mathbf{u}\|_2^2}{dt} = \mathbf{u}^\top \dot{\mathbf{u}} = L\mathcal{N}(\mathbf{u}(t)) \leq -L\gamma, \forall t \geq 0.$$

From the above equation we get that $\|\mathbf{u}(t)\|_2$ would become smaller than $\eta$ after some finite amount of time has elapsed. Since $\|\mathbf{u}(t)\|_2 \geq \eta$, for all $t \geq 0$, this leads to a contradiction. Therefore, $\lim_{t \to \infty} \mathcal{N}(\mathbf{u}(t)) = 0$. This also implies $\lim_{t \to \infty} \tilde{\mathcal{N}}(\mathbf{u}(t)) = 0$.

We next prove that $\lim_{t \to \infty} \mathbf{u}(t)/\|\mathbf{u}(t)\|_2$ exists. We first define $\hat{\mathbf{u}}(t) = 2\mathbf{u}(t)/\eta$. Now, if $\hat{\mathbf{u}}(t)$ converges in direction, then $\mathbf{u}(t)$ also converges in direction. To prove $\hat{\mathbf{u}}(t)$ converges in direction we can follow the same approach as in Case 1, specifically from eq. (27) onward. This transformation is essential because, recall, to use Lemma 6 in Case 1 we had to choose $T$ large enough such that $\|\mathbf{u}(t)\|_2 > 1$, for all $t \geq T$. Here, $\|\mathbf{u}(t)\|_2$ may never be greater than 1, but $\|\hat{\mathbf{u}}(t)\|_2 \geq 2$, for all $t \geq 0$.

So far we have established that either $\frac{\mathbf{u}(t)}{\|\mathbf{u}(t)\|_2}$ converges, or $\mathbf{u}(t)$ converges to $\mathbf{0}$. We next show that if $\frac{\mathbf{u}(t)}{\|\mathbf{u}(t)\|_2}$ converges to $\mathbf{u}_*$, then $\mathbf{u}_*$ must be a non-negative KKT point of the constrained NCF.

From the proof so far we already know that $\mathcal{N}(\mathbf{u}_*) = \tilde{\mathcal{N}}(\mathbf{u}_*) \geq 0$. Therefore, from Lemma 11, we only need to show

$$L\mathcal{N}(\mathbf{u}_*)\mathbf{u}_* = \nabla\mathcal{N}(\mathbf{u}_*), \tag{31}$$

For the sake of contradiction assume that there exists some $\gamma > 0$ such that

$$\|\nabla\mathcal{N}(\mathbf{u}_*) - L\mathcal{N}(\mathbf{u}_*)\mathbf{u}_*\|_2 \geq \gamma. \tag{32}$$

For any $\epsilon > 0$, we define the set $\mathbf{u}_\epsilon = \{\mathbf{u} : \|\mathbf{u} - \mathbf{u}_*\|_2 \leq \epsilon\}$. Since $\mathcal{N}(\mathbf{u})$ has locally Lipschitz gradient, given $\gamma$, we can choose sufficiently small $\epsilon \in (0, 1)$ such that for all $\mathbf{u} \in \mathbf{u}_\epsilon$, we have

$$\|\nabla\mathcal{N}(\mathbf{u}) - \nabla\mathcal{N}(\mathbf{u}_*)\|_2 \leq \gamma/4. \tag{33}$$

Since $\frac{\mathbf{u}(t)}{\|\mathbf{u}(t)\|_2}$ converges to $\mathbf{u}_*$ and $\mathcal{N}(\mathbf{u})$ is continuous, we choose $T$ large enough such that for all $t \geq T$,

$$\left\|\frac{\mathbf{u}(t)}{\|\mathbf{u}(t)\|_2} - \mathbf{u}_*\right\|_2 \leq \epsilon, \left\|\frac{L\mathbf{u}(t)}{\|\mathbf{u}(t)\|_2}\mathcal{N}\left(\frac{\mathbf{u}(t)}{\|\mathbf{u}(t)\|_2}\right) - L\mathbf{u}_*\mathcal{N}(\mathbf{u}_*)\right\| \leq \gamma/4. \tag{34}$$

From the two cases discussed above we know that if $\frac{\mathbf{u}(t)}{\|\mathbf{u}(t)\|_2}$ converges, then are two scenarios: $\|\mathbf{u}(t)\|_2$ either goes to infinity or $\|\mathbf{u}(t)\|_2$ remains finite, bounded away from 0, for all $t \geq 0$ and $\tilde{\mathcal{N}}(\mathbf{u}(t))$ converges to 0. Now, if $\|\mathbf{u}(t)\|_2$ goes to infinity, then we may assume $T$ is large enough such that for some $\kappa > 0$

$$\|\mathbf{u}(t)\|_2 \geq \frac{\kappa}{(T^* - t)^{1/(L-2)}}, \forall t \in [T, T^*), \tag{35}$$

where $\lim_{t \to T^*} \mathbf{u}(t)/\|\mathbf{u}(t)\|_2 = \mathbf{u}_*$. In the second scenario, we may assume that $T$ is large enough such that for some $\eta > 0$, we have

$$\|\mathbf{u}(t)\|_2 \geq \eta, \forall t \geq T. \tag{36}$$

Now, for all non-zero $\mathbf{u} \in \mathbb{R}^k$ we have

$$
\begin{aligned}
\left\|\left(\mathbf{I} - \frac{\mathbf{u}\mathbf{u}^\top}{\|\mathbf{u}\|_2^2}\right) \frac{\nabla\mathcal{N}(\mathbf{u})}{\|\mathbf{u}\|_2^{L-1}}\right\|_2 &= \left\|\frac{\nabla\mathcal{N}(\mathbf{u})}{\|\mathbf{u}\|_2^{L-1}} - \frac{L\mathbf{u}\mathcal{N}(\mathbf{u})}{\|\mathbf{u}\|_2^{L+1}}\right\|_2 \\
&= \left\|\nabla\mathcal{N}\left(\frac{\mathbf{u}}{\|\mathbf{u}\|_2}\right) - \frac{L\mathbf{u}}{\|\mathbf{u}\|_2}\mathcal{N}\left(\frac{\mathbf{u}}{\|\mathbf{u}\|_2}\right)\right\|_2 \\
&\geq \left\|\nabla\mathcal{N}(\mathbf{u}_*) - \frac{L\mathbf{u}}{\|\mathbf{u}\|_2}\mathcal{N}\left(\frac{\mathbf{u}}{\|\mathbf{u}\|_2}\right)\right\|_2 - \left\|\nabla\mathcal{N}\left(\frac{\mathbf{u}}{\|\mathbf{u}\|_2}\right) - \nabla\mathcal{N}(\mathbf{u}_*)\right\|_2 \\
&\geq \|\nabla\mathcal{N}(\mathbf{u}_*) - L\mathbf{u}_*\mathcal{N}(\mathbf{u}_*)\|_2 - \left\|L\mathbf{u}_*\mathcal{N}(\mathbf{u}_*) - \frac{L\mathbf{u}}{\|\mathbf{u}\|_2}\mathcal{N}\left(\frac{\mathbf{u}}{\|\mathbf{u}\|_2}\right)\right\|_2 \\
&\quad - \left\|\nabla\mathcal{N}\left(\frac{\mathbf{u}}{\|\mathbf{u}\|_2}\right) - \nabla\mathcal{N}(\mathbf{u}_*)\right\|_2,
\end{aligned}
$$

where in the second equality we used $(L-1)$-homogeneity of $\nabla\mathcal{N}(\mathbf{u})$ and $L$-homogeneity of $\mathcal{N}(\mathbf{u})$. The inequalities make repeated use of triangle inequality of norms. Hence, using eq. (32), eq. (33) and eq. (34), for all $t \geq T$, we have

$$
\left\|\left(\mathbf{I} - \frac{\mathbf{u}(t)\mathbf{u}(t)^\top}{\|\mathbf{u}(t)\|_2^2}\right) \frac{\nabla\mathcal{N}(\mathbf{u}(t))}{\|\mathbf{u}(t)\|_2^{L-1}}\right\|_2 \geq \gamma/2.
$$

Using the above inequality and eq. (23) we have

$$
\frac{d}{dt}\left(\mathcal{N}\left(\frac{\mathbf{u}(t)}{\|\mathbf{u}(t)\|_2}\right)\right) = \left\|\left(\mathbf{I} - \frac{\mathbf{u}(t)\mathbf{u}(t)^\top}{\|\mathbf{u}(t)\|_2^2}\right) \frac{\nabla\mathcal{N}(\mathbf{u}(t))}{\|\mathbf{u}(t)\|_2^{L/2}}\right\|_2^2 \geq \gamma^2\|\mathbf{u}(t)\|_2^{L-2}/4.
$$

Now, if $\|\mathbf{u}(t)\|_2$ goes to infinity, then, from eq. (35), we have

$$
\frac{d}{dt}\left(\mathcal{N}\left(\frac{\mathbf{u}(t)}{\|\mathbf{u}(t)\|_2}\right)\right) \geq \frac{\gamma^2 \kappa^{L-2}}{4(T^* - t)}, \forall t \in [T, T^*)
$$

Integrating the above equation from $T$ to $t \in (T, T^*)$, we get

$$
\mathcal{N}\left(\frac{\mathbf{u}(t)}{\|\mathbf{u}(t)\|_2}\right) - \mathcal{N}\left(\frac{\mathbf{u}(T)}{\|\mathbf{u}(T)\|_2}\right) \geq \frac{\gamma^2 \kappa^{L-2}}{4} \int_T^{T^*} \frac{1}{(T^* - t)} = \frac{\gamma^2 \kappa^{L-2}}{4} \ln\left(\frac{T^* - T}{T^* - t}\right).
$$

As $t$ approaches $T^*$, the LHS remains finite but the RHS goes to infinity, leading to a contradiction. In the second scenario, where $\|\mathbf{u}(t)\|_2$ remains finite, bounded away from 0, for all $t \geq 0$, we have, from eq. (36),

$$
\frac{d}{dt}\left(\mathcal{N}\left(\frac{\mathbf{u}(t)}{\|\mathbf{u}(t)\|_2}\right)\right) \geq \gamma^2 \eta^{L-2}/4, \forall t \geq T.
$$

Integrating the above equation from $T$ to $t \geq T$, we get

$$
\mathcal{N}\left(\frac{\mathbf{u}(t)}{\|\mathbf{u}(t)\|_2}\right) - \mathcal{N}\left(\frac{\mathbf{u}(T)}{\|\mathbf{u}(T)\|_2}\right) \geq \gamma^2 \eta^{L-2}(t - T)/4.
$$

Since $\mathcal{N}(\mathbf{u}/\|\mathbf{u}\|_2)$ is bounded, the above inequality can not be true for sufficiently large $t$, leading to a contradiction. This completes our proof of Lemma 2. $\qquad \square$

Before proceeding to proof of Theorem 1, we state a useful lemma.

**Lemma 14.** *Let $\mathbf{u}_0$ be a fixed vector and let $\mathbf{u}(t)$ be the solution of*

$$
\frac{d\mathbf{u}}{dt} = \nabla\mathcal{N}(\mathbf{u}) = \mathcal{J}(\mathbf{X}; \mathbf{u})^\top \mathbf{z}, \mathbf{u}(0) = \mathbf{u}_0. \tag{37}
$$

*If $\lim_{t\to\infty} \mathbf{u}(t) \neq \mathbf{0}$, then there exists $\eta > 0$ and $T \geq 0$ such that $\|\mathbf{u}(t)\|_2 \geq \eta$, for all $t \geq T$. Further, for any $\epsilon \in (0,1)$ there exists $T_\epsilon \geq T$ and $B_\epsilon$ such that*

$$\frac{\mathbf{u}(T_\epsilon)^\top \mathbf{u}_*}{\|\mathbf{u}(T_\epsilon)\|_2} \geq 1 - \epsilon, \ \ and \ \|\mathbf{u}(t)\|_2 \leq B_\epsilon, \forall t \leq T_\epsilon, \tag{38}$$

*where $\mathbf{u}_*$ is a non-negative KKT point of*

$$\max_{\|\mathbf{u}\|_2^2=1} \mathcal{N}(\mathbf{u}) = \mathbf{z}^\top \mathcal{H}(\mathbf{X}; \mathbf{u}). \tag{39}$$

*Proof.* Similar to the proof of Lemma 2, we consider two cases.

**Case 1: $\mathcal{N}(\mathbf{u}(0)) > 0$.**
Let $\mathcal{N}(\mathbf{u}(0)) = \gamma > 0$, thus $\|\mathbf{u}(0)\|_2 > 0$. From Lemma 13, $\|\mathbf{u}(t)\|_2 \geq \|\mathbf{u}(0)\|_2$, for all $t \geq 0$, which implies we can choose $\eta = \|\mathbf{u}(0)\|_2$ and $T = 0$.

Next, from Lemma 2 and its proof we know that for some finite $T^*$, $\lim_{t\to T^*} \frac{\mathbf{u}(t)}{\|\mathbf{u}(t)\|_2} = \mathbf{u}_*$, where $\mathbf{u}_*$ is a non-negative KKT point of the constrained NCF eq. (39). Thus, for any $\epsilon \in (0,1)$, we can choose $T_\epsilon \geq T = 0$, and large enough but less than $T^*$ such that

$$\frac{\mathbf{u}(T_\epsilon)^\top \mathbf{u}_*}{\|\mathbf{u}(T_\epsilon)\|_2} \geq 1 - \epsilon. \tag{40}$$

Since for all $t \in (0, T^*)$, $\mathbf{u}(t)$ is finite, there exists $B_\epsilon$ such that $\|\mathbf{u}(T_\epsilon)\|_2 \leq B_\epsilon$. We note that as $\epsilon$ gets smaller, we may have to choose $T_\epsilon$ closer to $T^*$. This implies that, since $\lim_{t\to T^*} \|\mathbf{u}(t)\|_2 = \infty$, $B_\epsilon$ will increase. However, for a fixed $\epsilon$, both $T_\epsilon$ and $B_\epsilon$ will be fixed.

**Case 2: $\mathcal{N}(\mathbf{u}(0)) \leq 0$.**
In this case, we can also assume that $\mathcal{N}(\mathbf{u}(t)) \leq 0$, for all $t \geq 0$, since if $\mathcal{N}(\mathbf{u}(\bar{t})) > 0$, for some $\bar{t}$, then from Lemma 13, we know $\|\mathbf{u}(t)\|_2 \geq \|\mathbf{u}(\bar{t})\|_2$, for all $t \geq \bar{t}$. Moreover, since $\mathcal{N}(\mathbf{u}(\bar{t})) > 0$ implies $\|\mathbf{u}(\bar{t})\|_2 > 0$, we can choose $\eta = \|\mathbf{u}(\bar{t})\|_2$ and $T = \bar{t}$. Also, in this case, from the proof of Lemma 2, we know that for some finite $T^*$, $\lim_{t\to T^*} \frac{\mathbf{u}(t)}{\|\mathbf{u}(t)\|_2} = \mathbf{u}_*$, where $\mathbf{u}_*$ is a non-negative KKT point of the constrained NCF eq. (39). Thus, similar to Case 1, for any $\epsilon \in (0,1)$ we can choose $T_\epsilon \geq \bar{t}$ and $B_\epsilon$ as desired.

Hence, suppose $\mathcal{N}(\mathbf{u}(t)) \leq 0$, for all $t \geq 0$. Then, since

$$\frac{1}{2}\frac{d\|\mathbf{u}\|_2^2}{dt} = \mathbf{u}^\top \dot{\mathbf{u}} = L\mathcal{N}(\mathbf{u}(t)) \leq 0, \forall t \geq 0,$$

we have that $\|\mathbf{u}(t)\|_2$ is a decreasing function with time, and hence, $\lim_{t\to\infty} \|\mathbf{u}(t)\|_2$ exists. Also,

$$\|\mathbf{u}(t)\|_2 \geq \lim_{t\to\infty} \|\mathbf{u}(t)\|_2, \ \text{for all } t \geq 0.$$

Since we have assumed $\lim_{t\to\infty} \mathbf{u}(t) \neq \mathbf{0}$, we know $\lim_{t\to\infty} \|\mathbf{u}(t)\|_2 > 0$, and we can choose $\eta = \lim_{t\to\infty} \|\mathbf{u}(t)\|_2$ and $T = 0$. Further, since $\lim_{t\to\infty} \frac{\mathbf{u}(t)}{\|\mathbf{u}(t)\|_2} = \mathbf{u}_*$, where $\mathbf{u}_*$ is a non-negative KKT point of the constrained NCF eq. (39), for any $\epsilon \in (0,1)$, we can choose $T_\epsilon$ large enough such that

$$\frac{\mathbf{u}(T_\epsilon)^\top \mathbf{u}_*}{\|\mathbf{u}(T_\epsilon)\|_2} \geq 1 - \epsilon. \tag{41}$$

Also, since $\|\mathbf{u}(t)\|_2$ is a decreasing function with time, we have $\|\mathbf{u}(t)\|_2 \in [\eta, \|\mathbf{u}(0)\|_2]$, for all $t \geq 0$. Thus, we can choose $B_\epsilon = \|\mathbf{u}(0)\|_2$. $\qquad\square$

### B.2 Proof of Theorem 1

*Proof.* Let $\mathbf{w}_0$ be a fixed unit norm vector. Suppose $\mathbf{u}(t)$ is the solution of

$$\frac{d\mathbf{u}}{dt} = \nabla \mathcal{N}_{-\ell'(\mathbf{0},\mathbf{y}),\mathcal{H}}(\mathbf{u}) = -\mathcal{J}\left(\mathbf{X};\mathbf{u}\right)^\top \ell'(\mathbf{0},\mathbf{y}), \mathbf{u}(0) = \mathbf{w}_0. \tag{42}$$

From Lemma 2, we know that there are two possibilities. Either $\mathbf{u}(t)$ converges to $\mathbf{0}$, or $\mathbf{u}(t)$ converges in direction to a non-negative KKT point of the constrained NCF.

If $\mathbf{u}(t)$ converges to $\mathbf{0}$, then we define $\eta = 2$. Then, for any $\epsilon \in (0, \eta/2)$, we define $B_\epsilon = \epsilon$ and choose $T_\epsilon$ large enough such that

$$\|\mathbf{u}(T_\epsilon)\|_2 \leq B_\epsilon = \epsilon. \tag{43}$$

Else, if $\mathbf{u}(t)$ does not converge to $\mathbf{0}$, then from Lemma 14, there exists $\tilde{\eta} > 0$ and $T$ such that $\|\mathbf{u}(t)\|_2 \geq \tilde{\eta}$, for all $t \geq T$. We further define $\eta = \min(\tilde{\eta}, 1)$. Then, from Lemma 14, for any $\epsilon \in (0, \eta/2)$ there exists $T_\epsilon \geq T$ and $B_\epsilon$ such that

$$\frac{\mathbf{u}(T_\epsilon)^\top \mathbf{u}_*}{\|\mathbf{u}(T_\epsilon)\|_2} \geq 1 - \epsilon, \text{ and } \|\mathbf{u}(t)\|_2 \leq B_\epsilon, \forall t \leq T_\epsilon, \tag{44}$$

where $\mathbf{u}_*$ is a non-negative KKT point of

$$\max_{\|\mathbf{u}\|_2^2 = 1} \mathcal{N}_{-\ell'(\mathbf{0},\mathbf{y}),\mathcal{H}}(\mathbf{u}) = -\mathcal{H}\left(\mathbf{X};\mathbf{u}\right)^\top \ell'(\mathbf{0},\mathbf{y}).$$

Further, since $T_\epsilon \geq T$, we have

$$\|\mathbf{u}(T_\epsilon)\|_2 \geq \tilde{\eta} \geq \eta. \tag{45}$$

We next define other $\epsilon$ dependent parameters which are useful in our proof. Let $\tilde{B}_\epsilon = B_\epsilon + \epsilon$. Since $\mathcal{H}(\mathbf{x};\mathbf{w})$ has locally Lipschitz gradient, there exists $K_\epsilon > 0$ such that if $\|\mathbf{w}_1\|_2, \|\mathbf{w}_2\|_2 \leq \tilde{B}_\epsilon$, then

$$\|\mathcal{J}\left(\mathbf{X};\mathbf{w}_1\right) - \mathcal{J}\left(\mathbf{X};\mathbf{w}_2\right)\|_2 \leq K_\epsilon \|\mathbf{w}_1 - \mathbf{w}_2\|_2. \tag{46}$$

Also, since $\nabla_{\hat{y}}\ell(\hat{y}, y)$ is locally Lipschitz in $\hat{y}$, there exists $\hat{\beta}$ such that if $\|\mathbf{z}_1\|_2, \|\mathbf{z}_2\|_2 \leq \beta\tilde{B}_\epsilon^L$, then

$$\|\ell'\left(\mathbf{z}_1,\mathbf{y}\right) - \ell'\left(\mathbf{z}_2,\mathbf{y}\right)\|_2 \leq \hat{\beta}\|\mathbf{z}_1 - \mathbf{z}_2\|_2, \tag{47}$$

where recall

$$\beta = \sup\{\|\mathcal{H}(\mathbf{X};\mathbf{w})\|_2 : \mathbf{w} \in \mathcal{S}^{k-1}\}. \tag{48}$$

We further define $C_\epsilon = \beta\hat{\beta}\tilde{B}_\epsilon^L$, and

$$\bar{\delta}^L = \min\left(1, \frac{\epsilon}{4\alpha\tilde{B}_\epsilon^{L-1}C_\epsilon T_\epsilon e^{2T_\epsilon K_\epsilon \|\ell'(\mathbf{0},\mathbf{y})\|_2}}\right), \tag{49}$$

where recall

$$\alpha = \sup\{\|\mathcal{J}(\mathbf{X};\mathbf{w})\|_2 : \mathbf{w} \in \mathcal{S}^{k-1}\}. \tag{50}$$

Now, for any $\delta \in (0, \bar{\delta})$, let $\mathbf{w}(t)$ be the solution of

$$\dot{\mathbf{w}}(t) = -\nabla\mathcal{L}(\mathbf{w}(t)), \mathbf{w}(0) = \delta\mathbf{w}_0, \tag{51}$$

then, we can rewrite the above differential equation as

$$\dot{\mathbf{w}} = -\sum_{i=1}^{n} \nabla_{\hat{y}}\ell\left(\mathcal{H}(\mathbf{x}_i;\mathbf{w}), y_i\right)\nabla\mathcal{H}(\mathbf{x}_i;\mathbf{w}) = -\mathcal{J}(\mathbf{X};\mathbf{w})^\top \ell'(\mathcal{H}(\mathbf{X};\mathbf{w}),\mathbf{y}), \mathbf{w}(0) = \delta\mathbf{w}_0. \tag{52}$$

We define $\xi(t) := \ell'(\mathcal{H}(\mathbf{X}; \mathbf{w}), \mathbf{y}) - \ell'(\mathbf{0}, \mathbf{y})$, then the dynamics of $\mathbf{w}(t)$ can be written as

$$\dot{\mathbf{w}} \in -\mathcal{J}(\mathbf{X}; \mathbf{w})^\top \ell'(\mathcal{H}(\mathbf{X}; \mathbf{w}), \mathbf{y}) = -\mathcal{J}(\mathbf{X}; \mathbf{w})^\top (\ell'(\mathbf{0}, \mathbf{y}) + \xi(t)). \tag{53}$$

Now, define $\mathbf{s}(t) = \frac{1}{\delta} \mathbf{w}\left(\frac{t}{\delta^{L-2}}\right)$, then $\mathbf{s}(0) = \mathbf{w}_0$, and

$$\begin{aligned}
\frac{d\mathbf{s}}{dt} = \frac{d}{dt}\left(\frac{1}{\delta}\mathbf{w}\left(\frac{t}{\delta^{L-2}}\right)\right) &= \frac{1}{\delta^{L-1}}\dot{\mathbf{w}}\left(\frac{t}{\delta^{L-2}}\right) \\
&= -\frac{1}{\delta^{L-1}}\mathcal{J}\left(\mathbf{X}; \mathbf{w}\left(\frac{t}{\delta^{L-2}}\right)\right)^\top \left(\ell'(\mathbf{0}, \mathbf{y}) + \xi\left(\frac{t}{\delta^{L-2}}\right)\right) \\
&= -\mathcal{J}\left(\mathbf{X}; \frac{1}{\delta}\mathbf{w}\left(\frac{t}{\delta^{L-2}}\right)\right)^\top \left(\ell'(\mathbf{0}, \mathbf{y}) + \xi\left(\frac{t}{\delta^{L-2}}\right)\right) \\
&= -\mathcal{J}\left(\mathbf{X}; \mathbf{s}(t)\right)^\top \left(\ell'(\mathbf{0}, \mathbf{y}) + \xi\left(\frac{t}{\delta^{L-2}}\right)\right),
\end{aligned} \tag{54}$$

where in third equality, from Lemma 9, we used $(L-1)$-homogeneity of $\mathcal{J}(\mathbf{X}; \mathbf{w})$. Using $L$-homogeneity of $\mathcal{H}(\mathbf{X}; \mathbf{w})$, we have

$$\begin{aligned}
\xi(t/\delta^{L-2}) &= \ell'(\mathcal{H}(\mathbf{X}; \mathbf{w}(t/\delta^{L-2})), \mathbf{y}) - \ell'(\mathbf{0}, \mathbf{y}) \\
&= \ell'(\delta^L \mathcal{H}(\mathbf{X}; \mathbf{w}(t/\delta^{L-2})/\delta), \mathbf{y}) - \ell'(\mathbf{0}, \mathbf{y}) \\
&= \ell'(\delta^L \mathcal{H}(\mathbf{X}; \mathbf{s}(t)), \mathbf{y}) - \ell'(\mathbf{0}, \mathbf{y}).
\end{aligned}$$

Let $\tilde{\xi}(t) = \xi(t/\delta^{L-2})$, then

$$\dot{\mathbf{s}} = -\mathcal{J}(\mathbf{X}; \mathbf{s}(t))^\top (\ell'(\mathbf{0}, \mathbf{y}) + \tilde{\xi}(t)), \mathbf{s}(0) = \mathbf{w}_0.$$

Next, we will show that $\|\mathbf{s}(t) - \mathbf{u}(t)\|_2 \le \epsilon$, for all $t \in [0, T_\epsilon]$. We first note that, $\mathbf{s}(0) = \mathbf{u}(0) = \mathbf{w}_0$. Define

$$\overline{T}_\delta = \inf_{t \ge 0}\{t : \|\mathbf{s}(t) - \mathbf{u}(t)\|_2 = \epsilon\}.$$

Here, $\overline{T}_\delta$ indicates the time when $\|\mathbf{s}(t) - \mathbf{u}(t)\|_2$ becomes $\epsilon$ for the first time, and $\delta$ in the subscript indicates that $\overline{T}_\delta$ could change with $\delta$. By definition of $\overline{T}_\delta$, for all $t \in [0, \overline{T}_\delta]$, $\|\mathbf{s}(t) - \mathbf{u}(t)\|_2 \le \epsilon$. Now, if we can show $\overline{T}_\delta > T_\epsilon$, then we would have proved $\|\mathbf{s}(t) - \mathbf{u}(t)\|_2 \le \epsilon$, for all $t \in [0, T_\epsilon]$.

For the sake of contradiction, suppose $\overline{T}_\delta \le T_\epsilon$. This implies that $\|\mathbf{u}(t)\|_2 \le B_\epsilon$ and $\|\mathbf{s}(t)\|_2 \le B_\epsilon + \epsilon = \tilde{B}_\epsilon$, for all $t \in [0, \overline{T}_\delta]$. Now, using eq. (46), we have

$$\|\mathcal{J}(\mathbf{X}; \mathbf{u}(t)) - \mathcal{J}(\mathbf{X}; \mathbf{s}(t))\|_2 \le K_\epsilon \|\mathbf{u}(t) - \mathbf{s}(t)\|_2, \forall t \in [0, \overline{T}_\delta]. \tag{55}$$

Next, since $\delta \le 1$ and

$$\|\mathcal{H}(\mathbf{X}; \mathbf{s}(t))\|_2 \le \beta\|\mathbf{s}(t)\|_2^L \le \beta\tilde{B}_\epsilon^L, \forall t \in [0, \overline{T}_\delta],$$

using eq. (47), we have

$$\|\tilde{\xi}(t)\|_2 = \|\ell'(\delta^L \mathcal{H}(\mathbf{X}; \mathbf{s}(t)), \mathbf{y}) - \ell'(\mathbf{0}, \mathbf{y})\|_2 \le \delta^L \hat{\beta}\|\mathcal{H}(\mathbf{X}; \mathbf{s}(t))\|_2 \le \delta^L \hat{\beta}\beta\tilde{B}_\epsilon^L, \forall t \in [0, \overline{T}_\delta].$$

Hence, by the definition of $C_\epsilon$, we have

$$\|\tilde{\xi}(t)\|_2 \le C_\epsilon \delta^L, \text{ for all } t \in [0, \overline{T}_\delta]. \tag{56}$$

Thus, for any $t \in [0, \overline{T}_\delta]$

$$\begin{aligned}
\frac{1}{2}\frac{d\|\mathbf{s} - \mathbf{u}\|_2^2}{dt} &= (\mathbf{s} - \mathbf{u})^\top (\dot{\mathbf{s}} - \dot{\mathbf{u}}) \\
&= -(\mathbf{s} - \mathbf{u})^\top (\mathcal{J}(\mathbf{X}; \mathbf{s}) - \mathcal{J}(\mathbf{X}; \mathbf{u}))^\top \ell'(\mathbf{0}, \mathbf{y}) - (\mathbf{s} - \mathbf{u})^\top \mathcal{J}(\mathbf{X}; \mathbf{s})^\top \tilde{\xi}(t) \\
&\le K_\epsilon \|\ell'(\mathbf{0}, \mathbf{y})\|_2 \|\mathbf{s} - \mathbf{u}\|_2^2 + \alpha\|\mathbf{s}\|_2^{L-1}\|\mathbf{s} - \mathbf{u}\|_2 \|\tilde{\xi}(t)\|_2 \\
&\le K_\epsilon \|\ell'(\mathbf{0}, \mathbf{y})\|_2 \|\mathbf{s} - \mathbf{u}\|_2^2 + \alpha\epsilon\tilde{B}_\epsilon^{L-1} C_\epsilon \delta^L,
\end{aligned}$$

where the first inequality follows from eq. (55) and eq. (50). For the second inequality we used eq. (56), and $\|\mathbf{s}(t)\|_2 \le \tilde{B}_\epsilon$, for all $t \in [0, \overline{T}_\delta]$. Integrating the above inequality on both sides from 0 to $t_1 \in [0, \overline{T}_\delta]$, we have

$$\frac{1}{2}\|\mathbf{s}(t_1) - \mathbf{u}(t_1)\|_2^2 \le (\alpha\epsilon\tilde{B}_\epsilon^{L-1}C_\epsilon\delta^L)t_1 + \int_0^{t_1} K_\epsilon\|\ell'(\mathbf{0}, \mathbf{y})\|_2\|\mathbf{s}(t) - \mathbf{u}(t)\|_2^2 dt.$$

Then, using the second Gronwall's inequality in Lemma 8, we have

$$\|\mathbf{s}(\overline{T}_\delta) - \mathbf{u}(\overline{T}_\delta)\|_2^2 \le 2\alpha\epsilon\tilde{B}_\epsilon^{L-1}C_\epsilon\delta^L\overline{T}_\delta e^{2\overline{T}_\delta K_\epsilon\|\ell'(\mathbf{0}, \mathbf{y})\|_2} \le 2\alpha\epsilon\tilde{B}_\epsilon^{L-1}C_\epsilon\delta^L T_\epsilon e^{2T_\epsilon K_\epsilon\|\ell'(\mathbf{0}, \mathbf{y})\|_2},$$

where the second inequality is true since $\overline{T}_\delta \le T_\epsilon$. By definition of $\overline{T}_\delta$, we know $\|\mathbf{s}(\overline{T}_\delta) - \mathbf{u}(\overline{T}_\delta)\|_2 = \epsilon$. Thus, using the above inequality, and since $\delta < \overline{\delta}$, where $\overline{\delta}$ satisfies eq. (49), we have

$$\epsilon^2 = \|\mathbf{s}(\overline{T}_\delta) - \mathbf{u}(\overline{T}_\delta)\|_2^2 \le 2\alpha\epsilon\tilde{B}_\epsilon^{L-1}C_\epsilon\delta^L T_\epsilon e^{2T_\epsilon K_\epsilon\|\ell'(\mathbf{0}, \mathbf{y})\|_2} \le \epsilon^2/2,$$

which leads to a contradiction. Hence, $\overline{T}_\delta > T_\epsilon$.

Since $\|\mathbf{s}(t) - \mathbf{u}(t)\|_2 \le \epsilon$, for all $t \in [0, T_\epsilon]$, we have

$$\|\mathbf{w}(t/\delta^{L-2})\|_2 \le \delta(\|\mathbf{u}(t)\|_2 + \epsilon) \le \tilde{B}_\epsilon\delta, \forall t \in [0, T_\epsilon].$$

We next prove the second part of the theorem. If $\mathbf{u}(t)$ converges to $\mathbf{0}$, then from eq. (43) we know $\|\mathbf{u}(T_\epsilon)\|_2 \le \epsilon$, which implies

$$\left\|\mathbf{w}\left(\frac{T_\epsilon}{\delta^{L-2}}\right)\right\|_2 = \delta\|\mathbf{s}(T_\epsilon)\|_2 \le 2\delta\epsilon. \tag{57}$$

Else, from eq. (44) and eq. (45), we know

$$\frac{\mathbf{u}(T_\epsilon)^\top\mathbf{u}_*}{\|\mathbf{u}(T_\epsilon)\|_2} \ge 1 - \epsilon, \text{ and } \|\mathbf{u}(T_\epsilon)\|_2 \ge \eta. \tag{58}$$

Define $\overline{T}_\epsilon = T_\epsilon/\delta^{L-2}$. Since $\left\|\frac{1}{\delta}\mathbf{w}\left(\frac{T_\epsilon}{\delta^{L-2}}\right) - \mathbf{u}(T_\epsilon)\right\|_2 \le \epsilon$, then for some $\zeta \in \mathbb{R}^k$ we have

$$\frac{\mathbf{w}(\overline{T}_\epsilon)}{\delta} = \mathbf{u}(T_\epsilon) + \zeta, \tag{59}$$

where $\|\zeta\|_2 \le \epsilon$. From $\epsilon \in (0, \eta/2)$ and $\|\mathbf{u}(T_\epsilon)\|_2 \ge \eta$, we have $\|\mathbf{u}(T_\epsilon) + \zeta\|_2 \ge \eta/2$, which implies

$$\left\|\mathbf{w}\left(\overline{T}_\epsilon\right)\right\|_2 \ge \delta\eta/2.$$

Next, from eq. (59) we have

$$\frac{\mathbf{w}(\overline{T}_\epsilon)}{\|\mathbf{w}(\overline{T}_\epsilon)\|_2} = \frac{\mathbf{u}(T_\epsilon) + \zeta}{\|\mathbf{u}(T_\epsilon) + \zeta\|_2}.$$

Hence,

$$\begin{aligned}
\frac{\mathbf{w}(\overline{T}_\epsilon)^\top\mathbf{u}_*}{\|\mathbf{w}(\overline{T}_\epsilon)\|_2} &= \frac{\mathbf{u}(T_\epsilon)^\top\mathbf{u}_* + \zeta^\top\mathbf{u}_*}{\|\mathbf{u}(T_\epsilon) + \zeta\|_2} \\
&= \left(\frac{\mathbf{u}(T_\epsilon)^\top\mathbf{u}_*}{\|\mathbf{u}(T_\epsilon)\|_2}\right)\frac{\|\mathbf{u}(T_\epsilon)\|_2}{\|\mathbf{u}(T_\epsilon) + \zeta\|_2} + \frac{\zeta^\top\mathbf{u}_*}{\|\mathbf{u}(T_\epsilon) + \zeta\|_2} \\
&\ge (1 - \epsilon)\frac{\|\mathbf{u}(T_\epsilon)\|_2}{\|\mathbf{u}(T_\epsilon)\|_2 + \|\zeta\|_2} - \frac{2\epsilon}{\eta} \\
&= (1 - \epsilon)\frac{1}{1 + \frac{\|\zeta\|_2}{\|\mathbf{u}(T_\epsilon)\|_2}} - \frac{2\epsilon}{\eta} \\
&\ge \frac{1 - \epsilon}{1 + \frac{\epsilon}{\eta}} - \frac{2\epsilon}{\eta} \ge (1 - \epsilon)\left(1 - \frac{\epsilon}{\eta}\right) - \frac{2\epsilon}{\eta} \ge 1 - \left(1 + \frac{3}{\eta}\right)\epsilon,
\end{aligned}$$

where in the first inequality we used eq. (58) along with the fact that $\|\zeta\|_2 \le \epsilon$ and $\|\mathbf{u}(T_\epsilon) + \zeta\|_2 \ge \eta/2$. The second inequality follows since $\|\zeta\|_2 \le \epsilon$ and $\|\mathbf{u}(T_\epsilon)\|_2 \ge \eta$. $\qquad\square$

## C  Proofs omitted from Section 5

We first state some key lemmata which are used to prove Theorem 3 and Theorem 4.

### C.1  Key Lemmata

Recall that for any locally Lipschitz continuous function $f : X \to \mathbb{R}$, $\nabla f(\mathbf{x})$ denotes its gradient at $\mathbf{x} \in X$, provided $f$ is differentiable at $\mathbf{x}$, and its Clarke Subdifferential is denoted by $\partial f(\mathbf{x})$, which is defined as

$$\partial f(\mathbf{x}) := \text{conv}\left\{ \lim_{i \to \infty} \nabla f(\mathbf{x}_i) : \lim_{i \to \infty} \mathbf{x}_i = \mathbf{x}, \mathbf{x}_i \in \Omega \right\},$$

where $\Omega$ is any full-measure subset of $X$ such that $f$ is differentiable at each of its points. Note that if $f$ is differentiable everywhere, then $\partial f(\mathbf{x}) = \{\nabla f(\mathbf{x})\}$.

To compute the Clarke subdifferentials of functions which are compositions of non-differentiable functions, we use Clarke's chain rule of differentiation described in the following lemma

**Lemma 15.** *(Clarke, 1983, Theorem 2.3.9 ) Let $h_1, \ldots, h_n : \mathbb{R}^d \to \mathbb{R}$ and $g : \mathbb{R}^n \to \mathbb{R}$ be locally Lipschitz functions, and $f(\mathbf{x}) = g(h_1(\mathbf{x}), \ldots, h_n(\mathbf{x}))$, then,*

$$\partial f(\mathbf{x}) \subseteq conv\left\{ \sum_{i=1}^{n} \alpha_i \zeta_i : \zeta_i \in \partial h_i(\mathbf{x}), \alpha \in \partial g(h_1(\mathbf{x}), \ldots, h_n(\mathbf{x})) \right\}.$$

To prove our results, we will frequently require the gradients of the NCF with respect to weights of each layer, which are derived next.

**Lemma 16.** *For $L \geq 2$, let $\mathcal{H}$ be an $L$-layer feed-forward neural network whose output is as defined in eq. (9). For all $l \in [L-1]$ and $i \in [n]$, define*

$$\phi_i^0 = \mathbf{x}_i, \mathbf{h}_i^1 = \mathbf{W}_1 \phi_i^0, \tag{60}$$

$$\phi_i^l = \sigma(\mathbf{h}_i^l), \mathbf{h}_i^{l+1} = \mathbf{W}_{l+1} \phi_i^l, \tag{61}$$

*and let $\mathbf{A}_i^l = diag\left(\sigma'(\mathbf{h}_i^l)\right)^4$, where $\sigma'(\cdot)$ denotes the Clarke subdifferential of $\sigma(\cdot)$. Also, let*

$$\mathbf{e}_i^l = \mathbf{A}_i^l \mathbf{W}_{l+1}^\top \cdots \mathbf{A}_i^{L-1} \mathbf{W}_L^\top y_i \text{ and } \mathbf{e}_i^L = y_i,$$

*for all $l \in [L-1]$ and $i \in [n]$, then*

$$\partial_{\mathbf{W}_l} \left( \sum_{i=1}^{n} y_i \mathcal{H}(\mathbf{x}_i; \mathbf{W}_1, \cdots \mathbf{W}_L) \right) \subseteq \sum_{i=1}^{n} \mathbf{e}_i^l (\phi_i^{l-1})^\top, \text{ for all } l \in [L].$$

*Proof.* We begin by noting that, for $l \in [L]$,

$$\partial_{\mathbf{W}_l} \left( \sum_{i=1}^{n} y_i \mathcal{H}(\mathbf{x}_i; \mathbf{W}_1, \cdots \mathbf{W}_L) \right) \subseteq \sum_{i=1}^{n} \partial_{\mathbf{W}_l} \left( y_i \mathcal{H}(\mathbf{x}_i; \mathbf{W}_1, \cdots \mathbf{W}_L) \right).$$

Thus, we need to prove

$$\partial_{\mathbf{W}_l} \left( y_i \mathcal{H}(\mathbf{x}_i; \mathbf{W}_1, \cdots \mathbf{W}_L) \right) \subseteq \mathbf{e}_i^l (\phi_i^{l-1})^\top, \text{ for all } l \in [L].$$

For $l = L$, the above equation is true since

$$\partial_{\mathbf{W}_L} \left( y_i \mathcal{H}(\mathbf{x}_i; \mathbf{W}_1, \cdots \mathbf{W}_L) \right) = \partial_{\mathbf{W}_L} \left( y_i \mathbf{W}_L \phi_i^{L-1} \right) = y_i \left( \phi_i^{L-1} \right)^\top.$$

For $l \in [L-1]$, we have

$$\partial_{\mathbf{W}_l} \left( y_i \mathcal{H}(\mathbf{x}_i; \mathbf{W}_1, \cdots \mathbf{W}_L) \right) = \partial_{\mathbf{W}_l} \left( y_i \mathbf{W}_L \sigma(\mathbf{W}_{L-1} \cdots \sigma(\mathbf{W}_l \phi_i^{l-1}) \cdots) \right) \subseteq \mathbf{e}_i^l (\phi_i^{l-1})^\top,$$

where the last inclusion follows from repeatedly applying the chain rule. This completes the proof. $\qquad\square$

---

[4]For non-differentiable activation functions such as ReLU, $\sigma'$ could be an interval, implying that $\mathbf{A}_i^l$ is a set-valued mapping.

We next restate and prove Lemma 5.

**Lemma 17.** *Let $\mathcal{H}$ be an $L$-layer feed-forward neural network whose output is as defined in eq. (9), where $L \geq 2$. Suppose $(\overline{\mathbf{W}}_1, \cdots, \overline{\mathbf{W}}_L)$ is a non-zero KKT points of the constrained NCF in eq. (10). If $\sigma(x) = \max(\alpha x, x)^p$, for some $p \in \mathbb{N}$ and $\alpha \in \mathbb{R}$, then*

$$diag\left(\overline{\mathbf{W}}_l \overline{\mathbf{W}}_l^\top\right) = p \cdot diag\left(\overline{\mathbf{W}}_{l+1}^\top \overline{\mathbf{W}}_{l+1}\right), \text{ for all } l \in [L-1], \tag{62}$$

*and, if $\sigma(x) = x$, then*

$$\overline{\mathbf{W}}_l \overline{\mathbf{W}}_l^\top = \overline{\mathbf{W}}_{l+1}^\top \overline{\mathbf{W}}_{l+1}, \text{ for all } l \in [L-1]. \tag{63}$$

*Furthermore, if $(\overline{\mathbf{W}}_1, \cdots, \overline{\mathbf{W}}_{L-1}, \overline{\mathbf{W}}_L) = (\mathbf{a}_1 \mathbf{b}_1^\top, \cdots, \mathbf{a}_{L-1} \mathbf{b}_{L-1}^\top, \overline{\mathbf{w}}^\top)$, where $\|\mathbf{a}_l\|_2 = \|\mathbf{b}_l\|_2$, for all $l \in [L-1]$, then $|\mathbf{a}_l| = p^{1/4}|\mathbf{b}_{l+1}|$, for all $l \in [L-2]$, $|\mathbf{a}_{L-1}| = (p\hat{p})^{1/4}|\overline{\mathbf{w}}|$, $\|\mathbf{a}_l\|_2^4 = p^{L-l}/\hat{p}$, for all $l \in [L-1]$, and $\|\overline{\mathbf{w}}\|_2^2 = 1/\hat{p}$, where $\hat{p} = p^{L-1} + p^{L-2} + \cdots + 1$.*

*Proof.* By KKT condition, for $l \in [L-1]$, we have

$$\mathbf{0} \in \partial_{\mathbf{W}_l} \left( \sum_{i=1}^n y_i \mathcal{H}(\mathbf{x}_i; \overline{\mathbf{W}}_1, \cdots \overline{\mathbf{W}}_L) \right) + \lambda \overline{\mathbf{W}}_l \tag{64}$$

Multiplying the above equation by $\overline{\mathbf{W}}_l^\top$ from the right, and using Lemma 16, we have

$$\mathbf{0} \in \sum_{i=1}^n \mathbf{e}_i^l (\phi_i^{l-1})^\top \overline{\mathbf{W}}_l^\top + \lambda \overline{\mathbf{W}}_l \overline{\mathbf{W}}_l^\top \tag{65}$$

Since $\overline{\mathbf{W}}_l \phi_i^{l-1} = \mathbf{h}_i^l$, we have

$$\mathbf{0} \in \sum_{i=1}^n \mathbf{e}_i^l (\mathbf{h}_i^l)^\top + \lambda \overline{\mathbf{W}}_l \overline{\mathbf{W}}_l^\top. \tag{66}$$

Since $\mathbf{e}_i^l = \mathbf{A}_i^l \overline{\mathbf{W}}_{l+1}^\top \cdots \mathbf{A}_i^{L-1} \overline{\mathbf{W}}_L^\top y_i$, if we let $\mathbf{q}_i = \overline{\mathbf{W}}_{l+1}^\top \cdots \mathbf{A}_i^{L-1} \overline{\mathbf{W}}_L^\top y_i$, then we have

$$\mathbf{0} \in \sum_{i=1}^n \mathbf{A}_i^l \mathbf{q}_i (\mathbf{h}_i^l)^\top + \lambda \overline{\mathbf{W}}_l \overline{\mathbf{W}}_l^\top. \tag{67}$$

Next, similarly using the KKT condition for $\mathbf{W}_{l+1}$, we have

$$\mathbf{0} \in \overline{\mathbf{W}}_{l+1}^\top \sum_{i=1}^n \mathbf{e}_i^{l+1} (\phi_i^l)^\top + \lambda \overline{\mathbf{W}}_{l+1}^\top \overline{\mathbf{W}}_{l+1}. \tag{68}$$

Expanding $\mathbf{e}_i^{l+1}$, we have

$$\mathbf{0} \in \sum_{i=1}^n \overline{\mathbf{W}}_{l+1}^\top \mathbf{A}_i^{l+1} \overline{\mathbf{W}}_{l+2}^\top \cdots \mathbf{A}_i^{L-1} \overline{\mathbf{W}}_L^\top \mathbf{y}_i^\top (\phi_i^l)^\top + \lambda \overline{\mathbf{W}}_{l+1}^\top \overline{\mathbf{W}}_{l+1}, \tag{69}$$

which implies

$$\mathbf{0} \in \sum_{i=1}^n \mathbf{q}_i (\phi_i^l)^\top + \lambda \overline{\mathbf{W}}_{l+1}^\top \overline{\mathbf{W}}_{l+1}. \tag{70}$$

We next show that

$$diag(\mathbf{A}_i^l \mathbf{q}_i (\mathbf{h}_i^l)^\top) = p \, diag(\mathbf{q}_i (\phi_i^l)^\top), \forall i \in [n] \text{ and } l \in [L-1],$$

which will prove the first part of the theorem. To show this, first note that since $\mathbf{A}_i^l$ is a diagonal matrix, and $\mathbf{q}_i, \mathbf{h}_i^l$ are vectors, we have

$$
\begin{aligned}
\text{diag}(\mathbf{A}_i^l \mathbf{q}_i (\mathbf{h}_i^l)^\top) = \mathbf{A}_i^l \text{diag}(\mathbf{q}_i)\text{diag}(\mathbf{h}_i^l) &= \text{diag}(\mathbf{q}_i)\mathbf{A}_i^l \text{diag}(\mathbf{h}_i^l) \\
&= \text{diag}(\mathbf{q}_i)\text{diag}(\sigma'(\mathbf{h}_i^l))\text{diag}(\mathbf{h}_i^l) \\
&= p\,\text{diag}(\mathbf{q}_i)\text{diag}(\sigma(\mathbf{h}_i^l)) \\
&= p\,\text{diag}(\mathbf{q}_i)\text{diag}(\phi_i^l) = p\,\text{diag}(\mathbf{q}_i(\phi_i^l)^\top),
\end{aligned}
$$

where in the third equality, we used the definition of $\mathbf{A}_i^l$ and the fourth equality follows from the fact $\sigma'(x)x = p\sigma(x)$.

For $\sigma(x) = x$, note that $\mathbf{A}_i^l$ will be identity and $\phi_i^l = \mathbf{h}_i^l$, for all $i \in [n]$ and $l \in [L-1]$. Hence,

$$
\mathbf{A}_i^l \mathbf{q}_i (\mathbf{h}_i^l)^\top = \mathbf{q}_i (\phi_i^l)^\top, \forall i \in [n] \text{ and } l \in [L-1],
$$

which proves eq. (63).

We next prove the second part. From eq. (62), we have

$$
\|\overline{\mathbf{W}}_1\|_F^2 = p\|\overline{\mathbf{W}}_2\|_F^2 = \cdots = p^{L-1}\|\overline{\mathbf{W}}_L\|_F^2.
$$

Since

$$
\|\overline{\mathbf{W}}_1\|_F^2 + \cdots \|\overline{\mathbf{W}}_L\|_F^2 = 1,
$$

if we define $\hat{p} = p^{L-1} + p^{L-2} + \cdots + 1$, then,

$$
1 = \|\overline{\mathbf{W}}_L\|_F^2 (p^{L-1} + p^{L-2} + \cdots + 1),
$$

which implies $\|\overline{\mathbf{W}}_l\|_F^2 = p^{L-l}/\hat{p}$, for all $l \in [L]$. Next, since $\|\mathbf{a}_l\|_2 = \|\mathbf{b}_l\|_2$, we further get $p^{L-l}/\hat{p} = \|\mathbf{a}_l\|_2^4$, for all $l \in [L-1]$ and $\|\overline{\mathbf{w}}\|_2^2 = 1/\hat{p}$.

Now, using eq. (62), for all $l \in [L-2]$, we have

$$
\text{diag}\left(\|\mathbf{b}_l\|_2^2 \mathbf{a}_l \mathbf{a}_l^\top\right) = p \cdot \text{diag}\left(\|\mathbf{a}_{l+1}\|_2^2 \mathbf{b}_{l+1}\mathbf{b}_{l+1}^\top\right), \text{ and } \text{diag}\left(\|\mathbf{b}_{L-1}\|_2^2 \mathbf{a}_{L-1}\mathbf{a}_{L-1}^\top\right) = p \cdot \text{diag}\left(\overline{\mathbf{w}}\overline{\mathbf{w}}^\top\right)
$$

which implies

$$
|\mathbf{a}_l| = p^{1/4}|\mathbf{b}_{l+1}| \text{ and } |\mathbf{a}_{L-1}| = (p\hat{p})^{1/4}|\overline{\mathbf{w}}|, \tag{71}
$$

where the last equality uses $p^{L-l}/\hat{p} = \|\mathbf{b}_{L-1}\|_2^4$. Hence, the proof is complete. $\qquad\square$

## C.2  Proof of Theorem 3

From Lemma 5, we know $\|\overline{\mathbf{w}}\|_2^2 = 1/L$ and

$$
\|\mathbf{a}_l\|_2^2 = \|\mathbf{b}_l\|_2^2 = \frac{1}{\sqrt{L}}, \text{ for all } l \in [L-1]. \tag{72}
$$

For the remaining proof, we handle each of the cases individually, and begin by $\alpha = 1$.
**Case 1** $(\alpha = 1)$ : We begin by proving the "only if" part. From Lemma 5, we have

$$
\overline{\mathbf{W}}_l \overline{\mathbf{W}}_l^\top = \overline{\mathbf{W}}_{l+1}^\top \overline{\mathbf{W}}_{l+1}, \forall l \in [L-1], \tag{73}
$$

which implies, for $l \in [L-2]$,

$$
\|\mathbf{b}_l\|_2^2 \mathbf{a}_l \mathbf{a}_l^\top = \|\mathbf{a}_{l+1}\|_2^2 \mathbf{b}_{l+1}\mathbf{b}_{l+1}^\top, \text{ and } \|\mathbf{b}_{L-1}\|_2^2 \mathbf{a}_{L-1}\mathbf{a}_{L-1}^\top = \overline{\mathbf{w}}\overline{\mathbf{w}}^\top.
$$

Combining the above equality with eq. (72) we get

$$q_l \mathbf{a}_l = \mathbf{b}_{l+1}, \forall l \in [L-2], \text{ and } q_{L-1}\mathbf{a}_{L-1} = L^{1/4}\overline{\mathbf{w}},$$

where $q_l \in \{-1, 1\}$. Hence,

$$\overline{\mathbf{W}}_1 = \mathbf{a}_1 \mathbf{b}_1^\top \text{ and } \overline{\mathbf{W}}_l = q_{l-1}\mathbf{a}_l\mathbf{a}_{l-1}^\top, \text{ if } 2 \le l \le L-1.$$

Now, since $(\overline{\mathbf{W}}_1, \cdots, \overline{\mathbf{W}}_L)$ is a non-zero KKT point of eq. (10), using Lemma 11, we have

$$\mathbf{0} = \lambda \mathbf{a}_1 \mathbf{b}_1^\top + \nabla_{\mathbf{W}_1}\left(\mathbf{y}^\top \mathcal{H}(\mathbf{X}; \overline{\mathbf{W}}_1, \cdots, \overline{\mathbf{W}}_L)\right), \tag{74}$$

where $\lambda \neq 0$. Let $q_{1:L-1} := q_1 q_2 \cdots q_{L-1}$. From Lemma 16, we have

$$\nabla_{\mathbf{W}_1}\left(\mathbf{y}^\top \mathcal{H}(\mathbf{x}_i; \overline{\mathbf{W}}_1, \cdots, \overline{\mathbf{W}}_L)\right) = \sum_{i=1}^n \overline{\mathbf{W}}_2^\top \cdots \overline{\mathbf{W}}_L^\top y_i \mathbf{x}_i^\top$$

$$= q_{1:L-1} \sum_{i=1}^n \mathbf{a}_1 \mathbf{a}_2^\top \mathbf{a}_2 \mathbf{a}_3^\top \cdots \mathbf{a}_{L-2}\mathbf{a}_{L-1}^\top \mathbf{a}_{L-1} y_i \mathbf{x}_i^\top / L^{1/4}$$

$$= q_{1:L-1}\left(\frac{1}{\sqrt{L}}\right)^{L-2} \frac{1}{L^{1/4}}\sum_{i=1}^n \mathbf{a}_1 y_i \mathbf{x}_i^\top. \tag{75}$$

From eq. (74) and eq. (75), we have

$$\mathbf{0} = \lambda \mathbf{b}_1^\top + q_{1:L-1}\left(\frac{1}{\sqrt{L}}\right)^{L-2}\frac{1}{L^{1/4}}\sum_{i=1}^n y_i \mathbf{x}_i^\top. \tag{76}$$

Now, $\mathbf{u}_*$ is a KKT point of

$$\max_{\mathbf{u}}\left(\sum_{i=1}^n y_i \mathbf{x}_i^\top\right)\mathbf{u}, \text{ such that } \|\mathbf{u}\|_2^2 = 1/\sqrt{L}, \tag{77}$$

if there exists $\lambda^*$ such that

$$\sum_{i=1}^n y_i \mathbf{x}_i + \lambda^* \mathbf{u}_* = \mathbf{0}, \text{ and } \|\mathbf{u}_*\|_2^2 = 1/\sqrt{L}. \tag{78}$$

Hence, from eq. (76), and since $\|\mathbf{b}_1\|_2^2 = 1/\sqrt{L}$, we observe that $\mathbf{b}_1$ satisfies the KKT conditions in eq. (78). We now turn towards proving the "if" part.

Note that $(\overline{\mathbf{W}}_1, \cdots, \overline{\mathbf{W}}_L) = (\mathbf{a}_1 \mathbf{b}_1^\top, q_1\mathbf{a}_2\mathbf{a}_1^\top, \cdots, q_{L-2}\mathbf{a}_{L-1}\mathbf{a}_{L-2}^\top, q_{L-1}\mathbf{a}_{L-1}^\top/L^{1/4})$. Now, since $\mathbf{b}_1$ is a non-zero KKT point of eq. (12), from Lemma 11, there exists $\lambda^* \neq 0$ such that

$$\sum_{i=1}^n y_i \mathbf{x}_i + \lambda^* \mathbf{b}_1 = \mathbf{0}, \tag{79}$$

We will complete the proof by showing that for $\lambda = q_{1:L-1}\lambda^*\left(1/\sqrt{L}\right)^{L-2}/L^{1/4}$,

$$\lambda \overline{\mathbf{W}}_l + \nabla_{\mathbf{W}_l}\left(\mathbf{y}^\top \mathcal{H}(\mathbf{X}; \overline{\mathbf{W}}_1, \cdots, \overline{\mathbf{W}}_L)\right) = \mathbf{0}, \forall l \in [L]. \tag{80}$$

For $l = 1$, we have

$$\lambda \overline{\mathbf{W}}_1 + \nabla_{\mathbf{W}_1} \left( \mathbf{y}^\top \mathcal{H}(\mathbf{X}; \overline{\mathbf{W}}_1, \cdots, \overline{\mathbf{W}}_L) \right)$$

$$= \lambda \mathbf{a}_1 \mathbf{b}_1^\top + \sum_{i=1}^n \overline{\mathbf{W}}_2^\top \cdots \overline{\mathbf{W}}_L^\top y_i \mathbf{x}_i^\top$$

$$= \lambda \mathbf{a}_1 \mathbf{b}_1^\top + q_{1:L-1} \sum_{i=1}^n \mathbf{a}_1 \mathbf{a}_2^\top \mathbf{a}_2 \mathbf{a}_3^\top \cdots \mathbf{a}_{L-2} \mathbf{a}_{L-1}^\top \mathbf{a}_{L-1} y_i \mathbf{x}_i^\top / L^{1/4}$$

$$= \lambda \mathbf{a}_1 \mathbf{b}_1^\top + q_{1:L-1} \left( \frac{1}{\sqrt{L}} \right)^{L-2} \frac{1}{L^{1/4}} \sum_{i=1}^n \mathbf{a}_1 y_i \mathbf{x}_i^\top$$

$$= \lambda \mathbf{a}_1 \mathbf{b}_1^\top - q_{1:L-1} \lambda^* \left( \frac{1}{\sqrt{L}} \right)^{L-2} \frac{1}{L^{1/4}} \mathbf{a}_1 \mathbf{b}_1^\top = \mathbf{0},$$

where the penultimate equality follows from eq. (79). For $2 \le l \le L - 1$, we have

$$\lambda \overline{\mathbf{W}}_l + \nabla_{\mathbf{W}_l} \left( \mathbf{y}^\top \mathcal{H}(\mathbf{X}; \overline{\mathbf{W}}_1, \cdots, \overline{\mathbf{W}}_L) \right)$$

$$= \lambda q_{l-1} \mathbf{a}_l \mathbf{a}_{l-1}^\top + \sum_{i=1}^n \overline{\mathbf{W}}_{l+1}^\top \cdots \overline{\mathbf{W}}_L^\top y_i \mathbf{x}_i^\top \overline{\mathbf{W}}_1^\top \cdots \overline{\mathbf{W}}_{l-1}^\top$$

$$= \lambda q_{l-1} \mathbf{a}_l \mathbf{a}_{l-1}^\top + \frac{q_{1:l-2} q_{l:L-1}}{L^{1/4}} \sum_{i=1}^n \mathbf{a}_l \mathbf{a}_{l+1}^\top \mathbf{a}_{l+1} \mathbf{a}_{l+2}^\top \cdots \mathbf{a}_{L-2} \mathbf{a}_{L-1}^\top \mathbf{a}_{L-1} y_i \mathbf{x}_i^\top \mathbf{b}_1 \mathbf{a}_1^\top \mathbf{a}_1 \mathbf{a}_2^\top \cdots \mathbf{a}_{l-2} \mathbf{a}_{l-1}^\top$$

$$= \lambda q_{l-1} \mathbf{a}_l \mathbf{a}_{l-1}^\top + \frac{q_{1:l-2} q_{l:L-1}}{L^{1/4}} \left( \frac{1}{\sqrt{L}} \right)^{L-3} \mathbf{a}_l \mathbf{a}_{l-1}^\top \sum_{i=1}^n y_i \mathbf{x}_i^\top \mathbf{b}_1.$$

$$= \lambda q_{l-1} \mathbf{a}_l \mathbf{a}_{l-1}^\top - \frac{q_{1:l-2} q_{l:L-1}}{L^{1/4}} \lambda^* \left( \frac{1}{\sqrt{L}} \right)^{L-2} \mathbf{a}_l \mathbf{a}_{l-1}^\top = \mathbf{0},$$

where the penultimate equality follows from eq. (79) and since $\|\mathbf{b}_1\|_2^2 = 1/\sqrt{L}$. Finally, for $l = L$, we have

$$\lambda \overline{\mathbf{W}}_L + \nabla_{\mathbf{W}_L} \left( \mathbf{y}^\top \mathcal{H}(\mathbf{X}; \overline{\mathbf{W}}_1, \cdots, \overline{\mathbf{W}}_L) \right)$$

$$= \lambda q_{L-1} \mathbf{a}_{L-1}^\top / L^{1/4} + \sum_{i=1}^n y_i \mathbf{x}_i^\top \overline{\mathbf{W}}_1^\top \cdots \overline{\mathbf{W}}_{L-1}^\top$$

$$= \lambda q_{L-1} \mathbf{a}_{L-1}^\top / L^{1/4} + q_{1:L-2} \sum_{i=1}^n y_i \mathbf{x}_i^\top \mathbf{b}_1 \mathbf{a}_1^\top \mathbf{a}_1 \mathbf{a}_2^\top \cdots \mathbf{a}_{L-3} \mathbf{a}_{L-2}^\top \mathbf{a}_{L-2} \mathbf{a}_{L-1}^\top$$

$$= \lambda q_{L-1} \mathbf{a}_{L-1}^\top / L^{1/4} + q_{1:L-2} \left( \frac{1}{\sqrt{L}} \right)^{L-2} \sum_{i=1}^n y_i \mathbf{x}_i^\top \mathbf{b}_1 \mathbf{a}_{L-1}^\top$$

$$= \lambda q_{L-1} \mathbf{a}_{L-1}^\top / L^{1/4} - q_{1:L-2} \lambda^* \left( \frac{1}{\sqrt{L}} \right)^{L-1} \mathbf{a}_{L-1}^\top = \mathbf{0},$$

where the penultimate equality follows from eq. (79).

**Case 2** ($\alpha \ne 1$) : We begin by proving the "only if" part. Since $\sigma(\cdot)$ is $1-$positively homogeneous, we have

$$\sigma(c\mathbf{p}) = c\sigma(\mathbf{p}), \text{ and } \sigma(d\mathbf{q}) = \mathbf{q}\sigma(d), \tag{81}$$

where $c > 0$, $\mathbf{p}$ is a vector, and $d \in \mathbb{R}$, $\mathbf{q}$ is a vector with non-negative entries.

Now, from Lemma 5, for $l \in [L-2]$, we have

$$|\mathbf{a}_l| = |\mathbf{b}_{l+1}|, \text{ and } |\mathbf{a}_{L-1}| = L^{1/4} |\overline{\mathbf{w}}|. \tag{82}$$

Now, since $\{\mathbf{a}_l\}_{l=1}^{L-1}, \{\mathbf{b}_l\}_{l=2}^{L-1}$ are non-negative, $\mathbf{a}_{L-1} = L^{1/4}|\overline{\mathbf{w}}|$, and

$$\mathbf{a}_l = \mathbf{b}_{l+1}, \forall l \in [L-2].$$

We next show that $q\mathbf{a}_{L-1} = L^{1/4}\overline{\mathbf{w}}$, for some $q \in \{-1, 1\}$. Since $(\overline{\mathbf{W}}_1, \cdots, \overline{\mathbf{W}}_L)$ is a KKT point of eq. (10), from Lemma 16, we have

$$\mathbf{0} \in \lambda\overline{\mathbf{W}}_L + \partial_{\mathbf{W}_L}\left(\mathbf{y}^\top\mathcal{H}(\mathbf{X}; \overline{\mathbf{W}}_1, \cdots, \overline{\mathbf{W}}_L)\right) = \lambda\overline{\mathbf{w}}^\top + \sum_{i=1}^n y_i\left(\phi_i^{L-1}\right)^\top, \tag{83}$$

where $\phi_i^{L-1} = \sigma(\overline{\mathbf{W}}_{L-1}\cdots\sigma(\overline{\mathbf{W}}_1\mathbf{x}_i)\cdots)$. Since $\{\mathbf{a}_l\}_{l=1}^{L-1}$ are non-negative, using eq. (81), we have

$$\phi_i^{L-1} = \sigma(\overline{\mathbf{W}}_{L-1}\cdots\sigma(\overline{\mathbf{W}}_1\mathbf{x}_i)\cdots) = \sigma(\mathbf{a}_{L-1}\mathbf{a}_{L-2}^\top\sigma(\mathbf{a}_{L-2}\mathbf{a}_{L-3}^\top\cdots\sigma(\mathbf{a}_2\mathbf{a}_1^\top\sigma(\mathbf{a}_1\mathbf{b}_1^\top\mathbf{x}_i))\cdots)$$
$$= \mathbf{a}_{L-1}\|\mathbf{a}_{L-2}\|_2^2\|\mathbf{a}_{L-3}\|_2^2\cdots\|\mathbf{a}_1\|_2^2\sigma^{L-1}(\mathbf{b}_1^\top\mathbf{x}_i).$$

Since $\|\mathbf{a}_l\|_2^2 = 1/\sqrt{L}$,

$$\mathbf{0} = \lambda\overline{\mathbf{w}}^\top + \left(\frac{1}{\sqrt{L}}\right)^{L-2}\left(\sum_{i=1}^n \sigma^{L-1}(\mathbf{b}_1^\top\mathbf{x}_i)y_i\right)\mathbf{a}_{L-1}^\top.$$

From the above equality it is clear that $\overline{\mathbf{w}}$ is parallel to $\mathbf{a}_{L-1}$. However, since $\mathbf{a}_{L-1} = L^{1/4}|\overline{\mathbf{w}}|$, we get $q\mathbf{a}_{L-1} = L^{1/4}\overline{\mathbf{w}}$, for some $q \in \{-1, 1\}$.

From Lemma 16, we also have

$$\mathbf{0} \in \lambda\overline{\mathbf{W}}_1 + \partial_{\mathbf{W}_1}\left(\mathbf{y}^\top\mathcal{H}(\mathbf{X}; \overline{\mathbf{W}}_1, \cdots, \overline{\mathbf{W}}_L)\right). \tag{84}$$

Without loss of generality, we assume $\mathbf{a}_{11} > 0$, where $\mathbf{a}_{11}$ denotes the first entry of $\mathbf{a}_1$. Let $\mathbf{w}_{11}$ denote the first row of $\mathbf{W}$. Then from the above equation we have

$$\mathbf{0} \in \lambda\mathbf{a}_{11}\mathbf{b}_1^\top + \partial_{\mathbf{w}_{11}}\left(\mathbf{y}^\top\mathcal{H}(\mathbf{X}; \overline{\mathbf{W}}_1, \cdots, \overline{\mathbf{W}}_L)\right). \tag{85}$$

Our goal is to simplify $\partial_{\mathbf{w}_{11}}\left(\mathbf{y}^\top\mathcal{H}(\mathbf{X}; \overline{\mathbf{W}}_1, \cdots, \overline{\mathbf{W}}_L)\right).$ Note that, for any $\mathbf{W}_1$, we have

$$\mathcal{H}(\mathbf{x}_i; \mathbf{W}_1, \overline{\mathbf{W}}_2, \cdots, \overline{\mathbf{W}}_L) = q\mathbf{a}_{L-1}^\top\sigma(\mathbf{a}_{L-1}\mathbf{a}_{L-2}^\top\sigma(\mathbf{a}_{L-2}\mathbf{a}_{L-3}^\top\cdots\sigma(\mathbf{a}_2\mathbf{a}_1^\top\sigma(\mathbf{W}_1\mathbf{x}_i))\cdots)))/L^{1/4}$$
$$= q\|\mathbf{a}_{L-1}\|_2^2\cdots\|\mathbf{a}_2\|_2^2\sigma^{L-2}(\mathbf{a}_1^\top\sigma(\mathbf{W}_1\mathbf{x}_i))/L^{1/4}$$
$$= q\left(\frac{1}{\sqrt{L}}\right)^{L-2}\sigma^{L-2}(\mathbf{a}_1^\top\sigma(\mathbf{W}_1\mathbf{x}_i))/L^{1/4}.$$

Next, choose $\mathbf{W}_1$ such that its equal to $\overline{\mathbf{W}}_1$ everywhere except for the first row, and the first row is assumed to be equal to $\mathbf{w}$. Thus, we have

$$g(\mathbf{w}) =: \mathbf{y}^\top\mathcal{H}(\mathbf{X}; \mathbf{W}_1, \overline{\mathbf{W}}_2, \cdots, \overline{\mathbf{W}}_L)$$
$$= q\left(\frac{1}{\sqrt{L}}\right)^{L-2}\frac{1}{L^{1/4}}\sum_{i=1}^n y_i\sigma^{L-2}\left(\mathbf{a}_{11}\sigma(\mathbf{w}^\top\mathbf{x}_i) + \sum_{j=2}^{k_1}\mathbf{a}_{1j}\sigma(\mathbf{a}_{1j}\mathbf{b}_1^\top\mathbf{x}_i)\right).$$

Now, it is easy to see that

$$\partial_{\mathbf{w}}g(\mathbf{a}_{11}\mathbf{b}_1) = \partial_{\mathbf{w}_{11}}\left(\mathbf{y}^\top\mathcal{H}(\mathbf{X}; \overline{\mathbf{W}}_1, \cdots, \overline{\mathbf{W}}_L)\right). \tag{86}$$

To compute $\partial_{\mathbf{w}}g(\mathbf{a}_{11}\mathbf{b}_1)$, we first characterize $g(\mathbf{w})$ near $\mathbf{a}_{11}\mathbf{b}_1$. If for some $i \in [n]$, $\mathbf{b}_1^\top\mathbf{x}_i = 0$, then

$$\sigma^{L-2}\left(\mathbf{a}_{11}\sigma(\mathbf{w}^\top\mathbf{x}_i) + \sum_{j=2}^{k_1}\mathbf{a}_{1j}\sigma(\mathbf{a}_{1j}\mathbf{b}_1^\top\mathbf{x}_i)\right) = \sigma^{L-1}(\mathbf{a}_{11}\mathbf{w}^\top\mathbf{x}_i) = \mathbf{a}_{11}\sigma^{L-1}(\mathbf{w}^\top\mathbf{x}_i).$$

Next, if for some $i \in [n]$, $\mathbf{b}_1^\top \mathbf{x}_i > 0$, then $\mathbf{w}^\top \mathbf{x}_i > 0$, if $\mathbf{w}$ is in a sufficiently small neighborhood of $\mathbf{a}_{11}\mathbf{b}_1$. Further, since $\mathbf{a}_{11} > 0$ and $\mathbf{a}_{1j}\sigma(\mathbf{a}_{1j}\mathbf{b}_1^\top \mathbf{x}_i) \geq 0$, for all $j \geq 2$, we have

$$\sigma^{L-2}\left(\mathbf{a}_{11}\sigma(\mathbf{w}^\top \mathbf{x}_i) + \sum_{j=2}^{k_1} \mathbf{a}_{1j}\sigma(\mathbf{a}_{1j}\mathbf{b}_1^\top \mathbf{x}_i)\right) = \sigma^{L-1}(\mathbf{a}_{11}\mathbf{w}^\top \mathbf{x}_i) + \sigma^{L-2}(\sum_{j=2}^{k_1} \mathbf{a}_{1j}\sigma(\mathbf{a}_{1j}\mathbf{b}_1^\top \mathbf{x}_i))$$

$$= \mathbf{a}_{11}\sigma^{L-1}(\mathbf{w}^\top \mathbf{x}_i) + \sigma^{L-2}(\sum_{j=2}^{k_1} \mathbf{a}_{1j}\sigma(\mathbf{a}_{1j}\mathbf{b}_1^\top \mathbf{x}_i)).$$

The above equation can also be shown to be true if $\mathbf{b}_1^\top \mathbf{x}_i < 0$ and $\mathbf{w}$ is in a sufficiently small neighborhood of $\mathbf{a}_{11}\mathbf{b}_1$. Therefore, if $\mathbf{w}$ is in a sufficiently small neighborhood of $\mathbf{a}_{11}\mathbf{b}_1$, we have

$$\partial_\mathbf{w} g(\mathbf{w}) = q\left(\frac{1}{\sqrt{L}}\right)^{L-2} \frac{1}{L^{1/4}} \partial_\mathbf{w}\left(\mathbf{a}_{11}\sum_{i=1}^n y_i\sigma^{L-1}(\mathbf{w}^\top \mathbf{x}_i)\right) = q\mathbf{a}_{11}\left(\frac{1}{\sqrt{L}}\right)^{L-2} \frac{1}{L^{1/4}} \partial_\mathbf{w}\left(\sum_{i=1}^n y_i\sigma^{L-1}(\mathbf{w}^\top \mathbf{x}_i)\right).$$

Since $\mathbf{a}_{11} > 0$ and $\sum_{i=1}^n y_i\sigma^{L-1}(\mathbf{w}^\top \mathbf{x}_i)$ is $1-$positively homogeneous in $\mathbf{w}$, using Lemma 7, we get

$$\partial_\mathbf{w} g(\mathbf{a}_{11}\mathbf{b}_1) = q\mathbf{a}_{11}\left(\frac{1}{\sqrt{L}}\right)^{L-2} \frac{1}{L^{1/4}} \partial_\mathbf{w}\left(\sum_{i=1}^n y_i\sigma^{L-1}(\mathbf{a}_{11}\mathbf{b}_1^\top \mathbf{x}_i)\right)$$

$$= q\mathbf{a}_{11}\left(\frac{1}{\sqrt{L}}\right)^{L-2} \frac{1}{L^{1/4}} \partial_\mathbf{w}\left(\sum_{i=1}^n y_i\sigma^{L-1}(\mathbf{b}_1^\top \mathbf{x}_i)\right). \tag{87}$$

From eq. (86), eq. (85) and the above equation, we have

$$\mathbf{0} \in \lambda\mathbf{b}_1 + q\left(\frac{1}{\sqrt{L}}\right)^{L-2} \frac{1}{L^{1/4}} \partial_\mathbf{w}\left(\sum_{i=1}^n y_i\sigma^{L-1}(\mathbf{b}_1^\top \mathbf{x}_i)\right). \tag{88}$$

Now, using Lemma 11, $\mathbf{u}_*$ is a KKT point of

$$\max_\mathbf{u} \sum_{i=1}^n y_i\sigma^{L-1}(\mathbf{x}_i^\top \mathbf{u}), \text{ such that } \|\mathbf{u}\|_2^2 = 1/\sqrt{L}, \tag{89}$$

if there exists $\lambda^*$ such that

$$\mathbf{0} \in \partial_\mathbf{u}\left(\sum_{i=1}^n y_i\sigma^{L-1}(\mathbf{x}_i^\top \mathbf{u}_*)\right) + \lambda^*\mathbf{u}_*, \text{ and } \|\mathbf{u}_*\|_2^2 = 1/\sqrt{L}. \tag{90}$$

Hence, from eq. (88), and since $\|\mathbf{b}_1\|_2^2 = 1/\sqrt{L}$, we get that $\mathbf{b}_1$ satisfies the KKT conditions in eq. (90).

We now turn towards proving the "if" condition.

Note that $(\overline{\mathbf{W}}_1, \cdots, \overline{\mathbf{W}}_L) = (\mathbf{a}_1\mathbf{b}_1^\top, \mathbf{a}_2\mathbf{a}_1^\top, \mathbf{a}_3\mathbf{a}_2^\top, \cdots, \mathbf{a}_{L-1}\mathbf{a}_{L-2}^\top, q\mathbf{a}_{L-1}^\top/L^{1/4})$. Since $\mathbf{b}_1$ is a non-zero KKT point of eq. (11), there exists a non-zero $\lambda^*$ such that

$$\mathbf{0} \in \partial_\mathbf{u}\left(\sum_{i=1}^n y_i\sigma^{L-1}(\mathbf{x}_i^\top \mathbf{b}_1)\right) + \lambda^*\mathbf{b}_1. \tag{91}$$

Multiplying the above inclusion by $\mathbf{b}_1^\top$ and using $\|\mathbf{b}_1\|_2^2 = 1/\sqrt{L}$, we get $\lambda^* = -\sqrt{L}\sum_{i=1}^n y_i\sigma^{L-1}(\mathbf{x}_i^\top \mathbf{b}_1)$. We will complete the proof by showing that for $\lambda = q\lambda^*\left(1/\sqrt{L}\right)^{L-2}/L^{1/4}$,

$$\mathbf{0} \in \lambda\overline{\mathbf{W}}_l + \partial_{\mathbf{W}_l}\left(\mathbf{y}^\top \mathcal{H}(\mathbf{X}; \overline{\mathbf{W}}_1, \cdots, \overline{\mathbf{W}}_L)\right), \forall l \in [L]. \tag{92}$$

Let $\phi_i^l = \sigma(\overline{\mathbf{W}}_l\sigma(\overline{\mathbf{W}}_{l-1}\cdots\sigma(\overline{\mathbf{W}}_2\sigma(\overline{\mathbf{W}}_1\mathbf{x}_i)\cdots)))$, for all $i \in [n]$ and $l \in [L-1]$. Since $\{\mathbf{a}_l\}_{l=1}^{L-1}$ are non-negative, using eq. (81), we have

$$
\begin{aligned}
\phi_i^l &= \sigma(\overline{\mathbf{W}}_l\sigma(\overline{\mathbf{W}}_{l-1}\cdots\sigma(\overline{\mathbf{W}}_2\sigma(\overline{\mathbf{W}}_1\mathbf{x}_i)\cdots)) \\
&= \sigma(\mathbf{a}_l\mathbf{a}_{l-1}^\top\sigma(\mathbf{a}_{l-1}\mathbf{a}_{l-2}^\top(\cdots\sigma(\mathbf{a}_2\mathbf{a}_1^\top\sigma(\mathbf{a}_1\mathbf{b}_1^\top\mathbf{x}_i))\cdots) \\
&= \|\mathbf{a}_{l-1}\|_2^2\cdots\|\mathbf{a}_1\|_2^2\mathbf{a}_l\sigma^l(\mathbf{b}_1^\top\mathbf{x}_i).
\end{aligned}
$$

Now, using Lemma 16 and since $\|\mathbf{a}_l\|_2^2 = 1/\sqrt{L}$, we have

$$
\begin{aligned}
\lambda\overline{\mathbf{W}}_L &+ \partial_{\mathbf{W}_L}\left(\mathbf{y}^\top\mathcal{H}(\mathbf{X};\overline{\mathbf{W}}_1,\cdots,\overline{\mathbf{W}}_L)\right) \\
&= \lambda q\mathbf{a}_{L-1}^\top/L^{1/4} + \sum_{i=1}^n y_i\left(\phi_i^{L-1}\right)^\top \\
&= \lambda q\mathbf{a}_{L-1}^\top/L^{1/4} + \sum_{i=1}^n y_i\|\mathbf{a}_{L-2}\|_2^2\cdots\|\mathbf{a}_1\|_2^2\mathbf{a}_{L-1}^\top\sigma^{L-1}(\mathbf{b}_1^\top\mathbf{x}_i) \\
&= \left(q\lambda/L^{1/4} + \left(\frac{1}{\sqrt{L}}\right)^{L-2}\sum_{i=1}^n y_i\sigma^{L-1}(\mathbf{b}_1^\top\mathbf{x}_i)\right)\mathbf{a}_{L-1}^\top \\
&= \left(q\lambda/L^{1/4} - \left(\frac{1}{\sqrt{L}}\right)^{L-1}\lambda_*\right)\mathbf{a}_{L-1}^\top = \mathbf{0}.
\end{aligned}
$$

Next, for any $\mathbf{W}_l$, where $2 \le l \le L-1$, we have

$$
\begin{aligned}
\mathcal{H}(\mathbf{x}_i;\overline{\mathbf{W}}_1,\cdots,\mathbf{W}_l,\cdots\overline{\mathbf{W}}_L) &= \overline{\mathbf{W}}_L\sigma(\overline{\mathbf{W}}_{L-1}\cdots\sigma(\mathbf{W}_l\sigma(\overline{\mathbf{W}}_{l-1}\cdots\sigma(\overline{\mathbf{W}}_2\sigma(\overline{\mathbf{W}}_1\mathbf{x}_i)\cdots) \\
&= q\mathbf{a}_{L-1}^\top\sigma(\mathbf{a}_{L-1}\mathbf{a}_{L-2}^\top\cdots\sigma(\mathbf{W}_l\phi_i^{l-1})/L^{1/4} \\
&= q\left(\frac{1}{\sqrt{L}}\right)^{L-3}\sigma^{L-l-1}(\mathbf{a}_l^\top\sigma(\mathbf{W}_l\mathbf{a}_{l-1}\sigma^{l-1}(\mathbf{b}_1^\top\mathbf{x}_i)))/L^{1/4}.
\end{aligned}
$$

Let $\mathbf{w}_{lj}$ denote the $j$-th row of $\mathbf{W}_l$, where $j$ is arbitrary. Choose $\mathbf{W}_l$ such that it equal to $\overline{\mathbf{W}}_l$ except for the $j$-th row, and the $j$-th row is equal to $\mathbf{w}_{lj}$. Then

$$
\begin{aligned}
g_1(\mathbf{w}_{lj}) &=: \mathbf{y}^\top\mathcal{H}(\mathbf{X};\overline{\mathbf{W}}_1,\cdots,\mathbf{W}_l,\cdots\overline{\mathbf{W}}_L) \\
&= q\left(\frac{1}{\sqrt{L}}\right)^{L-3}\frac{1}{L^{1/4}}\sum_{i=1}^n y_i\sigma^{L-l-1}\left(\mathbf{a}_{lj}\sigma(\mathbf{w}_{lj}^\top\mathbf{a}_{l-1}\sigma^{l-1}(\mathbf{b}_1^\top\mathbf{x}_i)) + \sum_{p=1,p\ne j}\mathbf{a}_{lp}^2\|\mathbf{a}_{l-1}\|_2^2\sigma^l(\mathbf{b}_1^\top\mathbf{x}_i)\right).
\end{aligned}
$$

Thus, to prove eq. (92) for $2 \le l \le L-1$, we need to prove

$$
\mathbf{0} \in \lambda\overline{\mathbf{w}}_{lj} + \partial_{\mathbf{w}_{lj}}\left(g_1(\overline{\mathbf{w}}_{lj}),\right) \text{ for all } j \in [k_l].
$$

Now, if $\mathbf{a}_{lj} = 0$, then $\overline{\mathbf{w}}_{lj} = \mathbf{0}$ and $g_1(\mathbf{w}_{lj})$ is independent of $\mathbf{w}_{lj}$, which implies the above equation is satisfied trivially. Now, if $\mathbf{a}_{lj} > 0$, then $\overline{\mathbf{w}}_{lj}^\top\mathbf{a}_{l-1} = \mathbf{a}_{lj}\|\mathbf{a}_{l-1}\|_2^2 > 0$, which implies $\mathbf{w}_{lj}^\top\mathbf{a}_{l-1} > 0$ and hence, $\sigma(\mathbf{w}_{lj}^\top\mathbf{a}_{l-1}\sigma^{l-1}(\mathbf{b}_1^\top\mathbf{x}_i)) = \mathbf{w}_{lj}^\top\mathbf{a}_{l-1}\sigma^l(\mathbf{b}_1^\top\mathbf{x}_i)$, for all $\mathbf{w}_{lj}$ in a sufficiently small neighborhood of $\overline{\mathbf{w}}_{lj}$. Therefore,

$$
\begin{aligned}
g_1(\mathbf{w}_{lj}) &= q\left(\frac{1}{\sqrt{L}}\right)^{L-3}\frac{1}{L^{1/4}}\sum_{i=1}^n y_i\sigma^{L-l-1}\left(\mathbf{a}_{lj}\mathbf{w}_{lj}^\top\mathbf{a}_{l-1}\sigma^l(\mathbf{b}_1^\top\mathbf{x}_i) + \sum_{p=1,p\ne j}\mathbf{a}_{lp}^2\|\mathbf{a}_{l-1}\|_2^2\sigma^l(\mathbf{b}_1^\top\mathbf{x}_i)\right) \\
&= q\left(\frac{1}{\sqrt{L}}\right)^{L-3}\frac{1}{L^{1/4}}\sum_{i=1}^n y_i\sigma^{L-1}(\mathbf{b}_1^\top\mathbf{x}_i)\left(\mathbf{a}_{lj}\mathbf{w}_{lj}^\top\mathbf{a}_{l-1} + \sum_{p=1,p\ne j}\mathbf{a}_{lp}^2\|\mathbf{a}_{l-1}\|_2^2\right),
\end{aligned}
$$

for all $\mathbf{w}_{lj}$ in a sufficiently small neighborhood of $\overline{\mathbf{w}}_{lj}$. The above equation implies

$$\lambda \overline{\mathbf{w}}_{lj} + \partial_{\mathbf{w}_{lj}} g_1(\overline{\mathbf{w}}_{lj}) = \lambda \mathbf{a}_{lj} \mathbf{a}_{l-1} + q \left( \frac{1}{\sqrt{L}} \right)^{L-3} \frac{1}{L^{1/4}} \sum_{i=1}^{n} y_i \sigma^{L-1}(\mathbf{b}_1^\top \mathbf{x}_i) \mathbf{a}_{lj} \mathbf{a}_{l-1}$$

$$= \lambda \mathbf{a}_{lj} \mathbf{a}_{l-1} - q\lambda^* \left( \frac{1}{\sqrt{L}} \right)^{L-2} \frac{1}{L^{1/4}} \mathbf{a}_{lj} \mathbf{a}_{l-1} = \mathbf{0}.$$

Now, for any $\mathbf{W}_1$, we have

$$\mathcal{H}(\mathbf{x}_i; \mathbf{W}_1, \overline{\mathbf{W}}_2, \cdots, \cdots \overline{\mathbf{W}}_L) = \overline{\mathbf{W}}_L \sigma(\overline{\mathbf{W}}_{L-1} \cdots \sigma(\overline{\mathbf{W}}_2 \sigma(\mathbf{W}_1 \mathbf{x}_i) \cdots)$$

$$= q\mathbf{a}_{L-1}^\top \sigma(\mathbf{a}_{L-1}\mathbf{a}_{L-2}\sigma(\cdots \mathbf{a}_2 \mathbf{a}_1^\top \sigma(\mathbf{W}_1 \mathbf{x}_i)/L^{1/4}$$

$$= q \left( \frac{1}{\sqrt{L}} \right)^{L-2} \sigma^{L-2}(\mathbf{a}_1^\top \sigma(\mathbf{W}_1 \mathbf{x}_i))/L^{1/4}.$$

Let $\mathbf{w}_{1j}$ denote the $j$-th row of $\mathbf{W}_1$, where $j$ is arbitrary. Choose $\mathbf{W}_1$ such that it equal to $\overline{\mathbf{W}}_1$ except for the $j$-th row, and the $j$-th row is equal to $\mathbf{w}_{1j}$. Then

$$g_2(\mathbf{w}_{1j}) =: \mathbf{y}^\top \mathcal{H}(\mathbf{X}; \mathbf{W}_1, \overline{\mathbf{W}}_2, \cdots, \cdots \overline{\mathbf{W}}_L)$$

$$= q \left( \frac{1}{\sqrt{L}} \right)^{L-2} \frac{1}{L^{1/4}} \sum_{i=1}^{n} y_i \sigma^{L-2} \left( \mathbf{a}_{1j} \sigma(\mathbf{w}_{1j}^\top \mathbf{x}_i) + \sum_{p=1, p \neq j} \mathbf{a}_{1p}^2 \sigma(\mathbf{b}_1^\top \mathbf{x}_i) \right)$$

Thus, to prove eq. (92) for $l = 1$, we need to prove

$$\mathbf{0} \in \lambda \overline{\mathbf{w}}_{1j} + \partial_{\mathbf{w}_{1j}} \left( g_2(\overline{\mathbf{w}}_{1j}) \right), \text{ for all } j \in [k_1].$$

Now, if $\mathbf{a}_{1j} = 0$, then $\overline{\mathbf{w}}_{1j} = \mathbf{0}$ and $g_2(\mathbf{w}_{1j})$ is independent of $\mathbf{w}_{1j}$, which implies the above equation is satisfied trivially. If $\mathbf{a}_{1j} > 0$, then, using the same approach to get eq. (87), we get

$$\partial_{\mathbf{w}_{1j}} g_2(\overline{\mathbf{w}}_{1j}) = \partial_{\mathbf{w}} g_2(\mathbf{a}_{1j} \mathbf{b}_j) = q\mathbf{a}_{1j} \left( \frac{1}{\sqrt{L}} \right)^{L-2} \frac{1}{L^{1/4}} \partial_{\mathbf{w}} \left( \sum_{i=1}^{n} y_i \sigma^{L-1}(\mathbf{b}_1^\top \mathbf{x}_i) \right). \tag{93}$$

Hence, using the above equation and eq. (91), we have

$$\lambda \overline{\mathbf{w}}_{1j} + \partial_{\mathbf{w}_{1j}} \left( \mathbf{y}^\top \mathcal{H}(\mathbf{X}; \overline{\mathbf{W}}_1, \cdots, \overline{\mathbf{W}}_L) \right) = \lambda \mathbf{a}_{1j} \mathbf{b}_1 + \partial_{\mathbf{w}_{1j}} g_2(\overline{\mathbf{w}}_{1j})$$

$$= \mathbf{a}_{1j} \left( \lambda \mathbf{b}_1 + q \left( \frac{1}{\sqrt{L}} \right)^{L-2} \frac{1}{L^{1/4}} \partial_{\mathbf{w}} \left( \sum_{i=1}^{n} y_i \sigma^{L-1}(\mathbf{b}_1^\top \mathbf{x}_i) \right) \right) \ni \mathbf{0},$$

which completes the proof.

### C.3 Proof of Theorem 4

From Lemma 5 we know
$$p^{L-l}/\hat{p} = \|\mathbf{a}_l\|_2^4, \forall l \in [L-1], \text{ and } \|\overline{\mathbf{w}}\|_2^2 = 1/\hat{p}$$

where $\hat{p} = p^{L-1} + p^{L-2} + \cdots + 1$.

**Case 1** ($\alpha = 1$) : We begin by proving the "only if" part. Since $\sigma(\cdot)$ is $p-$homogeneous,

$$\sigma(c\mathbf{p}) = c^p \sigma(\mathbf{p}), \sigma'(\mathbf{p}) = p\mathbf{p}^{p-1} \text{ and } \sigma(c\mathbf{p}) = \mathbf{p}^p \sigma(c), \tag{94}$$

where $c \in \mathbb{R}$, $\mathbf{p}$ is a vector. Note that, in this case, $\sigma(\cdot)$ is positively and negatively homogeneous. Also, we use $(\sigma^q)'(\cdot)$ to denote the derivative of $\sigma^q(\cdot)$, where $q \in \mathbb{N}$.

Now, from Lemma 5, for all $l \in [L-2]$,

$$|\mathbf{a}_l| = p^{1/4}|\mathbf{b}_{l+1}|, \text{ and } |\mathbf{a}_{L-1}| = (p\hat{p})^{1/4}|\overline{\mathbf{w}}| \tag{95}$$

We next show that, for $l \in [L-1]$, each non-zero entry of $\mathbf{a}_l$ has identical absolute values. Let $\phi_i^l = \sigma(\overline{\mathbf{W}}_l \cdots \sigma(\overline{\mathbf{W}}_1 \mathbf{x}_i) \cdots)$, for $l \in [L-1]$, then, using eq. (94),

$$\phi_i^l = \sigma(\overline{\mathbf{W}}_l \cdots \sigma(\overline{\mathbf{W}}_1 \mathbf{x}_i) \cdots) = \sigma(\mathbf{a}_l \mathbf{b}_l^\top \sigma(\mathbf{a}_{l-1} \mathbf{b}_{l-1}^\top \cdots \sigma(\mathbf{a}_2 \mathbf{b}_2^\top \sigma(\mathbf{a}_1 \mathbf{b}_1^\top \mathbf{x}_i)) \cdots)$$
$$= \mathbf{a}_l^p \sigma(\mathbf{b}_l^\top \sigma(\mathbf{a}_{l-1} \mathbf{b}_{l-1}^\top \cdots \sigma(\mathbf{a}_2 \mathbf{b}_2^\top \sigma(\mathbf{a}_1 \mathbf{b}_1^\top \mathbf{x}_i)) \cdots) = \mathbf{a}_l^p c_i^l,$$

where $c_i^l := \sigma(\mathbf{b}_l^\top \sigma(\mathbf{a}_{l-1} \mathbf{b}_{l-1}^\top \cdots \sigma(\mathbf{a}_2 \mathbf{b}_2^\top \sigma(\mathbf{a}_1 \mathbf{b}_1^\top \mathbf{x}_i)) \cdots)$ is a scalar. Since $(\overline{\mathbf{W}}_1, \cdots, \overline{\mathbf{W}}_L)$ is a non-zero KKT point of eq. (10), from Lemma 16 we have, for $2 \le l \le L-1$,

$$\mathbf{0} = \lambda \overline{\mathbf{W}}_l + \nabla_{\mathbf{W}_l} \left( \mathbf{y}^\top \mathcal{H}(\mathbf{X}; \overline{\mathbf{W}}_1, \cdots, \overline{\mathbf{W}}_L) \right) = \lambda \mathbf{a}_l \mathbf{b}_l^\top + \sum_{i=1}^n \mathbf{e}_i^l \left( \phi_i^{l-1} \right)^\top$$
$$= \lambda \mathbf{a}_l \mathbf{b}_l^\top + \sum_{i=1}^n \mathbf{e}_i^l c_i^{l-1} \left( \mathbf{a}_{l-1}^p \right)^\top. \tag{96}$$

Multiplying the above equation by $\mathbf{a}_l^\top$ from the left gives us

$$0 = \lambda \|\mathbf{a}_l\|_2^2 \mathbf{b}_l^\top + \left( \sum_{i=1}^n \mathbf{a}_l^\top \mathbf{e}_i^l c_i^{l-1} \right) \left( \mathbf{a}_{l-1}^p \right)^\top. \tag{97}$$

The above equation implies that $\mathbf{b}_l$ is parallel to $\mathbf{a}_{l-1}^p$. Now, if $p$ is even, then $\mathbf{a}_{l-1}^p$ is non-negative and thus, the non-zero entries of $\mathbf{b}_l$ must have same sign. Therefore, from eq. (95), we get

$$\mathbf{b}_l = q_l |\mathbf{a}_{l-1}|/p^{1/4}, \text{ for some } q_l \in \{-1, 1\} \text{ and for all } 2 \le l \le L-1.$$

Else, if $p$ is odd, then, $\mathbf{a}_{l-1}^p$ has the same sign as $\mathbf{a}_{l-1}$, and thus, from eq. (95), we get

$$\mathbf{b}_l = q_l \mathbf{a}_{l-1}/p^{1/4}, \text{ for some } q_l \in \{-1, 1\} \text{ and for all } 2 \le l \le L-1.$$

The above equations and eq. (95), implies that $\mathbf{a}_{l-1}^2$ is parallel to $\mathbf{a}_{l-1}^{2p}$. Now, since $p \ge 2$, therefore, for $l \in [L-2]$, the non-zero entries of $\mathbf{a}_l$ have identical absolute values.

Choose $l = L$ in eq. (96), and since $\mathbf{e}_i^L = y_i$, we get

$$\mathbf{0} = \lambda \overline{\mathbf{w}}^\top + \sum_{i=1}^n y_i c_i^{L-1} \left( \mathbf{a}_{L-1}^p \right)^\top. \tag{98}$$

Hence, $\overline{\mathbf{w}}$ is parallel to $\mathbf{a}_{L-1}^p$. If $p$ is even, then $\mathbf{a}_{L-1}^p$ is non-negative and thus, the non-zero entries of $\overline{\mathbf{w}}$ must have same sign. Therefore, from eq. (95), we get

$$\overline{\mathbf{w}} = q_L |\mathbf{a}_{L-1}|/(p\hat{p})^{1/4}, \text{ for some } q_L \in \{-1, 1\}.$$

Else, if $p$ is odd, then, $\mathbf{a}_{L-1}^p$ has the same sign as $\mathbf{a}_{l-1}$, and thus, from eq. (95), we get

$$\overline{\mathbf{w}} = q_L \mathbf{a}_{L-1}/(p\hat{p})^{1/4}, \text{ for some } q_L \in \{-1, 1\}.$$

The above equations and eq. (95), implies that $\mathbf{a}_{L-1}^2$ is parallel to $\mathbf{a}_{L-1}^{2p}$. Since $p \ge 2$, the non-zero entries of $\mathbf{a}_{L-1}$ have identical absolute values.

Next, from the KKT conditions, we have

$$\mathbf{0} = \lambda \overline{\mathbf{W}}_1 + \nabla_{\mathbf{W}_1} \left( \mathbf{y}^\top \mathcal{H}(\mathbf{X}; \overline{\mathbf{W}}_1, \cdots, \overline{\mathbf{W}}_L) \right). \tag{99}$$

We now aim to simplify $\nabla_{\mathbf{W}_1} \left( \mathbf{y}^\top \mathcal{H}(\mathbf{X}; \overline{\mathbf{W}}_1, \cdots, \overline{\mathbf{W}}_L) \right)$. For any $\mathbf{W}_1$, we have

$$
\begin{aligned}
\mathcal{H}(\mathbf{x}_i; \mathbf{W}_1, \overline{\mathbf{W}}_2, \cdots, \overline{\mathbf{W}}_L) &= \overline{\mathbf{W}}_L \sigma(\overline{\mathbf{W}}_{L-1} \cdots \sigma(\overline{\mathbf{W}}_2(\sigma(\mathbf{W}_1 \mathbf{x}_i) \cdots) \\
&= \overline{\mathbf{w}}^\top \sigma(\mathbf{a}_{L-1} \mathbf{b}_{L-1}^\top \cdots \sigma(\mathbf{a}_2 \mathbf{b}_2^\top \sigma(\mathbf{W}_1 \mathbf{x}_i)) \cdots) \\
&= \overline{\mathbf{w}}^\top \mathbf{a}_{L-1} \left( \Pi_{l=1}^{L-3} \sigma^l(\mathbf{b}_{L-l}^\top \mathbf{a}_{L-l-1}^p) \right) \sigma^{L-2}(\mathbf{b}_2^\top \sigma(\mathbf{W}_1 \mathbf{x}_i)).
\end{aligned}
$$

Now,

$$
\nabla_{\mathbf{W}_1} \sigma^{L-2}(\mathbf{b}_2^\top \sigma(\mathbf{W}_1 \mathbf{x}_i)) = (\sigma^{L-2})'(\mathbf{b}_2^\top \sigma(\mathbf{W}_1 \mathbf{x}_i)) \operatorname{diag}(\sigma'(\mathbf{W}_1 \mathbf{x}_i)) \mathbf{b}_2 \mathbf{x}_i^\top,
$$

which implies, using eq. (94),

$$
\begin{aligned}
\nabla_{\mathbf{W}_1} \sigma^{L-2}(\mathbf{b}_2^\top \sigma(\overline{\mathbf{W}}_1 \mathbf{x}_i)) &= (\sigma^{L-2})'(\mathbf{b}_2^\top \mathbf{a}_1^p \sigma(\mathbf{b}_1^\top \mathbf{x}_i)) \operatorname{diag}(\sigma'(\mathbf{a}_1 \mathbf{b}_1^\top \mathbf{x}_i)) \mathbf{b}_2 \mathbf{x}_i^\top \\
&= (\sigma^{L-2})'(\sigma(\mathbf{b}_1^\top \mathbf{x}_i))(\sigma^{L-2})'(\mathbf{b}_2^\top \mathbf{a}_1^p) \sigma'(\mathbf{b}_1^\top \mathbf{x}_i) \operatorname{diag}(\sigma'(\mathbf{a}_1)) \mathbf{b}_2 \mathbf{x}_i^\top \\
&= (\sigma^{L-1})'(\mathbf{b}_1^\top \mathbf{x}_i)(\sigma^{L-2})'(\mathbf{b}_2^\top \mathbf{a}_1^p) \operatorname{diag}(\sigma'(\mathbf{a}_1)) \mathbf{b}_2 \mathbf{x}_i^\top,
\end{aligned}
$$

where the last equality follows by using chain rule on $(\sigma^{L-1})'(\cdot)$. Let $\beta_1 = \overline{\mathbf{w}}^\top \mathbf{a}_{L-1} \left( \Pi_{l=1}^{L-3} \sigma^l(\mathbf{b}_{L-l}^\top \mathbf{a}_{L-l-1}^p) \right)$ and $\beta_2 = (\sigma^{L-2})'(\mathbf{b}_2^\top \mathbf{a}_1^p)$, then

$$
\nabla_{\mathbf{W}_1} \left( \mathbf{y}^\top \mathcal{H}(\mathbf{X}; \overline{\mathbf{W}}_1, \cdots, \overline{\mathbf{W}}_L) \right) = \beta_1 \beta_2 \sum_{i=1}^n y_i (\sigma^{L-1})'(\mathbf{b}_1^\top \mathbf{x}_i) \operatorname{diag}(\sigma'(\mathbf{a}_1)) \mathbf{b}_2 \mathbf{x}_i^\top. \tag{100}
$$

Multiplying eq. (99) by $\mathbf{a}_1^\top$ from the left and using the above equation, we get

$$
\begin{aligned}
\mathbf{0} &= \lambda \|\mathbf{a}_1\|_2^2 \mathbf{b}_1^\top + \beta_1 \beta_2 \sum_{i=1}^n y_i (\sigma^{L-1})'(\mathbf{b}_1^\top \mathbf{x}_i) \mathbf{a}_1^\top \operatorname{diag}(\sigma'(\mathbf{a}_1)) \mathbf{b}_2 \mathbf{x}_i^\top \\
&= \lambda \|\mathbf{a}_1\|_2^2 \mathbf{b}_1^\top + p \beta_1 \beta_2 \mathbf{b}_2^\top \mathbf{a}_1^p \sum_{i=1}^n y_i (\sigma^{L-1})'(\mathbf{b}_1^\top \mathbf{x}_i) \mathbf{x}_i^\top.
\end{aligned} \tag{101}
$$

Now, $\mathbf{u}_*$ is a KKT point of

$$
\max_{\mathbf{u}} \sum_{i=1}^n y_i \sigma^{L-1}(\mathbf{x}_i^\top \mathbf{u}), \text{ such that } \|\mathbf{u}\|_2^2 = \sqrt{p^{L-1}/\hat{p}}, \tag{102}
$$

if $\|\mathbf{u}_*\|_2^2 = \sqrt{p^{L-1}/\hat{p}}$, and there exists $\lambda^*$ such that

$$
\mathbf{0} = \sum_{i=1}^n y_i (\sigma^{L-1})'(\mathbf{x}_i^\top \mathbf{u}_*) \mathbf{x}_i + \lambda^* \mathbf{u}_*. \tag{103}
$$

From eq. (101) and since $\|\mathbf{b}_1\|_2^2 = \sqrt{p^{L-1}/\hat{p}}$, we observe that $\mathbf{b}_1$ is a non-zero KKT point of eq. (102).

We now turn towards proving the "if" part.

Note that $(\overline{\mathbf{W}}_1, \cdots, \overline{\mathbf{W}}_{L-1}, \overline{\mathbf{W}}_L) = (\mathbf{a}_1 \mathbf{b}_1^\top, \cdots, \mathbf{a}_{L-1} \mathbf{b}_{L-1}^\top, \overline{\mathbf{w}}^\top)$. Since $\mathbf{b}_1$ is a non-zero KKT point of

$$
\max_{\mathbf{u}} \sum_{i=1}^n y_i \sigma^{L-1}(\mathbf{x}_i^\top \mathbf{u}), \text{ such that } \|\mathbf{u}\|_2^2 = \sqrt{p^{L-1}/\hat{p}}, \tag{104}
$$

then there exists $\lambda^* \neq 0$ such that

$$
\mathbf{0} = \sum_{i=1}^n y_i (\sigma^{L-1})'(\mathbf{x}_i^\top \mathbf{b}_1) \mathbf{x}_i + \lambda^* \mathbf{b}_1. \tag{105}
$$

Our aim is to show that for some non-zero $\lambda$,

$$\lambda \overline{\mathbf{W}}_l + \nabla_{\mathbf{W}_l}\left(\sum_{i=1}^n \mathbf{y}_i^\top \mathcal{H}(\mathbf{x}_i; \overline{\mathbf{W}}_1, \cdots, \overline{\mathbf{W}}_L)\right) = \mathbf{0}, \forall l \in [L]. \tag{106}$$

Let $\hat{\lambda} = \mathbf{y}^\top \mathcal{H}(\mathbf{X}; \overline{\mathbf{W}}_1, \cdots, \overline{\mathbf{W}}_L)$, then, since $\sum_{i=1}^n y_i \sigma^{L-1}(\mathbf{b}_1^\top \mathbf{x}_i) \neq 0$, we get

$$\mathbf{y}^\top \mathcal{H}(\mathbf{X}; \overline{\mathbf{W}}_1, \overline{\mathbf{W}}_2, \cdots, \overline{\mathbf{W}}_L) = \sum_{i=1}^n y_i \overline{\mathbf{w}}^\top \sigma(\mathbf{a}_{L-1}\mathbf{b}_{L-1}^\top \cdots \sigma(\mathbf{a}_1 \mathbf{b}_1^\top \mathbf{x}_i)\cdots)$$

$$= \overline{\mathbf{w}}^\top \mathbf{a}_{L-1}^p \left(\Pi_{l=1}^{L-2}\sigma^l(\mathbf{b}_{L-l}^\top \mathbf{a}_{L-l-1}^p)\right)\sum_{i=1}^n y_i \sigma^{L-1}(\mathbf{b}_1^\top \mathbf{x}_i) \neq 0.$$

Define $\mathbf{Z}_l = \nabla_{\mathbf{W}_l}\left(\mathbf{y}^\top \mathcal{H}(\mathbf{X}; \overline{\mathbf{W}}_1, \cdots, \overline{\mathbf{W}}_L)\right)$. Since $\mathcal{H}(\mathbf{x}; \mathbf{W}_1, \cdots, \mathbf{W}_L)$ is $p^{L-l}$-homogeneous with respect to $\mathbf{W}_l$, from Lemma 7 we have

$$\text{trace}(\overline{\mathbf{W}}_l^\top \mathbf{Z}_l) = p^{L-l}\mathbf{y}^\top \mathcal{H}(\mathbf{X}; \overline{\mathbf{W}}_1, \cdots, \overline{\mathbf{W}}_L) = p^{L-l}\hat{\lambda}. \tag{107}$$

Now, $\hat{\lambda} \neq 0$ implies $\|\mathbf{Z}_l\|_F \neq 0$, $\forall l \in [L]$. Suppose we are able to show

$$\mathbf{Z}_l/\|\mathbf{Z}_l\|_F = \overline{\mathbf{W}}_l/\|\overline{\mathbf{W}}_l\|_F, \forall l \in [L].$$

Then we can write $\mathbf{Z}_l = \alpha_l \overline{\mathbf{W}}_l$, for some $\alpha_l$. Since $\|\overline{\mathbf{W}}_l\|_F^2 = p^{L-l}/\hat{p}$, from eq. (107), we get

$$p^{L-l}\hat{\lambda} = \text{trace}(\overline{\mathbf{W}}_l^\top \mathbf{Z}_l) = \alpha_l\|\overline{\mathbf{W}}_l\|_F^2 = \alpha_l p^{L-l}/\hat{p}, \tag{108}$$

which implies $\alpha_l = \hat{p}\hat{\lambda}$. Thus, eq. (106) holds for $\lambda = -\hat{p}\hat{\lambda}$. We next complete the proof by showing $\mathbf{Z}_l/\|\mathbf{Z}_l\|_F = \overline{\mathbf{W}}_l/\|\overline{\mathbf{W}}_l\|_F$, for all $l \in [L]$.

For any two matrices (or vectors) $\mathbf{A}, \mathbf{B}$, we say $\mathbf{A} \propto \mathbf{B}$ if $\mathbf{A} = s\mathbf{B}$, for some non-zero scalar $s$. We aim to show $\mathbf{Z}_l \propto \overline{\mathbf{W}}_l$, which will imply $\mathbf{Z}_l/\|\mathbf{Z}_l\|_F = \overline{\mathbf{W}}_l/\|\overline{\mathbf{W}}_l\|_F$, for all $l \in [L]$.

From eq. (100), eq. (105), and using eq. (94), we know

$$\mathbf{Z}_1 \propto \sum_{i=1}^n y_i(\sigma^{L-1})'(\mathbf{b}_1^\top \mathbf{x}_i)\text{diag}(\sigma'(\mathbf{a}_1))\mathbf{b}_2\mathbf{x}_i^\top \propto \text{diag}(\sigma'(\mathbf{a}_1))\mathbf{b}_2\mathbf{b}_1^\top \propto \text{diag}(\mathbf{a}_1^{p-1})\mathbf{b}_2\mathbf{b}_1^\top. \tag{109}$$

If $p$ is odd, then $p-1$ is even. In this case, since $\mathbf{b}_2 = q_l\mathbf{a}_1/p^{1/4}$ and the entries of $\mathbf{a}_1$ have identical absolute values, $\text{diag}(\mathbf{a}_1^{p-1})\mathbf{b}_2 \propto \mathbf{b}_2 \propto \mathbf{a}_1$. Hence, $\mathbf{Z}_1 \propto \mathbf{a}_1\mathbf{b}_1^\top \propto \overline{\mathbf{W}}_1$. Now, if $p$ is even, then $p-1$ is odd. Since $\mathbf{b}_2 = q_1|\mathbf{a}_1|/p^{1/4}$, we have $\text{diag}(\mathbf{a}_1^{p-1})\mathbf{b}_2 \propto \mathbf{a}_1$, which implies $\mathbf{Z}_1 \propto \mathbf{a}_1\mathbf{b}_1^\top \propto \overline{\mathbf{W}}_1$.

Next, as shown in eq. (98), we have

$$\mathbf{Z}_L \propto (\mathbf{a}_{L-1}^p)^\top. \tag{110}$$

If $p$ is odd, then $p-1$ is even. Since the entries of $\mathbf{a}_{L-1}$ have identical absolute values, the entries of $(\mathbf{a}_{L-1}^{p-1})$ are identical and non-negative. Thus, $\mathbf{a}_{L-1}^p = \text{diag}(\mathbf{a}_{L-1}^{p-1})\mathbf{a}_{L-1} \propto \mathbf{a}_{L-1}$. Now, since $\overline{\mathbf{w}} = q_L\mathbf{a}_{L-1}/(p\hat{p})^{1/4}$, we have $\mathbf{a}_{L-1}^p \propto \overline{\mathbf{w}}$, which implies $\mathbf{Z}_L \propto \overline{\mathbf{W}}_L$. Next, if $p$ is even, then $\mathbf{a}_{L-1}^p = |\mathbf{a}_{L-1}|^p$. Since the entries of $\mathbf{a}_{L-1}$ have identical absolute values, $|\mathbf{a}_{L-1}|^p \propto |\mathbf{a}_{L-1}|$. Also, since $\overline{\mathbf{w}} = q_{l-1}|\mathbf{a}_{L-1}|/(p\hat{p})^{1/4}$, we have $\mathbf{a}_{L-1}^p \propto \overline{\mathbf{w}}$, which implies $\mathbf{Z}_L \propto \overline{\mathbf{W}}_L$.

We next consider $\mathbf{Z}_l$, for $2 \leq l \leq L-1$, which requires some additional results. For any $\mathbf{W}_l$, using eq. (94),

$$\mathcal{H}(\mathbf{x}_i; \overline{\mathbf{W}}_1, \cdots, \mathbf{W}_l, \cdots \overline{\mathbf{W}}_L) = \overline{\mathbf{W}}_L\sigma(\overline{\mathbf{W}}_{L-1}\cdots\sigma(\mathbf{W}_l\sigma(\overline{\mathbf{W}}_{l-1}\cdots\sigma(\overline{\mathbf{W}}_1\mathbf{x}_i)\cdots)))$$

$$= \overline{\mathbf{w}}^\top\sigma(\mathbf{a}_{L-1}\mathbf{b}_{L-1}^\top\cdots\sigma(\mathbf{W}_l\phi_i^{l-1}))$$

$$= \overline{\mathbf{w}}^\top\mathbf{a}_{L-1}^p\sigma(\mathbf{b}_{L-1}^\top\mathbf{a}_{L-2}^p)\cdots\sigma^{L-l-2}(\mathbf{b}_{l+2}^\top\mathbf{a}_{l+1}^p)\sigma^{L-l-1}(\mathbf{b}_{l+1}^\top\sigma(\mathbf{W}_l\phi_i^{l-1})),$$

where

$$\phi_i^{l-1} = \sigma(\overline{\mathbf{W}}_{l-1} \cdots \sigma(\overline{\mathbf{W}}_1 \mathbf{x}_i) \cdots) = \sigma(\mathbf{a}_{l-1} \mathbf{b}_{l-1}^\top \sigma(\mathbf{a}_{l-2} \mathbf{b}_{l-2}^\top \cdots \sigma(\mathbf{a}_2 \mathbf{b}_2^\top \sigma(\mathbf{a}_1 \mathbf{b}_1^\top \mathbf{x}_i)) \cdots))$$
$$= \mathbf{a}_{l-1}^p \sigma(\mathbf{b}_{l-1}^\top \sigma(\mathbf{a}_{l-2} \mathbf{b}_{l-2}^\top \cdots \sigma(\mathbf{a}_2 \mathbf{b}_2^\top \sigma(\mathbf{a}_1 \mathbf{b}_1^\top \mathbf{x}_i)) \cdots))$$
$$= \mathbf{a}_{l-1}^p \sigma^{l-1}(\mathbf{b}_1^\top \mathbf{x}_i) \sigma^{l-2}(\mathbf{b}_2^\top \mathbf{a}_1^p) \cdots \sigma(\mathbf{b}_{l-1}^\top \mathbf{a}_{l-2}^p).$$

Hence,

$$\mathbf{Z}_l \propto \mathrm{diag}(\sigma'(\overline{\mathbf{W}}_l \mathbf{a}_{l-1}^p)) \mathbf{b}_{l+1} (\mathbf{a}_{l-1}^p)^\top \propto \mathrm{diag}(\mathbf{a}_l^{p-1}) \mathbf{b}_{l+1} (\mathbf{a}_{l-1}^p)^\top.$$

Now, if $p$ is odd, then $p - 1$ is even. Since the entries of $\mathbf{a}_{l-1}$ have identical absolute values, $(\mathbf{a}_{l-1}^{p-1})$ have identical entries, and thus, $\mathbf{a}_{l-1}^p = \mathrm{diag}(\mathbf{a}_{l-1}^{p-1})\mathbf{a}_{l-1} \propto \mathbf{a}_{l-1}$. Also, since $\mathbf{b}_l = q_{l-1}\mathbf{a}_{l-1}/p^{1/4}$, we have $\mathbf{a}_{l-1}^p \propto \mathbf{b}_l$. Furthermore, since $\mathbf{b}_{l+1} = q_l \mathbf{a}_l/p^{1/4}$ and the entries of $\mathbf{a}_l$ have identical absolute values, $\mathrm{diag}(\mathbf{a}_l^{p-1})\mathbf{b}_{l+1} \propto \mathbf{b}_{l+1} \propto \mathbf{a}_l$. Hence $\mathbf{Z}_l \propto \mathbf{a}_l \mathbf{b}_l^\top \propto \overline{\mathbf{W}}_l$. Next, if $p$ is even, then $\mathbf{a}_{l-1}^p = |\mathbf{a}_{l-1}|^p$. Since the entries of $\mathbf{a}_{l-1}$ have identical absolute values, $|\mathbf{a}_{l-1}|^{p-1}$ have identical entries, and thus, $|\mathbf{a}_{l-1}|^p \propto |\mathbf{a}_{l-1}|$. Also, since $\mathbf{b}_l = q_{l-1}|\mathbf{a}_{l-1}|/p^{1/4}$, we have $\mathbf{a}_{l-1}^p \propto \mathbf{b}_l$. Furthermore, since $\mathbf{b}_{l+1} = q_l|\mathbf{a}_l|/p^{1/4}$, we have $\mathrm{diag}(\mathbf{a}_l^{p-1})\mathbf{b}_{l+1} \propto \mathbf{a}_l$, which implies $\mathbf{Z}_l \propto \mathbf{a}_l \mathbf{b}_l^\top \propto \overline{\mathbf{W}}_l$.

**Case 2** ($\alpha \neq 1$) : We begin by proving the "only if" part. Since $\sigma(\cdot)$ is $p-$positively homogeneous, therefore,

$$\sigma(c\mathbf{p}) = c^p \sigma(\mathbf{p}), \sigma'(\mathbf{q}) = p\mathbf{q}^{p-1} \text{ and } \sigma(d\mathbf{q}) = \mathbf{q}^p \sigma(d), \tag{111}$$

where $c > 0$, $\mathbf{p}$ is a vector, and $d \in \mathbb{R}$, $\mathbf{q}$ is a vector with non-negative entries. Also, we use $(\sigma^q)'(\cdot)$ to denote the derivative of $\sigma^q(\cdot)$, where $q \in \mathbb{N}$.

From Lemma 5, for all $l \in [L - 2]$,

$$|\mathbf{a}_l| = p^{1/4}|\mathbf{b}_{l+1}|, \text{ and } |\mathbf{a}_{L-1}| = (p\hat{p})^{1/4}|\overline{\mathbf{w}}|. \tag{112}$$

Now, since the entries of $\{\mathbf{a}_l\}_{l=1}^{L-1}, \{\mathbf{b}_l\}_{l=2}^{L-1}$ are non-negative, from eq. (112), we have $\mathbf{b}_l = \mathbf{a}_{l-1}/p^{1/4}$, for $2 \leq l \leq L - 1$, and $|\overline{\mathbf{w}}| = \mathbf{a}_{L-1}/(p\hat{p})^{1/4}$.

We next show that, for $l \in [L - 1]$, each non-zero entry of $\mathbf{a}_l$ has identical values. Let $\phi_i^l = \sigma(\overline{\mathbf{W}}_l \cdots \sigma(\overline{\mathbf{W}}_1 \mathbf{x}_i) \cdots)$, for $l \in [L - 1]$, for $1 \leq l \leq L - 1$, then, using eq. (111),

$$\phi_i^l = \sigma(\overline{\mathbf{W}}_l \cdots \sigma(\overline{\mathbf{W}}_1 \mathbf{x}_i) \cdots) = \sigma(\mathbf{a}_l \mathbf{b}_l^\top \sigma(\mathbf{a}_{l-1} \mathbf{b}_{l-1}^\top \cdots \sigma(\mathbf{a}_2 \mathbf{b}_2^\top \sigma(\mathbf{a}_1 \mathbf{b}_1^\top \mathbf{x}_i)) \cdots)$$
$$= \mathbf{a}_l^p \sigma(\mathbf{b}_l^\top \sigma(\mathbf{a}_{l-1} \mathbf{b}_{l-1}^\top \cdots \sigma(\mathbf{a}_2 \mathbf{b}_2^\top \sigma(\mathbf{a}_1 \mathbf{b}_1^\top \mathbf{x}_i)) \cdots) = \mathbf{a}_l^p c_i^l,$$

where $c_i^l := \sigma(\mathbf{b}_l^\top \sigma(\mathbf{a}_{l-1} \mathbf{b}_{l-1}^\top \cdots \sigma(\mathbf{a}_2 \mathbf{b}_2^\top \sigma(\mathbf{a}_1 \mathbf{b}_1^\top \mathbf{x}_i)) \cdots)$ is a scalar. Since $(\overline{\mathbf{W}}_1, \cdots, \overline{\mathbf{W}}_L)$ is a non-zero KKT point of eq. (10), from Lemma 16 we have, for $2 \leq l \leq L - 1$,

$$\mathbf{0} = \lambda \overline{\mathbf{W}}_l + \nabla_{\mathbf{W}_l} \left( \mathbf{y}^\top \mathcal{H}(\mathbf{X}; \overline{\mathbf{W}}_1, \cdots, \overline{\mathbf{W}}_L) \right) = \lambda \mathbf{a}_l \mathbf{b}_l^\top + \sum_{i=1}^n \mathbf{e}_i^l \left( \phi_i^{l-1} \right)^\top$$
$$= \lambda \mathbf{a}_l \mathbf{b}_l^\top + \sum_{i=1}^n \mathbf{e}_i^l c_i^{l-1} \left( \mathbf{a}_{l-1}^p \right)^\top. \tag{113}$$

Multiplying the above equation by $\mathbf{a}_l^\top$ from the left gives us

$$0 = \lambda \|\mathbf{a}_l\|_2^2 \mathbf{b}_l^\top + \left( \sum_{i=1}^n \mathbf{a}_l^\top \mathbf{e}_i^l c_i^{l-1} \right) \left( \mathbf{a}_{l-1}^p \right)^\top. \tag{114}$$

The above equation implies that $\mathbf{b}_l$ is parallel to $\mathbf{a}_{l-1}^p$. Hence, from eq. (112), $\mathbf{a}_{l-1}$ is parallel to $\mathbf{a}_{l-1}^p$. Now, since $p \geq 2$, and $\mathbf{a}_l$ has non-negative entries, we get that, for $1 \leq l \leq L - 2$, the non-zero entries of $\mathbf{a}_l$ are identical.

From the KKT condition for $l = L$, we have

$$\mathbf{0} = \lambda \overline{\mathbf{w}}^\top + \sum_{i=1}^n \mathbf{y}_i c_i^{L-1} \left( \mathbf{a}_{L-1}^p \right)^\top. \tag{115}$$

Hence, $\overline{\mathbf{w}}$ is parallel to $\mathbf{a}_{L-1}^p$. Since $\mathbf{a}_{L-1}$ has non-negative entries, we get that the entries of $\overline{\mathbf{w}}$ have the same sign. Therefore, from $|\overline{\mathbf{w}}| = \mathbf{a}_{L-1}/(p\hat{p})^{1/4}$, we have $\overline{\mathbf{w}} = q\mathbf{a}_{L-1}/(p\hat{p})^{1/4}$, for some $q \in \{-1, 1\}$. Also, $\mathbf{a}_{L-1}$ is parallel to $\mathbf{a}_{L-1}^p$. Since $p \geq 2$, and $\mathbf{a}_{L-1}$ has non-negative entries, the non-zero entries of $\mathbf{a}_{L-1}$ are identical.

Next, from the KKT conditions, we have

$$\mathbf{0} = \lambda \overline{\mathbf{W}}_1 + \nabla_{\mathbf{W}_1} \left( \mathbf{y}^\top \mathcal{H}(\mathbf{X}; \overline{\mathbf{W}}_1, \cdots, \overline{\mathbf{W}}_L) \right). \tag{116}$$

We now aim to simplify $\nabla_{\mathbf{W}_1} \left( \mathbf{y}^\top \mathcal{H}(\mathbf{X}; \overline{\mathbf{W}}_1, \cdots, \overline{\mathbf{W}}_L) \right)$. For any $\mathbf{W}_1$, we have

$$\begin{aligned}
\mathcal{H}(\mathbf{x}_i; \mathbf{W}_1, \overline{\mathbf{W}}_2, \cdots, \overline{\mathbf{W}}_L) &= \overline{\mathbf{W}}_L \sigma(\overline{\mathbf{W}}_{L-1} \cdots \sigma(\overline{\mathbf{W}}_2(\sigma(\mathbf{W}_1 \mathbf{x}_i) \cdots) \\
&= \overline{\mathbf{w}}^\top \sigma(\mathbf{a}_{L-1}\mathbf{b}_{L-1}^\top \cdots \sigma(\mathbf{a}_2 \mathbf{b}_2^\top \sigma(\mathbf{W}_1 \mathbf{x}_i)) \cdots) \\
&= \overline{\mathbf{w}}^\top \mathbf{a}_{L-1} \left( \Pi_{l=1}^{L-3} \sigma^l(\mathbf{b}_{L-l}^\top \mathbf{a}_{L-l-1}^p) \right) \sigma^{L-2}(\mathbf{b}_2^\top \sigma(\mathbf{W}_1 \mathbf{x}_i)).
\end{aligned}$$

Now,

$$\nabla_{\mathbf{W}_1} \sigma^{L-2}(\mathbf{b}_2^\top \sigma(\mathbf{W}_1 \mathbf{x}_i)) = (\sigma^{L-2})'(\mathbf{b}_2^\top \sigma(\mathbf{W}_1 \mathbf{x}_i)) \mathrm{diag}(\sigma'(\mathbf{W}_1 \mathbf{x}_i)) \mathbf{b}_2 \mathbf{x}_i^\top,$$

which implies, using eq. (111),

$$\begin{aligned}
\nabla_{\mathbf{W}_1} \sigma^{L-2}(\mathbf{b}_2^\top \sigma(\overline{\mathbf{W}}_1 \mathbf{x}_i)) &= (\sigma^{L-2})'(\mathbf{b}_2^\top \mathbf{a}_1^p \sigma(\mathbf{b}_1^\top \mathbf{x}_i)) \mathrm{diag}(\sigma'(\mathbf{a}_1 \mathbf{b}_1^\top \mathbf{x}_i)) \mathbf{b}_2 \mathbf{x}_i^\top \\
&= (\sigma^{L-2})'(\sigma(\mathbf{b}_1^\top \mathbf{x}_i))(\sigma^{L-2})'(\mathbf{b}_2^\top \mathbf{a}_1^p) \sigma'(\mathbf{b}_1^\top \mathbf{x}_i) \mathrm{diag}(\sigma'(\mathbf{a}_1)) \mathbf{b}_2 \mathbf{x}_i^\top \\
&= (\sigma^{L-1})'(\mathbf{b}_1^\top \mathbf{x}_i)(\sigma^{L-2})'(\mathbf{b}_2^\top \mathbf{a}_1^p) \mathrm{diag}(\sigma'(\mathbf{a}_1)) \mathbf{b}_2 \mathbf{x}_i^\top,
\end{aligned}$$

where the last equality follows by using chain rule on $(\sigma^{L-1})'(\cdot)$. Let $\beta_1 = \overline{\mathbf{w}}^\top \mathbf{a}_{L-1} \left( \Pi_{l=1}^{L-3} \sigma^l(\mathbf{b}_{L-l}^\top \mathbf{a}_{L-l-1}^p) \right)$ and $\beta_2 = (\sigma^{L-2})'(\mathbf{b}_2^\top \mathbf{a}_1^p)$, then

$$\nabla_{\mathbf{W}_1} \left( \mathbf{y}^\top \mathcal{H}(\mathbf{X}; \overline{\mathbf{W}}_1, \cdots, \overline{\mathbf{W}}_L) \right) = \beta_1 \beta_2 \sum_{i=1}^n y_i (\sigma^{L-1})'(\mathbf{b}_1^\top \mathbf{x}_i) \mathrm{diag}(\sigma'(\mathbf{a}_1)) \mathbf{b}_2 \mathbf{x}_i^\top. \tag{117}$$

Multiplying eq. (116) by $\mathbf{a}_1^\top$ from the left and using the above equation, we get

$$\begin{aligned}
\mathbf{0} &= \lambda \|\mathbf{a}_1\|_2^2 \mathbf{b}_1^\top + \beta_1 \beta_2 \sum_{i=1}^n y_i (\sigma^{L-1})'(\mathbf{b}_1^\top \mathbf{x}_i) \mathbf{a}_1^\top \mathrm{diag}(\sigma'(\mathbf{a}_1)) \mathbf{b}_2 \mathbf{x}_i^\top \\
&= \lambda \|\mathbf{a}_1\|_2^2 \mathbf{b}_1^\top + p \beta_1 \beta_2 \mathbf{b}_2^\top \mathbf{a}_1^p \sum_{i=1}^n y_i (\sigma^{L-1})'(\mathbf{b}_1^\top \mathbf{x}_i) \mathbf{x}_i^\top.
\end{aligned} \tag{118}$$

Now, $\mathbf{u}_*$ is a non-zero KKT point of

$$\max_{\mathbf{u}} \sum_{i=1}^n y_i \sigma^{L-1}(\mathbf{x}_i^\top \mathbf{u}), \text{ such that } \|\mathbf{u}\|_2^2 = \sqrt{p^{L-1}/\hat{p}}, \tag{119}$$

if $\|\mathbf{u}_*\|_2^2 = \sqrt{p^{L-1}/\hat{p}}$, and there exists $\lambda^*$ such that

$$\mathbf{0} = \sum_{i=1}^n y_i (\sigma^{L-1})'(\mathbf{x}_i^\top \mathbf{u}_*) \mathbf{x}_i + \lambda^* \mathbf{u}_*. \tag{120}$$

From eq. (118) and since $\|\mathbf{b}_1\|_2^2 = \sqrt{p^{L-1}/\hat{p}}$, we observe that $\mathbf{b}_1$ is a non-zero KKT point of eq. (119).

We now turn towards proving the "if" condition.

Note that $(\overline{\mathbf{W}}_1, \cdots, \overline{\mathbf{W}}_{L-1}, \overline{\mathbf{W}}_L) = (\mathbf{a}_1 \mathbf{b}_1^\top, \cdots, \mathbf{a}_{L-1} \mathbf{b}_{L-1}^\top, \overline{\mathbf{w}}^\top)$. Since $\mathbf{b}_1$ is a non-zero KKT point of

$$\max_{\mathbf{u}} \sum_{i=1}^n y_i \sigma^{L-1}(\mathbf{x}_i^\top \mathbf{u}), \text{ such that } \|\mathbf{u}\|_2^2 = \sqrt{p^{L-1}/\hat{p}}, \tag{121}$$

then there exists $\lambda^* \neq 0$ such that

$$\mathbf{0} = \sum_{i=1}^n y_i (\sigma^{L-1})'(\mathbf{x}_i^\top \mathbf{b}_1) \mathbf{x}_i + \lambda^* \mathbf{b}_1. \tag{122}$$

Our aim is to show that for some non-zero $\lambda$,

$$\lambda \overline{\mathbf{W}}_l + \nabla_{\mathbf{W}_l} \left( \sum_{i=1}^n \mathbf{y}_i^\top \mathcal{H}(\mathbf{x}_i; \overline{\mathbf{W}}_1, \cdots, \overline{\mathbf{W}}_L) \right) = \mathbf{0}, \forall l \in [L]. \tag{123}$$

Let $\hat{\lambda} = \mathbf{y}^\top \mathcal{H}(\mathbf{X}; \overline{\mathbf{W}}_1, \cdots, \overline{\mathbf{W}}_L)$, then since $\sum_{i=1}^n y_i \sigma^{L-1}(\mathbf{x}_i^\top \mathbf{b}_1) \neq 0$, we get

$$\mathbf{y}^\top \mathcal{H}(\mathbf{X}; \overline{\mathbf{W}}_1, \overline{\mathbf{W}}_2, \cdots, \overline{\mathbf{W}}_L) = \sum_{i=1}^n y_i \overline{\mathbf{w}}^\top \sigma(\mathbf{a}_{L-1} \mathbf{b}_{L-1}^\top \cdots \sigma(\mathbf{a}_1 \mathbf{b}_1^\top \mathbf{x}_i) \cdots)$$

$$= \overline{\mathbf{w}}^\top \mathbf{a}_{L-1}^p \left( \Pi_{l=1}^{L-2} \sigma^l (\mathbf{b}_{L-l}^\top \mathbf{a}_{L-l-1}^p) \right) \sum_{i=1}^n y_i \sigma^{L-1}(\mathbf{b}_1^\top \mathbf{x}_i) \neq 0.$$

Define $\mathbf{Z}_l = \nabla_{\mathbf{W}_l} \left( \mathbf{y}^\top \mathcal{H}(\mathbf{X}; \overline{\mathbf{W}}_1, \cdots, \overline{\mathbf{W}}_L) \right)$. Since $\mathcal{H}(\mathbf{x}; \mathbf{W}_1, \cdots, \mathbf{W}_L)$ is $p^{L-l}$-homogeneous with respect to $\mathbf{W}_l$, therefore, from Lemma 7,

$$\text{trace}(\overline{\mathbf{W}}_l^\top \mathbf{Z}_l) = p^{L-l} \mathbf{y}^\top \mathcal{H}(\mathbf{X}; \overline{\mathbf{W}}_1, \cdots, \overline{\mathbf{W}}_L) = p^{L-l} \hat{\lambda}. \tag{124}$$

Now, $\hat{\lambda} \neq 0$ implies $\|\mathbf{Z}_l\|_F \neq 0$, $\forall l \in [L]$. Suppose we are able to show

$$\mathbf{Z}_l / \|\mathbf{Z}_l\|_F = \overline{\mathbf{W}}_l / \|\overline{\mathbf{W}}_l\|_F, \forall l \in [L].$$

Then we can write $\mathbf{Z}_l = \alpha_l \overline{\mathbf{W}}_l$, for some $\alpha_l$. Since $\|\overline{\mathbf{W}}_l\|_F^2 = p^{L-l}/\hat{p}$, from eq. (124), we get

$$p^{L-l} \hat{\lambda} = \text{trace}(\overline{\mathbf{W}}_l^\top \mathbf{Z}_l) = \alpha_l \|\overline{\mathbf{W}}_l\|_F^2 = \alpha_l p^{L-l}/\hat{p}, \tag{125}$$

which implies $\alpha_l = \hat{p} \hat{\lambda}$. Thus, eq. (123) holds for $\lambda = -\hat{p} \hat{\lambda}$. We next complete the proof by showing $\mathbf{Z}_l / \|\mathbf{Z}_l\|_F = \overline{\mathbf{W}}_l / \|\overline{\mathbf{W}}_l\|_F$, for all $l \in [L]$.

For any two matrices (or vectors) $\mathbf{A}, \mathbf{B}$, we say $\mathbf{A} \propto \mathbf{B}$ if $\mathbf{A} = s\mathbf{B}$, for some non-zero scalar $s$. We aim to show $\mathbf{Z}_l \propto \overline{\mathbf{W}}_l$, which will imply $\mathbf{Z}_l / \|\mathbf{Z}_l\|_F = \overline{\mathbf{W}}_l / \|\overline{\mathbf{W}}_l\|_F$, for all $l \in [L]$.

From eq. (117), eq. (122), and using eq. (111), we know

$$\mathbf{Z}_1 \propto \sum_{i=1}^n y_i (\sigma^{L-1})'(\mathbf{b}_1^\top \mathbf{x}_i) \text{diag}(\sigma'(\mathbf{a}_1)) \mathbf{b}_2 \mathbf{x}_i^\top \propto \text{diag}(\sigma'(\mathbf{a}_1)) \mathbf{b}_2 \mathbf{b}_1^\top \propto \text{diag}(\mathbf{a}_1^{p-1}) \mathbf{b}_2 \mathbf{b}_1^\top. \tag{126}$$

Since $\mathbf{b}_2 = \mathbf{a}_1 / p^{1/4}$ and the entries of $\mathbf{a}_1$ are identical and non-negative, $\text{diag}(\mathbf{a}_1^{p-1}) \mathbf{b}_2 \propto \mathbf{b}_2 \propto \mathbf{a}_1$. Hence, $\mathbf{Z}_1 \propto \mathbf{a}_1 \mathbf{b}_1^\top \propto \overline{\mathbf{W}}_1$.

Next, as shown in eq. (115), we have $\mathbf{Z}_L \propto (\mathbf{a}_{L-1}^p)^\top$. Since $\overline{\mathbf{w}} = q \mathbf{a}_{L-1}/(p\hat{p})^{1/4}$ and the entries of $\mathbf{a}_{L-1}$ are identical and non-negative, $\mathbf{a}_{L-1}^p \propto \mathbf{a}_{L-1} \propto \overline{\mathbf{w}}$. Hence, $\mathbf{Z}_L \propto \overline{\mathbf{W}}_L$.

We next consider $\mathbf{Z}_l$, for $2 \leq l \leq L-1$, which requires some additional results. For any $\mathbf{W}_l$, using eq. (111),

$$
\begin{aligned}
\mathcal{H}(\mathbf{x}_i; \overline{\mathbf{W}}_1, \cdots, \mathbf{W}_l, \cdots \overline{\mathbf{W}}_L) &= \overline{\mathbf{W}}_L \sigma(\overline{\mathbf{W}}_{L-1} \cdots \sigma(\mathbf{W}_l \sigma(\overline{\mathbf{W}}_{l-1} \cdots \sigma(\overline{\mathbf{W}}_1 \mathbf{x}_i) \cdots) \\
&= \overline{\mathbf{w}}^\top \sigma(\mathbf{a}_{L-1} \mathbf{b}_{L-1}^\top \cdots \sigma(\mathbf{W}_l \phi_i^{l-1})) \\
&= \overline{\mathbf{w}}^\top \mathbf{a}_{L-1}^p \sigma(\mathbf{b}_{L-1}^\top \mathbf{a}_{L-2}^p) \cdots \sigma^{L-l-2}(\mathbf{b}_{l+2}^\top \mathbf{a}_{l+1}^p) \sigma^{L-l-1}(\mathbf{b}_{l+1}^\top \sigma(\mathbf{W}_l \phi_i^{l-1})),
\end{aligned}
$$

where

$$
\begin{aligned}
\phi_i^{l-1} = \sigma(\overline{\mathbf{W}}_{l-1} \cdots \sigma(\overline{\mathbf{W}}_1 \mathbf{x}_i) \cdots) &= \sigma(\mathbf{a}_{l-1} \mathbf{b}_{l-1}^\top \sigma(\mathbf{a}_{l-2} \mathbf{b}_{l-2}^\top \cdots \sigma(\mathbf{a}_2 \mathbf{b}_2^\top \sigma(\mathbf{a}_1 \mathbf{b}_1^\top \mathbf{x}_i)) \cdots) \\
&= \mathbf{a}_{l-1}^p \sigma(\mathbf{b}_{l-1}^\top \sigma(\mathbf{a}_{l-2} \mathbf{b}_{l-2}^\top \cdots \sigma(\mathbf{a}_2 \mathbf{b}_2^\top \sigma(\mathbf{a}_1 \mathbf{b}_1^\top \mathbf{x}_i)) \cdots) \\
&= \mathbf{a}_{l-1}^p \sigma^{l-1}(\mathbf{b}_1^\top \mathbf{x}_i) \sigma^{l-2}(\mathbf{b}_2^\top \mathbf{a}_1^p) \cdots \sigma(\mathbf{b}_{l-1}^\top \mathbf{a}_{l-2}^p).
\end{aligned}
$$

Hence,

$$
\mathbf{Z}_l \propto \operatorname{diag}(\sigma'(\overline{\mathbf{W}}_l \mathbf{a}_{l-1}^p)) \mathbf{b}_{l+1} (\mathbf{a}_{l-1}^p)^\top \propto \operatorname{diag}(\mathbf{a}_l^{p-1}) \mathbf{b}_{l+1} (\mathbf{a}_{l-1}^p)^\top.
$$

Since $\mathbf{b}_l = \mathbf{a}_{l-1}/p^{1/4}$ and the entries of $\mathbf{a}_{l-1}$ are identical and non-negative, $\mathbf{a}_{l-1}^p \propto \mathbf{a}_{l-1} \propto \mathbf{b}_l$. Also, since $\mathbf{b}_{l+1} = \mathbf{a}_l/p^{1/4}$ and the entries of $\mathbf{a}_l$ are identical and non-negative, $\operatorname{diag}(\mathbf{a}_l^{p-1}) \mathbf{b}_{l+1} \propto \mathbf{a}_l$. Hence, $\mathbf{Z}_l \propto \mathbf{a}_l \mathbf{b}_l^\top \propto \overline{\mathbf{W}}_l$.

## C.4 Experimental details

This section provides more details about the experiments in this paper. They were conducted using PyTorch Paszke et al. (2019), and the code is submitted along with the paper.

**Training data.** For our experiments, the training data consists of 100 points with inputs in $\mathbb{R}^{10}$, drawn uniformly from the unit-norm sphere, and outputs in $\mathbb{R}$, sampled i.i.d. from $\mathcal{N}(0,1)$.

**Algorithm.** To obtain a KKT point of the constrained NCF, the NCF is maximized using projected gradient ascent with sufficiently small step-size, i.e., after each gradient step, the weights are projected back to the unit sphere. This is primarily done because gradient ascent can diverge to infinity. However, from Lemma 19, it is known that, for homogeneous objectives, projected gradient ascent with fixed learning is equivalent to gradient ascent with adaptive step-size up to a scaling factor. Hence, projected gradient ascent is also a valid method to discretize gradient flow in this case.

**Initialization.** The initial weights are drawn uniformly at random from the unit norm sphere. Since gradient flow can converge to the origin, we only choose those initial weights for which the NCF is positive, which ensures gradient flow diverges to infinity (see proof of Lemma 2 and (Kumar & Haupt, 2024, Lemma 5.3)).

**Stopping criterion.** From Lemma 11, we know that at a first-order KKT point of the NCF, gradient of the NCF will be aligned with the weights. We use this fact as the stopping criterion for our algorithm, that is, we run the projected gradient ascent algorithm until the weights are sufficiently aligned with the gradient of the NCF.

**Remark 2.** *We make some changes to the algorithm when the activation function is not continuously differentiable, such as ReLU or Leaky ReLU. Such activation functions require computation of the Clarke subdifferential. However, the gradient computed using automatic differentiation provided in PyTorch uses the chain rule of function differentiation, which may not give the true Clarke subdifferential for non-smooth functions. As noted in Lemma 15, the true Clarke subdifferential belongs to a set which is obtained using the chain rule. Therefore, when using such gradients, projected gradient ascent may not converge towards a true KKT point which satisfies Lemma 11. To overcome this issue, a line of work has focused on the behavior of stochastic gradient descent Bianchi et al. (2022); Bolte & Pauwels (2021). The authors of Bianchi et al. (2022) show that the constant step-size stochastic gradient descent, where the gradient is computed using automatic differentiation, asymptotically converges to the true KKT point, for almost all initializations. They also derive similar result for stochastic projected gradient descent. Motivated by these results, we run mini-batch stochastic projected gradient ascent for sufficiently long time, when the activation function is not continuously differentiable such as ReLU or Leaky ReLU.*

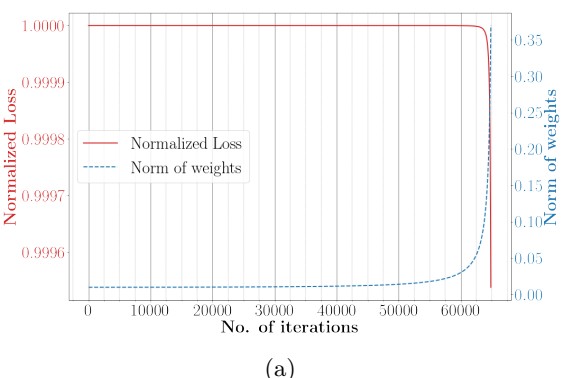 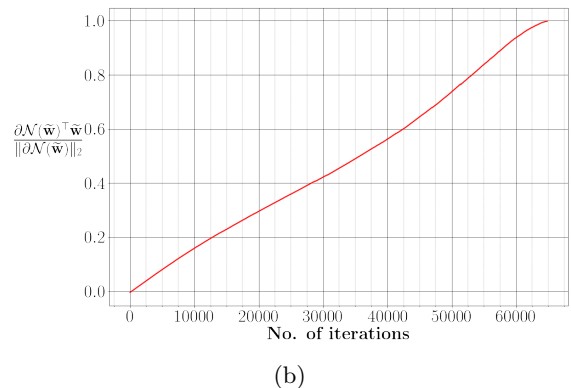

(a) (b)

Figure 4: Early training dynamics of a three layer ReLU neural network. Panel $(a)$: the evolution of training loss (normalized with respect to loss at initialization) and the $\ell_2$-norm of all the weights with iterations. Panel $(b)$: the evolution of $\nabla \mathcal{N}(\widetilde{\mathbf{w}}(t))^\top \widetilde{\mathbf{w}}(t)/\|\nabla \mathcal{N}(\widetilde{\mathbf{w}}(t))\|_2$ (a measure of directional convergence of $\mathbf{w}(t)$). The loss has barely changed, and the norm of weights remains small. Also, the weights approximately converge in direction to a KKT point of the constrained NCF.

## D Challenges for non-differentiable neural networks

Recall that Theorem 1 holds for neural networks with locally Lipschitz gradient, and thus excludes ReLU neural networks. We first provide an empirical evidence that early directional convergence also occurs in ReLU networks, and then briefly describe the challenges in extending Theorem 1 to ReLU networks.

Let $\sigma(x) = \max(x, 0)$. We train the following 3−homogeneous ReLU network $\mathcal{H}(\mathbf{x}; \mathbf{W}_1, \mathbf{W}_2, \mathbf{v}) = \mathbf{v}^\top \sigma(\mathbf{W}_2 \sigma(\mathbf{W}_1 \mathbf{x})))$, where $\mathbf{W}_1 \in \mathbb{R}^{20 \times 20}$, $\mathbf{W}_2 \in \mathbb{R}^{30 \times 20}$ and $\mathbf{v} \in \mathbb{R}^{30}$. For training, we minimize the square loss with respect to the output of a smaller neural network $\mathcal{H}^*(\mathbf{x}; \mathbf{W}_1^*, \mathbf{W}_2^*, \mathbf{v}_*) = 10 \cdot \mathbf{v}_*^\top \sigma(|\mathbf{W}_2^*|\sigma(\mathbf{W}_1^* \mathbf{x}))$, where the entries of $\mathbf{W}_1^* \in \mathbb{R}^{2 \times 20}$, $\mathbf{W}_2^* \in \mathbb{R}^{2 \times 2}$, $\mathbf{v}_* \in \mathbb{R}^{2 \times 1}$ are drawn from standard normal distribution. The training data has 100 points sampled uniformly from unit sphere in $\mathbb{R}^{20}$. We define $\mathbf{w}$ to be the vector that contains all the entries of $\mathbf{W}_1, \mathbf{W}_2$ and $\mathbf{v}$, $\widetilde{\mathbf{w}} = \mathbf{w}/\|\mathbf{w}\|_2$, and $\mathcal{N}(\mathbf{w})$ denotes the NCF for this case. The network is trained for 64900 iterations using gradient descent with step-size $5 \cdot 10^{-3}$, and the initialization $\mathbf{w}(0) = \delta \mathbf{w}_0$, where $\delta = 0.01$ and $\mathbf{w}_0$ is a random unit norm vector. From Figure 4a, we observe that the training loss does not change much, and the norm of the weights increases by a multiplicative factor but remains small. Also, from Figure 4b, it is clear that the weights approximately converge in direction to a KKT point of the constrained NCF. In Figure 5, we plot the normalized absolute value of the weights at initialization and at iteration 64900. Similar to the experiment in Section 4.1.2, low-rank structure emerges among the hidden weights of neural networks. In fact, the two largest singular values of $\mathbf{W}_1/\|\mathbf{w}\|_2$ and $\mathbf{W}_2/\|\mathbf{w}\|_2$ are $(0.5741, 0.0346)$ and $(0.5772, 0.0067)$ respectively, indicating that they have approximately rank one.

We next describe the difficulty in extending Theorem 1 to ReLU networks. Consider a neural network that satisfies Assumption 1, except that the gradient is not assumed to be locally Lipschitz (this will include ReLU networks). Then, assuming square loss, we can define the gradient flow dynamics using the Clarke subdifferential as follows

$$\dot{\mathbf{w}} \in \sum_{i=1}^n (y_i - \mathcal{H}(\mathbf{x}_i; \mathbf{w})) \, \partial \mathcal{H}(\mathbf{x}_i; \mathbf{w}), \mathbf{w}(0) = \delta \mathbf{w}_0. \tag{127}$$

It is also well-known that the Clarke differential of an $L$-homogeneous function is $(L-1)$-homogeneous (Lyu & Li, 2020, Theorem B.2). Thus, if we define $\mathbf{s}(t) = \frac{1}{\delta} \mathbf{w} \left(\frac{t}{\delta^{L-2}}\right)$, then similar to eq. (7), we can show that $\mathbf{s}(t)$ will evolve according to

$$\dot{\mathbf{s}} \in \sum_{i=1}^n (y_i - \delta^L \mathcal{H}(\mathbf{x}_i; \mathbf{s})) \partial \mathcal{H}(\mathbf{x}_i; \mathbf{s}), \mathbf{s}(0) = \mathbf{w}_0. \tag{128}$$

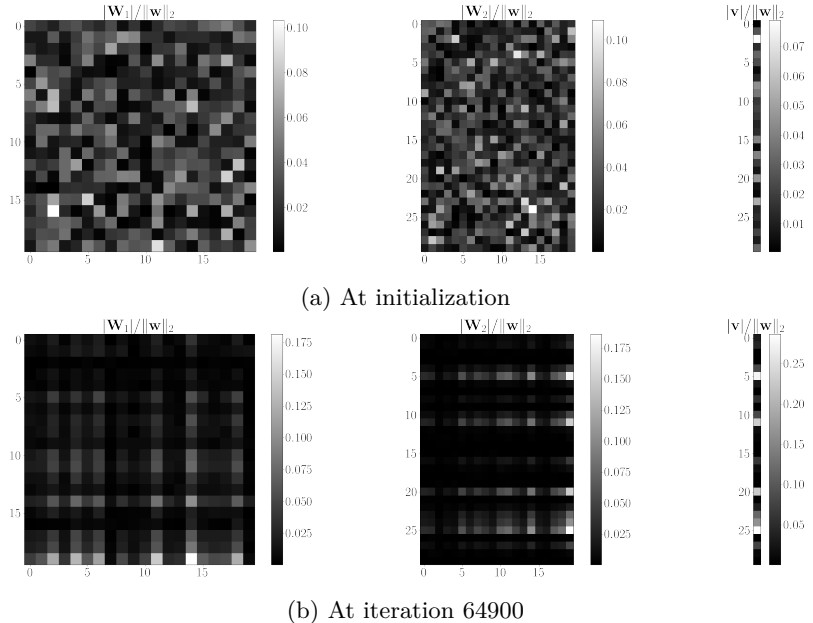

(a) At initialization

(b) At iteration 64900

Figure 5: Normalized absolute value of the weights at different stages of training

Further, the gradient flow for the NCF can be defined by the differential inclusion

$$\dot{\mathbf{u}} \in \sum_{i=1}^{n} y_i \partial \mathcal{H}(\mathbf{x}_i; \mathbf{u}), \mathbf{u}(0) = \mathbf{w}_0. \tag{129}$$

Following the proof sketch of Theorem 1 described in Section 4.1.1, to show approximate directional converegence of $\mathbf{w}(t)$, we would need to first show that for small enough $\delta$, the solutions of eq. (129) and eq. (128) will be sufficiently small for a sufficiently long time. From comparing eq. (129) and eq. (128), it is clear that for small $\delta$, the two solutions will be close in the beginning. However, we *can not* claim that the solutions will remain close for sufficiently long time by choosing small enough $\delta$. In fact, the proof of the analogous result in the context of Theorem 1 crucially relied on gradients being locally Lipschitz, which is not the case here. We believe that proving this would be a crucial step towards establishing directional convergence for ReLU networks, and defer it for future investigations.

## E   Additional lemmata

In the following lemma we show that for two-homogeneous neural networks, the gradient flow dynamics of the NCF remains finite for any finite time.

**Lemma 18.** *Let $\mathcal{H}(\mathbf{x}; \mathbf{w})$ be a two-homogeneous neural network that satisfies first and third property of Assumption 1. Suppose $\mathbf{u}(t)$ is a solution of*

$$\frac{d\mathbf{u}}{dt} = \nabla \mathcal{N}_{\mathbf{z}, \mathcal{H}}(\mathbf{u}) = \mathcal{J}\left(\mathbf{X}; \mathbf{u}\right)^{\top} \mathbf{z}, \mathbf{u}(0) = \mathbf{u}_0. \tag{130}$$

*Then, for any finite time $T \geq 0$, $\|\mathbf{u}(T)\|_2$ is finite.*

*Proof.* Let $\beta = \sup\{\|\mathcal{H}(\mathbf{X}; \mathbf{w})\|_2 : \mathbf{w} \in \mathcal{S}^{k-1}\}$. Then

$$\frac{1}{2}\frac{d\|\mathbf{u}\|_2^2}{dt} = \mathbf{u}^{\top}\dot{\mathbf{u}} = \mathbf{u}^{\top}\mathcal{J}\left(\mathbf{X}; \mathbf{u}\right)^{\top}\mathbf{z} = 2\mathcal{H}\left(\mathbf{X}; \mathbf{u}\right)^{\top}\mathbf{z} \leq 2\beta\|\mathbf{u}\|_2^2\|\mathbf{z}\|_2,$$

where in the last equality we used Lemma 9, and the inequality follows from Cauchy-Schwartz. The above equation implies $\|\mathbf{u}(T)\|_2^2 \leq \|\mathbf{u}_0\|_2^2 + e^{4T\beta\|\mathbf{z}\|_2}$, for $T \geq 0$, which proves our claim. $\qquad\square$

**Lemma 19.** *Let $F(\mathbf{w})$ be a q-homogeneous function, for some $q \geq 2$. Consider the following two sequences:*

$$\mathbf{v}(t+1) = c_t(\mathbf{v}(t) + \eta \nabla F(\mathbf{v}(t))), \mathbf{v}(0) = \mathbf{w}_0, c_t > 0, \forall t \geq 0, \tag{131}$$

*and*

$$\mathbf{u}(t+1) = \mathbf{u}(t) + \eta_t \nabla F(\mathbf{u}(t)), \mathbf{u}(0) = \mathbf{w}_0, \eta_0 = \eta, \eta_t = \eta \left(\Pi_{m=0}^{t-1} c_m\right)^{L-2}, \forall t \geq 0. \tag{132}$$

*Then, for all $T \geq 1$, $\mathbf{v}(T) = \left(\Pi_{t=0}^{T-1} c_t\right) \mathbf{u}(T)$.*

*Proof.* We prove this via induction. The claim is true for $T = 1$, since

$$\mathbf{v}(0) = \mathbf{w}_0 = \mathbf{u}(0), \text{ and } \mathbf{v}(1) = c_0(\mathbf{v}(0) + \eta \nabla \mathbf{F}(\mathbf{v}(0))) = c_0 \mathbf{u}(1).$$

Suppose the claim holds for some $t_0 > 1$. Then, we have

$$\mathbf{v}(t_0) = \left(\Pi_{t=0}^{t_0-1} c_t\right) \mathbf{u}(t_0). \tag{133}$$

Now,

$$\mathbf{u}(t_0 + 1) = \mathbf{u}(t_0) + \eta_{t_0} \nabla F(\mathbf{u}(t_0)).$$

and

$$\begin{aligned}
\mathbf{v}(t_0 + 1) &= c_{t_0}(\mathbf{v}(t_0) + \eta \nabla F(\mathbf{v}(t_0))) \\
&= c_{t_0}\left(\left(\Pi_{t=0}^{t_0-1} c_t\right) \mathbf{u}(t_0) + \eta \left(\Pi_{t=0}^{t_0-1} c_t\right)^{L-1} \nabla F(\mathbf{u}(t_0))\right) \\
&= c_{t_0} \left(\Pi_{t=0}^{t_0-1} c_t\right)\left(\mathbf{u}(t_0) + \eta \left(\Pi_{t=0}^{t_0-1} c_t\right)^{L-2} \nabla F(\mathbf{u}(t_0))\right) \\
&= \left(\Pi_{t=0}^{t_0} c_t\right)(\mathbf{u}(t_0) + \eta_{t_0} \nabla F(\mathbf{u}(t_0))) = \left(\Pi_{t=0}^{t_0} c_t\right) \mathbf{u}(t_0 + 1),
\end{aligned}$$

which completes the proof. $\qquad\square$

If we choose $c_t = 1/\|\mathbf{v}(t) + \eta \nabla F(\mathbf{v}(t))\|_2$ in eq. (132), then eq. (132) becomes the iterates of projected gradient ascent with constant step-size. Therefore, the above lemma shows that, for homogeneous objectives, projected gradient ascent with fixed step-size is equivalent to gradient ascent with adaptive step-size, up to a scaling factor. Consequently, for any finite $T$, the direction of $\mathbf{v}(T)$ and $\mathbf{u}(T)$ will be the same.

