# OpenReview forum: "Early Directional Convergence in Deep Homogeneous Neural Networks for Small Initializations"
_TMLR — Accepted by TMLR_

### Review · Reviewer_ZdJ2 · 2024-12-19

**Summary Of Contributions:**

In this paper, the authors propose a theoretical study of the directional convergence of deep neural networks with small initializations. Within the context of homogeneous neural networks, a common focus in theoretical research, the authors establish that the weights of these networks will theoretically converge in direction towards a KKT point of a constrained optimization problem or remain extremely small. Specifically, the objective function of this optimization problem is the Neural Correlation Function (NCF), which captures the inner product between the output of these neural networks and the gradient of the loss function at zero. Additionally, the authors also propose necessary and sufficient conditions indicating that KKT points of the aforementioned constrained optimization problem consist of rank-one matrices.

**Audience:**

Yes

**Broader Impact Concerns:**

Since this paper focuses on a theoretical phenomenon, there is no need for an ethics review.

**Claims And Evidence:**

Yes

**Requested Changes:**

- Could the author provide more discussions regarding their theoretical findings? Specifically, the interpretation of constrained optimization problem and the NCF function. Since compared to the classic directional convergence results [1, 2], this function and its implication is not straightforward enough.
- I suggest that the authors could consider revising the representation of their Theorems as I indicated in the weaknesses part.

[1] Lyu, K. and Li, J., Gradient Descent Maximizes the Margin of Homogeneous Neural Networks. In International Conference on Learning Representations.

[2] Soudry, D., Hoffer, E., Nacson, M.S., Gunasekar, S. and Srebro, N., 2018. The implicit bias of gradient descent on separable data. Journal of Machine Learning Research, 19(70), pp.1-57.

**Strengths And Weaknesses:**

**Strengths**:
- Comparing to the theoretical study regarding ''NTK'' regime, the theoretical understanding of the regime where the initialization is small is very limited. To the best of my knowledge, this paper is the first to provide an implicit bias analysis on deep neural networks with small initializations. The results of this paper are very interesting and novel, which is a good supplement to the existing results in shallow
- The writing of this paper is clear. Specifically, the proof sketch of Theorem 1 precisely illustrates the intuition of the directional convergence in the early convergence.

**Weaknesses**:
Although I admire the clear writing of this paper, there still exist some minor unclear presentations that make me confused. I'm not sure whether they are my misunderstandings or typos, and I hope the authors could resolve my concerns.
- First of all, I'm a little confused about the factor $1/\delta^{L-2}$ in the definition of $\bar T_\epsilon$. Notice that according to the $\epsilon-\delta$ language in calculus, $\delta$ can be arbitrarily small compared to $\epsilon$. Therefore the first claim in Theorem 1 is a little counterintuitive. Since intuitively we usually have ``the weights are close to the initializations (bounded by a small value) at the initial training stage'', however, as the factor $1/\delta^{L-2}$ could be extremely large, the $\bar T_\epsilon$ actually indicates a long training phase. Is this caused by some properties of homogeneity? For a $L$-homogeneous function $f$, I guess it holds that $x^\top \nabla f(x) = Lf(x)$. But I'm not clear whether such property can necessarily induce the desired result.
- The presentation of Theorem 3 and Theorem 4 are confusing to some extent. Take Theorem 3 as an example, the authors first claim that some properties regarding the norms given the assumption $\bar W_{1:L}$ is a KKT point, and then they provide the necessary and sufficient condition for $\bar W_{1:L}$ being non-zero KKT point. Is this based on the assumption that $\bar W_{1:L}$ is already a KKT point?

---

> ### Author Response · Authors · 2025-02-01
> **Official Comment by Authors**
>
> We thank the reviewer for their detailed feedback.
>
> * (First of all, I'm a little confused about the factor $1/\delta^{L-2}$...) Indeed, the factor of $1/\delta^{L-2}$ causes $\overline{T}_\epsilon$ to become large when $\delta$ is small. Also, if $f(\mathbf{x})$ is $L-$homogeneous, then $\mathbf{x}^\top\nabla f(\mathbf{x}) = Lf(\mathbf{x})$. However, the factor of $1/\delta^{L-2}$ can be attributed to another important property of homogeneous functions: the gradient of an $L$-homogeneous function is $(L-1)-$homogeneous. Specifically, $\nabla f(c\mathbf{x}) = c^{L-1}\nabla f(\mathbf{x}) $, where $c\geq 0$ and $f(\mathbf{x})$ is $L-$homogeneous (see Lemma 7). This property implies that in the initial stages of training, if the norm of the weights is $O(\delta)$, then the norm of the gradient is $O(\delta^{L-1})$. Consequently, gradient flow requires approximately $O(1/\delta^{L-2})$ time to have an impact on the weights. Additionally, we discuss this idea in more detail in the paragraph starting with ``As a quick aside, we note that...'' on page 6.
>
>
> * (Could the author provide more discussions regarding their theoretical findings?...)  In [1,2], the authors showed that in the terminal stages of training, the weights converged towards a KKT point of the margin-maximization problem. However, it is difficult for us to provide a similar interpretation for the constrained NCF problem. For instance, in the case of square loss, the goal of training is to minimize the error between $\mathbf{y}$ and $\mathcal{H}(\mathbf{X};\mathbf{w})$. Finding the set of weights that maximize the correlation between $\mathbf{y}$ and $\mathcal{H}(\mathbf{X};\mathbf{w})$ in the early stages of training does not seem like a bad idea, but hard to explain if it is the right idea. Perhaps, a better understanding of the later training dynamics will help us in explaining the benefit of maximizing the NCF in the early stages of training.
>
>
> * (I suggest that the authors could consider revising the representation of their Theorems....) We understand the cause of confusion. We have edited Theorems 3 and 4 such that the necessary and sufficient condition for $\overline{\mathbf{W}}_{1:L}$ includes the norm constraint.

---

> > ### Author Response · Authors · 2025-03-11
> > **Official Comment by Authors**
> >
> > Dear Reviewer,
> >
> > Please let us know if you have any other questions.
> >
> > Regards,
> > The authors

---

### Review · Reviewer_JVoi · 2024-12-25

**Summary Of Contributions:**

The paper analyzes the gradient flow dynamics of training deep homogenous networks with at least 3 layers. The authors show the gradient flow either exhibits early directional convergence or converges to zero when the initialization is small. They establish the connection between gradient flow on deep networks and the gradient flow on maximizing the neural correlation function (NCF). Moreover, the authors empirically observed that the  gradient flow finds low-rank stationary points of the constrained NCF optimization problem. They also derived necessary and sufficient conditions which charaterize these low-rank  stationary points.

**Audience:**

Yes

**Claims And Evidence:**

Yes

**Requested Changes:**

(1).  I wonder if the authors have done experiments on the gradient flow dynamics of the neural network weights when Assumption 1 is violated (e.g., for ReLu activation)? Does the weights empirically exhibits low-rank structure (or properties similar to those the theorem 1, e.g., directional convergence)?

(2). I wonder if the authors could provide more intuition on how the gradient flow dynamics (early directional convergence or convergence to zero) depends on the initialization. For example, what happens if the normalized initialization $w_0$ is a gaussian vector.

Minor issues:

(1) From the statement of Theorem 1, it is unclear what are the  assumptions required for the theorem to hold.

**Strengths And Weaknesses:**

Strengths: the paper is well-written and the contribution and limitations are discussed in detail. The proof ideas are easy to follow and the empirical observations align with the theoretical findings.

Limitations: as the authors have mentioned, the gradient flow dynamics is characterized only for networks with locally Lipschitz gradients, which rules out ReLU activation. Also, when assumption 1 is not satisfied, there is no guarantee that the gradient flow will find some low-rank weights.

---

> ### Author Response · Authors · 2025-02-01
> **Official Comment by Authors**
>
> We thank the reviewer for their detailed feedback.
>
> * ( I wonder if the authors have done experiments on the gradient flow dynamics ....)  In the revised draft, we have added experiments in Appendix D (Figures 4 and 5) to address this question. These experiments demonstrate early directional convergence among the weights of ReLU neural networks. Additionally, they show the emergence of a low-rank structure in the weights during the early stages of training.
>
>
> * ( I wonder if the authors could provide more intuition ...)  The gradient flow dynamics of the NCF is complex and challenging to analyze. At this stage, it is difficult for us to provide precise intuition on how these dynamics depend on the initialization, such as when using a normalized Gaussian vector. This remains an open question that requires further investigation.
>
>
> * (From the statement of Theorem 1, it is unclear what are the assumptions required for the theorem to hold. ) We have fixed this in the revised draft by adding ``under Assumption 1''.

---

> > ### Comment · Reviewer_JVoi · 2025-02-26
> >
> > Thanks for the response! My questions have been addressed.

---

### Review · Reviewer_Aicg · 2025-01-20

**Summary Of Contributions:**

This paper presents an analysis of gradient flow dynamics in training deep homogeneous neural networks with small initializations. By assuming locally Lipschitz gradients and focusing on networks whose order of homogeneity exceeds two, the authors establish that, in the early phase of training, the network weights remain small in norm and converge directionally toward Karush-Kuhn-Tucker (KKT) points of a so-called “neural correlation function” (NCF). Furthermore, when employing a square loss and under a separability assumption on the weights, the authors observe a similar directional convergence near certain saddle points of the loss function.

**Audience:**

Yes

**Broader Impact Concerns:**

I believe there is no concern on the ethical implication.

**Claims And Evidence:**

Yes

**Requested Changes:**

- Is it possible to derive some explicit convergence rates for the early convergence?
- In Assumption 1, it's better to add that you assume $H$ is first order differentiable.
- Please refer the assumptions in Theorem 1.

**Strengths And Weaknesses:**

### Strengths

- **Comprehensive examination of the initial training phase**
    The paper delves into the dynamics of deep homogeneous network weights during the early stage of training, a period during which weights remain small. This precise focus sheds light on how parameter direction becomes influential even before the network grows to a more conventional operating scale.

- **Identification of directional convergence at KKT points**
    By leveraging the NCF framework, the authors reveal that parameters follow trajectories converging toward the KKT points of this function. This perspective yields a geometrically intuitive explanation for how gradient flow evolves in deep homogeneous networks.

- **Applicability to saddle points**
    Beyond local minima, the analysis extends to examining convergence properties near saddle points of the loss function. Such a broader scope is particularly relevant in deep learning, given the complex critical-point landscape frequently encountered in large-scale networks.

- **Low-rank parameter observations**
    The authors note that, within high-dimensional spaces, certain low-rank behaviors of the parameters can emerge and significantly shape the optimization trajectory. This observation is both intriguing and suggestive of potential reduced-dimensional descriptions of training dynamics.


### Weaknesses

- **Limited quantitative detail**
    The paper principally offers qualitative or semi-quantitative insights into convergence. More rigorous quantitative treatments—such as explicit convergence rates or error bounds—would lend greater depth and practical significance to the findings.

- **Potential oversimplification of the directional aspect**
    While the work emphasizes the direction of convergence, some readers may perceive the conclusion as an overarching average-direction argument. A more in-depth exploration of the subtle or local-scale dynamics underlying this macroscopic behavior could strengthen the contribution.

- **Insufficient discussion of discrete optimization methods**
    Although the continuous-time gradient flow approach offers valuable theoretical clarity, it remains one step removed from standard (stochastic) gradient descent approaches used in practice. Establishing stronger connections between the continuous and discrete realms, possibly including empirical validation, would bolster the paper’s relevance to broader applications.

---

> ### Author Response · Authors · 2025-02-01
> **Official Comment by Authors**
>
> Thank you for the feedback.
>
> *  (Is it possible to derive some explicit convergence rates for the early convergence?)  The duration for the early converegnce is $T_\epsilon/\delta^{L-2}$. Now, the value of $T_\epsilon$ depends on how fast the gradient flow of the NCF converges in direction or becomes $0$, for initialization $\mathbf{w}\_0$. For general initializations, it is not possible to precisely obtain bounds on $T_\epsilon$. Our proofs rely on the Kurdyka-Lojasiewicz inequality (Lemma 6), which establishes the existence of a desingularizing function $\Psi(\cdot)$, where $\Psi(0) = 0$ and $\Psi$ is an increasing function in $(0,\nu)$, for some $\nu>0$. To get a precise value of $T_\epsilon$, we need to know the rate at which $\Psi(\cdot)$ grows near the origin, which is not available in general.
>
>
> * (In Assumption 1, it's better to add that you assume $H$ is first order differentiable.) We have added it in the revised draft.
>
>
> * (Please refer the assumptions in Theorem 1.) We have added it in the revised draft.
>
>
> * (Insufficient discussion of discrete optimization methods) We agree that extending of our results to discrete methods is an important future step, but we also note that the experiments in our work is conducted using discrete methods such as gradient descent. This suggests our results are also likely to hold when using discrete methods.

---

> > ### Comment · Reviewer_Aicg · 2025-02-25
> >
> > Thank you for your reply! It addressed all my questions!

---

### Decision · Action_Editor_2HYN · 2025-03-09

**Recommendation:** Accept as is

**Comment:**

There is unanimous and strong support from the reviewers following the author’s rebuttal and paper revision.

**Audience:**

Yes, this paper falls within the field of deep learning theory and has a broad appeal to the TMLR audience.

**Claims And Evidence:**

This paper investigates gradient flow dynamics in deep homogeneous neural networks with locally Lipschitz gradients and homogeneity greater than two. It shows that for small initializations, the network's weights stay small and approximately align with the Karush-Kuhn-Tucker (KKT) points of the neural correlation function during early training. The paper further derives necessary and sufficient conditions for rank-one KKT points in feed-forward networks with (Leaky) ReLU and polynomial (Leaky) ReLU activations. All claims are rigorously proved and supported by simulation experiments.